# Counterion docking: a general approach to reducing energetic disorder in doped polymeric semiconductors

Miao Xiong[1,2,5], Xin-Yu Deng [1,5], Shuang-Yan Tian[1], Kai-Kai Liu[1], Yu-Hui Fang [2], Juan-Rong Wang[1], Yunfei Wang [3], Guangchao Liu[1], Jupeng Chen[1], Diego Rosas Villalva [4], Derya Baran [4], Xiaodan Gu [3] & Ting Lei [1] ✉

Molecular doping plays an important role in controlling the carrier concentration of organic semiconductors. However, the introduction of dopant counterions often results in increased energetic disorder and traps due to the molecular packing disruption and Coulomb potential wells. To date, no general strategy has been proposed to reduce the counterion-induced structural and energetic disorder. Here, we demonstrate the critical role of non-covalent interactions (NCIs) between counterions and polymers. Employing a computer-aided approach, we identified the optimal counterions and discovered that NCIs determine their docking positions, which significantly affect the counterion-induced energetic disorder. With the optimal counterions, we successfully reduced the energetic disorder to levels even lower than that of the undoped polymer. As a result, we achieved a high n-doped electrical conductivity of over $200\,\text{S}\,\text{cm}^{-1}$ and an eight-fold increase in the thermoelectric power factor. We found that the NCIs have substantial effects on doping efficiency, polymer backbone planarity, and Coulomb potential landscape. Our work not only provides a general strategy for identifying the most suitable counterions but also deepens our understanding of the counterion effects on doped polymeric semiconductors.

Molecular doping has been widely used to increase the charge carrier concentration and reduce the charge injection/extraction barriers in various organic electronic devices including organic field-effect transistors (OFETs)[1,2], organic light-emitting diodes (OLEDs)[3,4], organic thermoelectrics (OTEs)[5–7], and organic photovoltaics (OPVs)[8,9]. For doped organic semiconductors, both doping efficiency and energetic disorder play crucial roles. Doping efficiency determines charge carrier concentrations, while energetic disorder largely affects the charge carrier mobilities[10,11]. Introducing dopant counterions increases the

charge carrier concentration, but at the same time introduces energetic disorder as they often distort molecular conformations and disrupt the molecular packing. Moreover, the Coulomb interactions between the charge carriers and counterions are also important due to the low dielectric constant and the strong charge carrier localization of organic semiconductors. These interactions profoundly affect energetic disorder and, consequently, charge carrier transport[12–15]. Therefore, enhancing doping efficiency and reducing energetic disorder simultaneously is critical for doped organic semiconductors.

[1]Key Laboratory of Polymer Chemistry and Physics of Ministry of Education, School of Materials Science and Engineering, Peking University, Beijing 100871, China. [2]Beijing National Laboratory for Molecular Science, College of Chemistry and Molecular Engineering, Peking University, Beijing 100871, China. [3]School of Polymer Science and Engineering, Center for Optoelectronic Materials and Devices, The University of Southern Mississippi, Hattiesburg, MS 39406, USA. [4]Materials Science and Engineering Program (MSE), Physical Sciences and Engineering Division (PSE), King Abdullah University of Science and Technology (KAUST), Thuwal 23955-6900, Saudi Arabia. [5]These authors contributed equally: Miao Xiong, Xin-Yu Deng. ✉e-mail: tinglei@pku.edu.cn

Recently, Watanabe et al. proposed an ion-exchange doping method that replaced the dopant anion with another anion provided by the ionic liquid, leading to a remarkable increase in doping efficiency[16]. Their findings indicated that larger anions exhibited higher exchange-doping efficiency and thus enhanced electrical conductivity. However, Sirringhaus et al. discovered that the electrical conductivity of many polymers showed little correlation with the counterion size but exhibited a strong correlation with the para-crystalline disorder[17,18]. These studies underscore the importance of doping efficiency and energetic disorder in doped polymeric semiconductors. Nonetheless, the challenge remains in reducing the counterion-induced disorder at high doping levels. Recently, we proposed a "disorder-tolerant" polymer design strategy to tackle this issue[19]. However, this approach relies on the meticulous design and synthesis of new polymers, limiting its general applicability. Therefore,

a more universally applicable strategy is desirable to effectively reduce the energetic disorder in heavily doped polymeric semiconductors.

Compared to doped inorganic semiconductors, where the dopant atoms are covalently bonded, molecular dopants in polymer films are embedded through intermolecular non-covalent interactions (NCIs), such as electrostatic interactions, Coulomb interactions, and van der Waals forces[2,20]. Surprisingly, these intermolecular NCIs between the polymer and counterions have not been given adequate attention in previous studies, and only the size and shape of dopant counterions are considered[16,18,21]. Here, we propose that these intermolecular NCIs between the polymer and counterions are critical for reducing the counterion-induced energetic disorder. Based on this concept, we report a computer-aided strategy to screen a large number of different counterions and try to find the best one for doping (Fig. 1). We found that the NCIs determine the docking positions and distribution of the

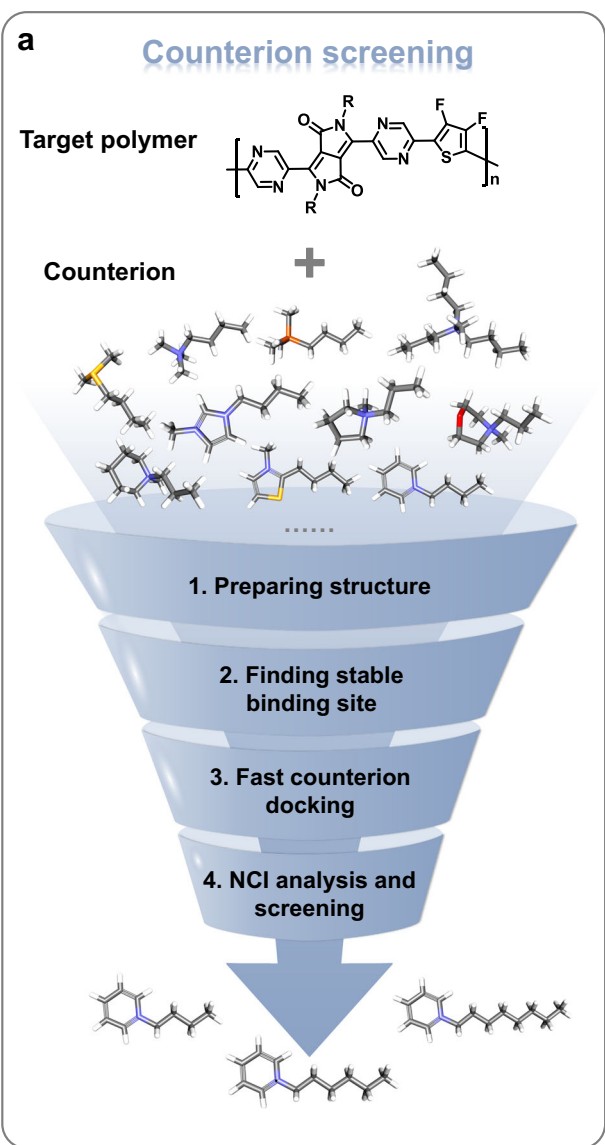

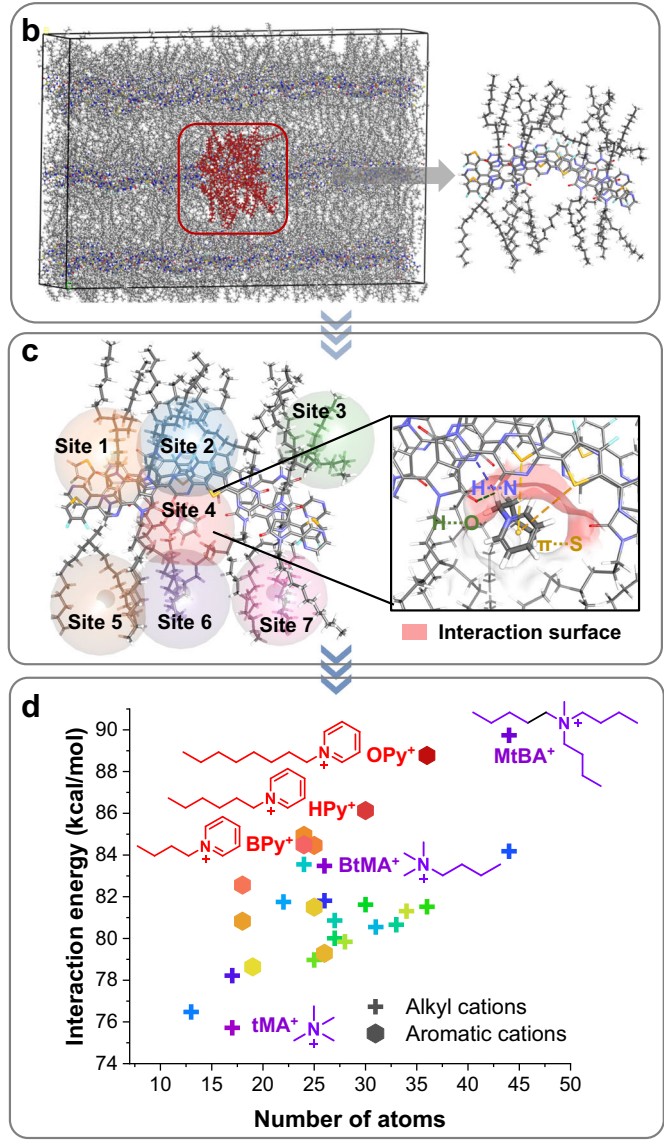

**Fig. 1 | Workflow of the counterion screening method.** The left frame (**a**) provides an overview of the entire screening process, while the right frames (**b**–**d**) offer detailed explanations of each step: (**b**), Step 1 involves preparing the polymer packing models through molecular dynamics simulations. The molecular structure of the polymers was extracted from the MD supercell, as highlighted by the red rectangle. **c** Step 2 comprises identifying potential counterion-binding sites by docking several representative cations into these sites. The site with the highest binding energy is selected as the docking cavity for subsequent screening. Step 3 includes docking all 28 counterions into the selected cavity and generating 100 docking poses for each cation. The interaction surface between the polymer backbone and the counterion is rendered in red. **d** Step 4 focuses on the analysis of NCIs between the polymer and counterions. The insert provides information about the relationship between DFT-calculated interaction energy and the number of atoms in the counterions. Further NCI analysis based on MD simulations is provided in Fig. 2.

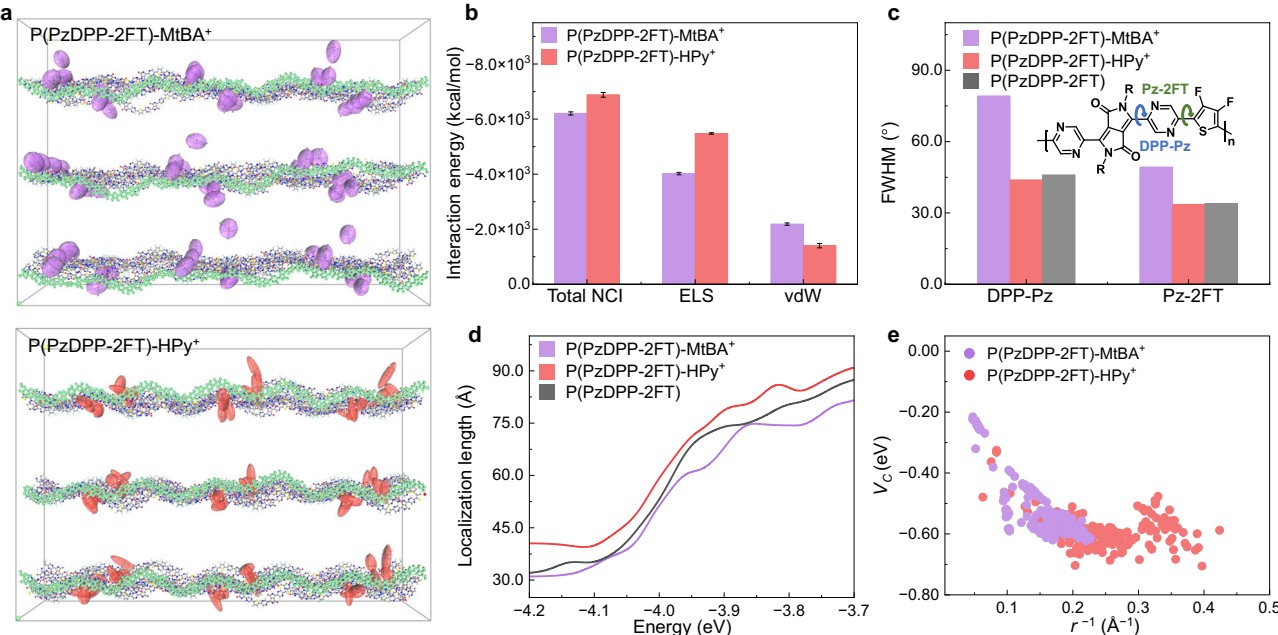

**Fig. 2 | Understanding of the counterion effects on the energetic disorder of the doped polymers. a** MD simulation snapshot of the polymer P(PzDPP-2FT) with MtBA$^+$ and HPy$^+$ (bottom) as the counterions. The polymer backbone in the front is highlighted for better contrast. **b** Energy decomposition analysis (EDA) based on the force field. The electrostatic and van der Waals interactions constitute the total NCIs between 24 polymer chains and 72 counterions within the supercell. Negative values indicate attractive interactions between polymers and counterions. Error bars of interaction energies represent the energy variation during the last 200 ps dynamic simulations. **c** FWHM of the torsion angle distributions for the dihedral angles DPP-Pz and Pz-2FT. **d** Orbital localization length of the undoped and counterion-exchanged P(PzDPP-2FT). **e** Coulomb interaction energies estimated for the polymer-counterion pairs, obtained from MD simulations.

counterions, consequently affecting the doping efficiency, polymer backbone planarity, and Coulomb potential landscape. By docking counterions near the polymer backbone, we effectively reduced the counterion-induced energetic disorder, which led to an ultrahigh n-doped electrical conductivity exceeding 200 S cm$^{-1}$ and an eight-fold enhancement in the thermoelectric power factor of up to 170 μW m$^{-1}$ K$^{-2}$. We also proved that our counterion docking strategy was suitable for other polymer systems, suggesting it is a general approach to reduce the energetic disorder in doped polymeric semiconductors.

## Results and discussion
### Screening the optimal counterion by counterion docking
P-type dopants are commonly strong oxidants or Lewis acids that can be incorporated into the polymer matrix without significantly disrupting the molecular packing, leading to high electrical conductivities[22]. In contrast, commonly used n-type dopants usually possess weak reducing abilities and larger sizes, leading to lower doping efficiency and larger energetic disorder[23]. Consequently, achieving high-efficiency and low-energetic-disorder n-doping is more challenging. Therefore, for this study, we chose to focus on n-doping due to its greater complexity and significance.

Figure 1 depicts the workflow of the counterion screening method. A high-performance n-type polymer, namely P(PzDPP-2FT), was used to exemplify this approach. For n-doped polymers, the counterions are cations. We selected 28 cations from 9 different classes (Fig. S3 and Table S1). First, the three-dimensional polymer packing structure of the targeted polymer was built using molecular dynamic (MD) simulations (Fig. 1b, Step 1)[19,24]. Since n-type dopant counterions were typically inserted into the alkyl sidechain region[25], we explored the potential docking sites within the 3D polymer structure (Fig. 1c, Step 2). We found that counterions docked in site 4 exhibited considerably higher binding energies with the polymer backbone compared to other sites (Fig. S2). Subsequently, we

proceeded to dock the 28 cations into the final refined docking pocket, generating 100 docking poses for each counterion (Step 3). Figure 1c shows the various NCIs formed between the polymer backbone and the counterion. Analyzing the interaction energy between the polymer and counterions (Fig. 1d, Step 4), we observed that counterions possessing longer alkyl chains had more available atoms for binding and thus exhibited stronger affinity with the polymer chains, such as MtBA$^+$ (longer chains) vs. BtMA$^+$ (shorter chains). Interestingly, cations with aromatic structures exhibited stronger interactions compared to alkyl ions with similar sidechain lengths (e.g., BPy$^+$ vs. BtMA$^+$), suggesting different types of NCIs for aromatic rings[26,27].

To gain deeper insights into the NCIs, we then conducted MD simulations of polymer-counterion systems in Step 4 (see Supplementary Information for more details). MtBA$^+$ and HPy$^+$ were selected as the representative alkyl and aromatic cations, respectively. Initially, the counterions were positioned within the refined binding site (site 4), where they exhibited the strongest NCIs with the polymer. Upon reaching equilibrium, a considerable portion of the MtBA$^+$ cations diffused into the alkyl sidechain packing region of the polymers, leading to a broad and multimodal distribution of polymer-counterion distances from 4 to 19 Å (Fig. 2a and Fig. S7c). In contrast, most of the HPy$^+$ ions docked close to the polymer backbone, exhibiting a narrower distribution of distances. These results were unexpected given the higher binding energy of MtBA$^+$ compared to HPy$^+$ (Fig. 1d).

Using energy decomposition analysis (EDA) based on the force field[28,29], we identified NCI compositions as the key factor behind the different counterion docking behaviors between MtBA$^+$ and HPy$^+$. As shown in Fig. 2b, MtBA$^+$ exhibited almost twice higher van der Waals (vdW) interactions with the polymer than HPy$^+$. In contrast, HPy$^+$ showed significantly stronger electrostatic (ELS) interactions, accounting for 80% of its NCIs, compared to 65% for MtBA$^+$ (Fig. S12b). The accumulation of the vdW interactions across multiple neutral polymer sidechains drew MtBA$^+$ into the sidechain regions, while the more substantial ELS interactions made HPy$^+$ dock closer to the

charged polymer backbone (Fig. S12c, d). To validate this conclusion, we conducted NCI analysis on the MD simulation results of another aromatic cation, OPy⁺. We observed that OPy⁺ exhibited a similar docking behavior as HPy⁺, with a preference for docking close to the polymer backbone (Fig. S6c) and an approximately 80% composition of ELS interactions (Fig. S12). These results further confirm that the counterion docking behaviors are determined by the NCI composition.

The docking of dopant counterions has a notable impact on polymer conformations[30–32]. MtBA⁺ in the polymer sidechain region resulted in increased sidechain disorder and greater twisting of the conjugated backbones, while the polymer backbone surrounded by HPy⁺ maintained its zig-zag conformation (Fig. 2a)[19]. We further analyzed the torsion angle distributions of the two dihedral angles: DPP-Pz and Pz-2FT (Fig. 2c and Fig. S15). The results revealed a notable difference for the two counterions: the polymer with MtBA⁺ showed significantly broader torsion angle distributions for both dihedral angles, with full width at half maximum (FWHM) recorded at 79.2° for DPP-Pz and 49.3° for Pz-DPP. In contrast, the polymer with HPy⁺ displayed much narrower torsion angle distributions: 43.9° for DPP-Pz and 33.6° for Pz-DPP. These values are even smaller than those of the undoped polymer, suggesting an enhancement in backbone planarity in HPy⁺ exchange-doped film. The polymer conformation disorder directly influences the electronic structure and charge transport properties. We quantify this effect by calculating the orbital localization length[33,34] (Fig. 2d and Table S3). We observed that the HPy⁺-exchanged polymer showed an extended localization length at the band edge, reaching approximately 40 Å, which is significantly larger than that of the undoped polymer (35 Å) and the polymer with MtBA⁺ (32 Å). These results suggest a reduced energetic disorder in the polymer conformation exchanged with HPy⁺.

In addition to backbone conformation, energetic disorder in doped polymer systems also arises from Coulomb traps, which are strongly associated with the counterion distributions. A recent study used distance-dependent Coulomb interaction energy ($V_C$) to quantify such an effect and prove the importance of quadrupole components in the conductivity of doped small organic molecules[35]. Unlike the symmetric dopant molecules used in the study, which have negligible dipole moments, our system involves counterions with significant and variable dipole moments (Fig. S17). We observed that both the dipole and quadrupole moments of counterions played a crucial role in the electrostatic interactions (Fig. S19). To better understand the electrostatic interactions, we have redefined $V_C$ to include the contributions from monopole, dipole, and quadrupole interactions, assessing their combined effects on the system. As shown in Fig. 2e, $V_C$ increases with decreasing polymer-counterion distances ($r$). However, when $r < 5$ Å ($r^{-1} > 0.2$ Å⁻¹), $V_C$ levels off to a flat electrostatic potential due to the screening effect of the ionic charges[35,36]. Although HPy⁺ docked closer to the polymer backbone, it does not induce deeper Coulomb traps compared to MtBA⁺. Apart from the depth of individual Coulomb traps, the energy barrier for charge carrier hopping strongly correlates with the distance between these traps[10,37]. The stronger NCIs between the polymer backbone and HPy⁺ may contribute to a higher counterion density (we will experimentally prove this later) and consequently a stronger overlap of the Coulomb traps[16,38]. In addition, the counterion docking position also plays a role in the Coulomb potential landscape. The scattered distribution of MtBA⁺ within the polymer leads to a larger average ion-ion distance than those with HPy⁺ (Figs. S7 and S11), which could generate isolated deep Coulomb traps and thus higher barrier heights for carrier transport. In contrast, the shorter ion-ion distance of HPy⁺ can facilitate a strong overlap of the Coulomb traps at high doping concentrations, reducing activation energies for carrier transport. Therefore, by strategically docking counterions to suitable positions, we might be able to fine-tune the overall energetic disorder and optimize the charge transport performance in doped polymeric semiconductors.

## Experimental validation of counterion docking

To experimentally validate the above theoretical analyses, we selected three categories of cations, including alkyl, imidazole, and pyridine cations for experimental studies (Fig. 3a). We use two strong n-dopants tetrakis(dimethylamino)ethylene (TDAE)[39] and bis(cyclopentadienyl)cobalt (CoCp₂)[40] to dope the polymers. Using the ion-exchange doping method, we replaced the dopant counterions with cations from their bis(trifluoromeththylsulfonyl)imide (TFSI⁻) salts, because TFSI⁻ salts generally showed better performance than other anions (Fig. S21).

The electrical conductivities of the polymer films exchange-doped with different categories of cations are presented in Fig. 3b. CoCp₂-doped films exhibited systematically higher electrical conductivities than TDAE-doped films, primarily due to the stronger reducibility of CoCp₂[41,42]. Interestingly, the P(PzDPP-2FT) films doped with either of the two dopants exhibited a similar trend when exchanged with different cations. The P(PzDPP-2FT) films exchanged with alkyl cations consistently displayed lower conductivities of approximately 70 S cm⁻¹, while the films exchanged with imidazole and pyridine cations displayed higher conductivities of up to 170 S cm⁻¹ and over 200 S cm⁻¹, respectively (Fig. 3b). Remarkably, among all the cations, the films exchanged with HPy⁺ exhibited the highest conductivity of 218 S cm⁻¹, which is a four-fold increase over the films exchanged with MtBA⁺ and stands among the highest reported values in n-doped organic semiconductors. Note that the electrical conductivities had little correlation with the size or alkyl chain length of the counterions, indicating that the relationship between counterions and polymer performance is not simply a size or shape dependent effect. These results suggest that aromatic cations are more favorable as the counterions for the doped polymers, which is consistent with the counterion docking screening results.

To assess the generalizability of our findings, we applied the counterion docking method to another widely studied high-performance n-type polymer, oFBDPPV (Section 1 in Supplementary Information). Compared to P(PzDPP-2FT) with a zigzag conformation, oFBDPPV with linear backbones exhibited weaker binding energies to the same counterions (Fig. S2). However, the interaction energies between the 28 cations and oFBDPPV still exhibited the same trend as observed with P(PzDPP-2FT) (Fig. S3). To experimentally validate these findings, we selected three types of cations (MtBA⁺, BMIM⁺, HPy⁺) and conducted conductivity measurements. As shown in Fig. 3c, oFBDPPV showed a significant improvement in electrical conductivities when exchanged with aromatic cations (BMIM⁺ and HPy⁺). Particularly, when exchanged with HPy⁺, its conductivity increased over threefold (from 8.2 to 31.4 S cm⁻¹) compared to N-DMBI doped oFBDPPV[43]. Similarly, we observed a significant conductivity improvement in the exchange-doped N2200, which achieved an almost six-fold increase from 0.007 to 0.042 S cm⁻¹ compared to the N-DMBI doped polymer films[44]. These results suggest that the effect of counterions on conductivity can be observed across different polymeric semiconductors, highlighting the broad applicability of our method. We also investigated the electrical conductivities and thermoelectric performance of P(PzDPP-2FT) after counterion-exchange doping (Fig. S22). P(PzDPP-2FT) exchanged with HPy⁺ achieved an impressive power factor of 171 µW m⁻¹ K⁻², which is one of the highest values among n-doped conjugated polymers (Fig. 3d and Table S4). In comparison, the power factors of P(PzDPP-2FT) exchanged with BMIM⁺ and MtBA⁺ were 103 µW m⁻¹ K⁻² and 23 µW m⁻¹ K⁻², respectively. Recently, many efforts have been devoted to designing polymers with reduced energetic disorder, including designing rigid polymer backbones[45–47] and engineering sidechains[48,49]. However, the meticulous design and synthesis of new polymers can be costly and time-consuming. Using the "counterion docking" strategy, we achieved an eight-fold increase in the thermoelectric performance, representing a promising approach to enhance the thermoelectric performance of conjugated polymers.

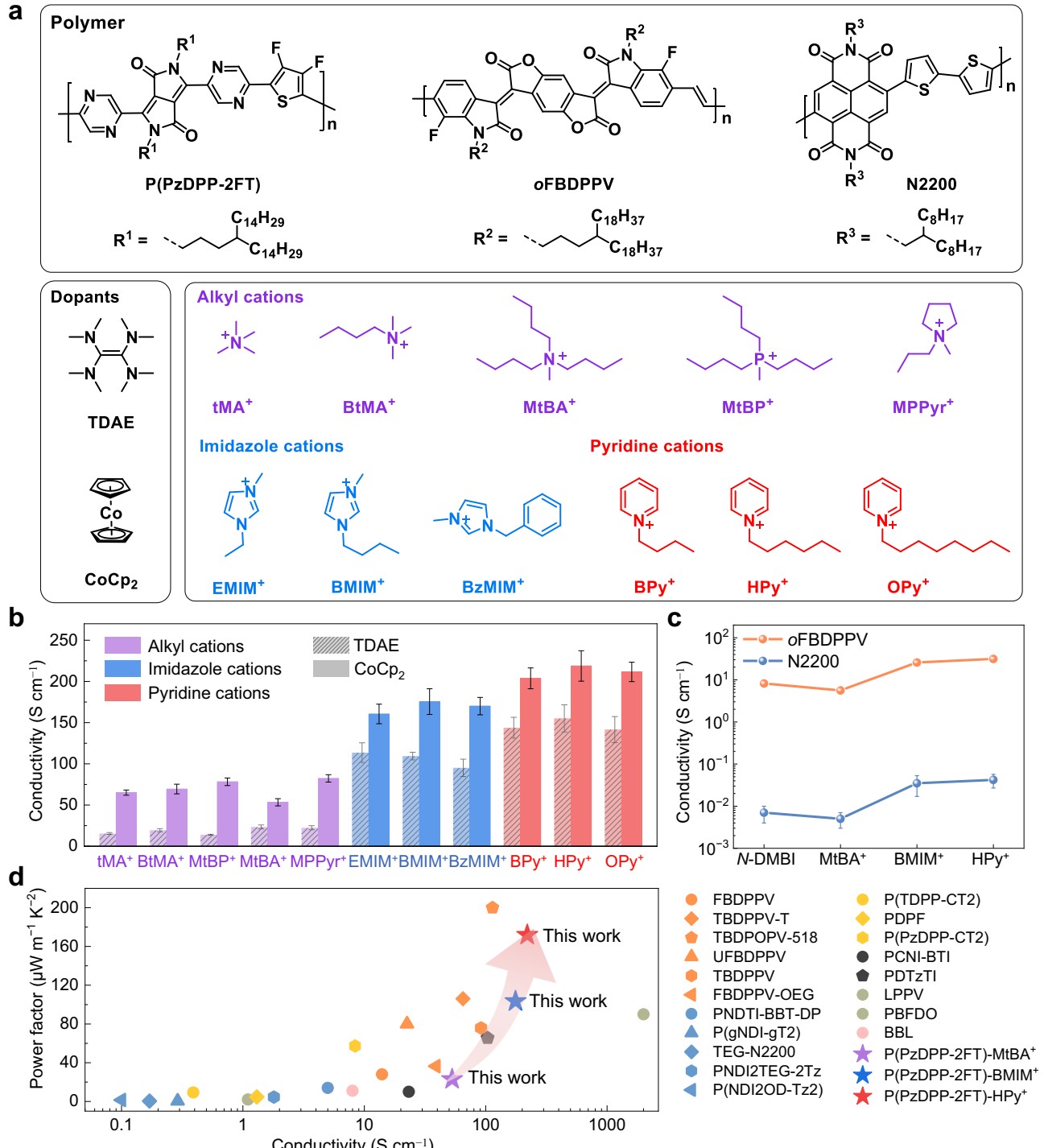

**Fig. 3 | Experimental validation of counterion docking strategy. a** Chemical structures of the polymers (P(PzDPP-2FT), *o*FBDPPV & N2200), n-dopants (TDAE & CoCp₂), and the alkyl, imidazole, and pyridine cations used for this study. **b** Comparison of the electrical conductivities of the ion-exchange doped P(PzDPP-2FT) films. **c** Comparison of the electrical conductivities of the *N*-DMBI-doped and ion-exchange doped *o*FBDPPV and N2200 polymer films. TDAE was utilized as the n-dopant for the ion-exchange doping of *o*FBDPPV and N2200. **d** Comparison of the power factors and conductivities among this work and other reported n-doped polymers. Error bars of conductivity indicate the standard deviation (SD) of eight experimental replicates.

## Understanding of the counterion docking effects

Previous studies have found that electrical conductivity was strongly correlated with paracrystalline disorder[18,50]. However, our investigation using grazing-incidence wide-angle X-ray scattering (GIWAXS) revealed no significant changes in the paracrystallinity (*g*) of P(PzDPP-2FT) exchange-doped with various cations (Fig. 4a). Furthermore, we found no clear correlation between *g* and electrical conductivity.

Atomic force microscopy (AFM) images also supported this, showing no notable differences in the surface roughness of the films exchanged with different cations (Fig. S28). These results indicate that the counterion-exchange did not disrupt the microstructure of the polymer films. Nonetheless, we did observe differences in the GIWAXS results. The lamellar stacking distance of the polymer films expanded after exchange-doped with the three counterions (Fig. 4b, c and

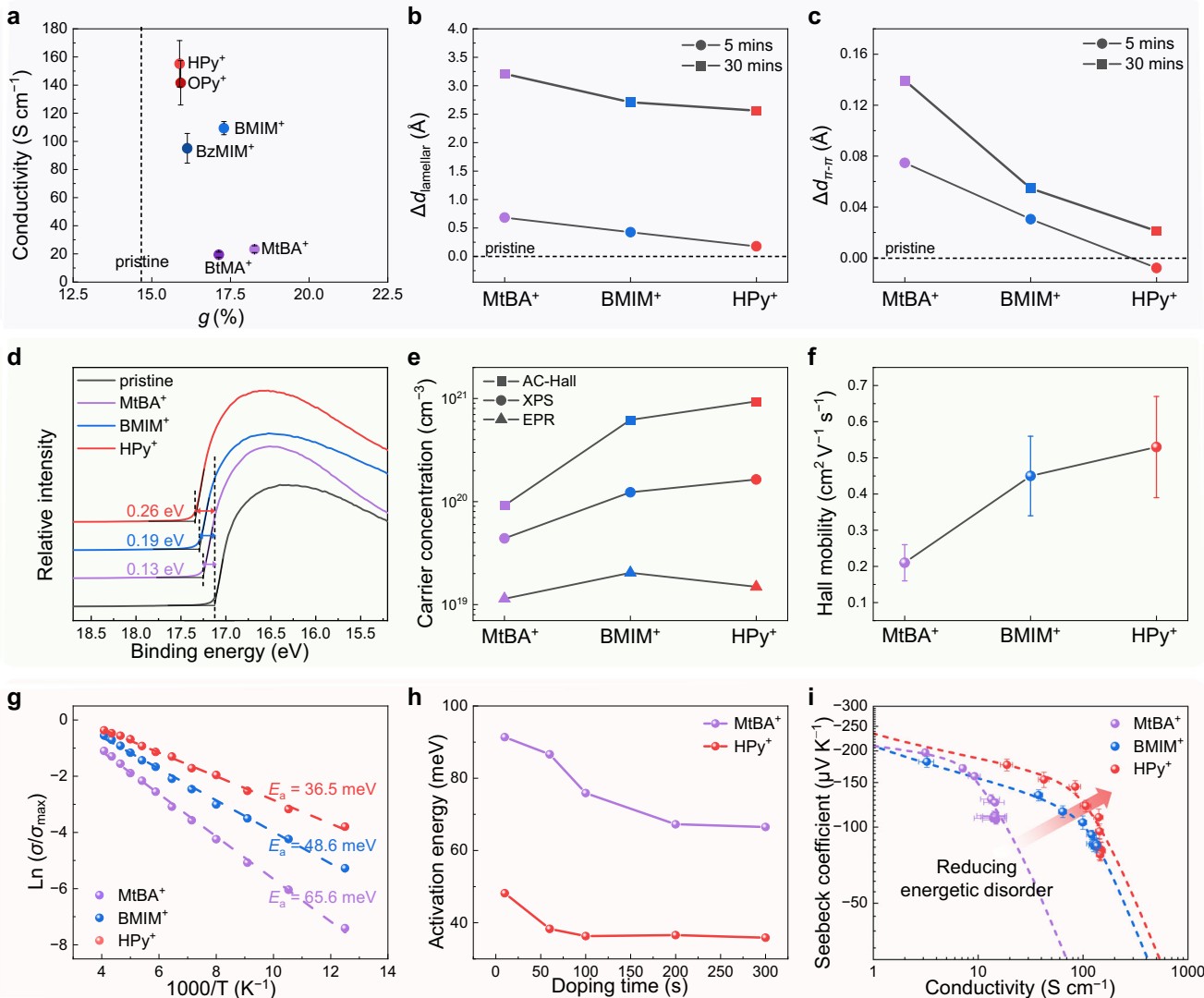

**Fig. 4 | Microstructure, doping efficiency, energetic disorder characterization, and analysis of the exchange-doped polymer films. a** Conductivity vs. π-π stacking paracrystallinity of the P(PzDPP-2FT) film exchange-doped with various cations. Variations in (**b**), lamellar distances, and (**c**), π-π stacking distances of the exchange-doped P(PzDPP-2FT) at 5 and 30 min compared to the pristine polymer film. Data were obtained from the GIWAXS measurements. **d** UPS binding energy of the pristine P(PzDPP-2FT) and the exchange-doped P(PzDPP-2FT). **e** Charge carrier concentration determined by AC-Hall measurement and estimated using XPS and EPR analysis. **f** Hall mobility of the exchange-doped P(PzDPP-2FT) with three cations. **g** Activation energies of the exchange-doped P(PzDPP-2FT) with three cations. **h** Activation energies of the polymer films exchanged with MtBA$^+$ and HPy$^+$ at different doping levels. **i** Fitting the $S$-$\sigma$ relationship using the SLoT model for the polymer films exchanged with three cations. Error bars indicate the SD of eight experimental replicates.

Fig. S27), indicating the presence of counterions within the crystalline regions[17,51]. Particularly, polymer films exchange-doped with HPy$^+$ and BMIM$^+$ consistently exhibited smaller lamellar and π-π stacking distances than those exchange-doped with MtBA$^+$, across different doping times (Fig. 4b, c). This trend was also observed in MD simulations (Fig. S16). The diffusion of MtBA$^+$ into the sidechain regions led to increased sidechain disorder (lamellar expansion) and a greater twisting of the polymer backbone (larger π-π distance). Conversely, HPy$^+$ docking near the polymer backbone contributed to smaller lamellar expansion, and the stronger interactions between the charged polymer backbone and HPy$^+$ contributed to the suppression of backbone torsion and smaller π-π stacking distance. Better backbone planarity and smaller π-π distance could benefit intrachain and interchain charge transport[52]. Conductive AFM (c-AFM) measurements revealed that the films exchanged with MtBA$^+$ displayed isolated conductive regions with limited connectivity and low conductivity, while the films exchanged with HPy$^+$ exhibited more continuous transport channels

with higher conductivity (Fig. S29). These findings suggest that aromatic counterions like HPy$^+$ may lead to a more ordered distribution within the polymer, introducing less disorder to the polymer microstructure, and thereby enhancing charge transport properties. All these results corroborate well with our simulation results and analysis above.

Ultraviolet photoelectron spectroscopy (UPS) analysis showed that the P(PzDPP-2FT) exchange-doped with HPy$^+$ and BMIM$^+$ experienced a larger up-shift in the polymer's Fermi level (0.26 eV for HPy$^+$ and 0.19 eV for BMIM$^+$, respectively) compared to MtBA$^+$ (0.13 eV), suggesting higher doping levels in HPy$^+$ and BMIM$^+$ exchange-doped films (Fig. 4d). To quantitatively assess the charge carrier concentrations ($n$), the alternating current Hall (AC-Hall) measurement was employed (Fig. 4e). The charge carrier concentrations in the film exchange-doped with HPy$^+$ ($9.37 \times 10^{20}$ cm$^{-3}$) and BMIM$^+$ ($6.02 \times 10^{20}$ cm$^{-3}$) were notably higher than that of MtBA$^+$ ($9.13 \times 10^{19}$ cm$^{-3}$). Although the AC-Hall method tends to slightly

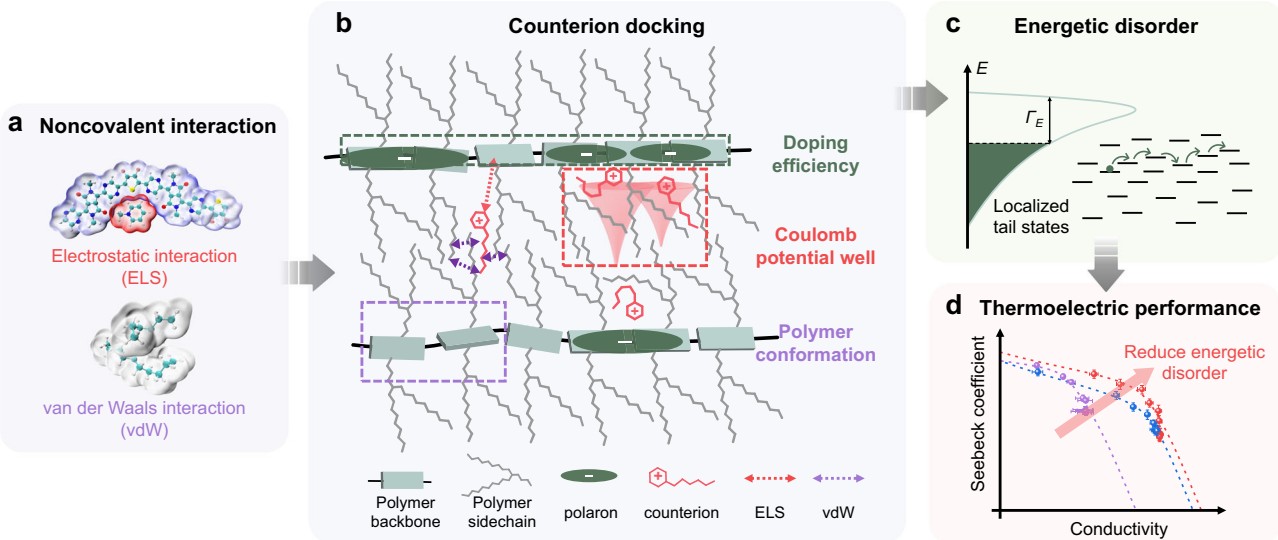

**Fig. 5 | Schematic illustration of how the noncovalent interactions (NCIs) affect the charge transport properties and thermoelectric performance. a** NCIs can be divided into two contributors: electrostatic interactions (ELS) and van der Waals interactions (vdW), which critically influence the docking behavior of counterions. **b** Counterion docking affects the doping efficiency (green dashed rectangle), the distribution of Coulomb potential well (red dashed rectangle), and the polymer backbone conformation (purple dashed rectangle). These three factors collectively influence the (**c**), energetic disorder (sketched based on variable range hopping model[15]), (**d**) charge transport properties, and thermoelectric performance.

overestimate the carrier density due to partial screening by hopping carriers[53], the higher carrier density in the polymer with HPy$^+$ and BMIM$^+$ was further supported by electron paramagnetic resonance (EPR) and X-ray photoelectron spectroscopy (XPS) measurements (Fig. 4e and Table S5). Moreover, the Hall mobility of P(PzDPP-2FT) exchange-doped with HPy$^+$ (0.53 cm$^2$ V$^{-1}$ s$^{-1}$) was twice higher than that of MtBA$^+$ (0.21 cm$^2$ V$^{-1}$ s$^{-1}$) (Fig. 4f). This simultaneous enhancement in charge carrier density and mobility proved the improved exchange-doping efficiency and reduced energetic disorder in the polymer with HPy$^{+}$ [54], as predicted by our simulations.

To quantify the energy barrier of carrier transport, we extracted activation energies ($E_a$) from the temperature-dependent conductivity measurements. The polymer films exchanged with HPy$^+$ and BMIM$^+$ exhibited lower $E_a$ of 36.5 meV and 48.6 meV, respectively, compared to those exchanged with MtBA$^+$ (65.6 meV) (Fig. 4g). Furthermore, at different doping levels, the polymer exchanged with HPy$^+$ consistently displayed lower charge transport barrier heights than that of MtBA$^+$ (Fig. 4h). To analyze the charge carrier transport characteristics, we used the SLoT model based on the Seebeck-conductivity (S-σ) relationship[55] (Fig. 4i and Table S6). Our fitting analysis revealed that the polymer exchanged with HPy$^+$ exhibited the highest transport coefficient ($\sigma_O$), indicating enhanced mobility compared to other cations[56]. Additionally, the depth of the potential wells ($W_H$) in the polymer with HPy$^+$ was found to be zero using the experimentally measured carrier density. This result suggests a stronger overlap of the Coulomb traps, which is closely related to how the counterion is distributed. In counterion docking, we observed that the uniform distribution of HPy$^+$ around the polymer backbone led to a shorter average ion-ion distance ($R$). This observation was supported by experimental results obtained from charge carrier concentrations ($R \approx n^{-3}$)[10], indicating that the $R$ of HPy$^+$ exchanged film was approximately 10 Å smaller than that of MtBA$^+$ (Table S5). Consequently, this reduced $R$ enhances the overlap of Coulomb traps, leading to a lower charge transport barrier height and more efficient charge transport.

## Discussion

Figure 5 illustrates the NCI effects on the charge transport properties and thermoelectric performance in heavily doped polymer films. For aromatic counterions, the stronger ELS interactions stabilize the aromatic counterions around the charged polymer backbone, resulting in enhanced doping efficiency and the suppression of the backbone torsions. Additionally, this uniform counterion distribution facilitates shorter counterion-counterion distances and consequently smoothens the Coulomb potential landscape. In contrast, for alkyl counterions, the pronounced vdW interactions between the alkyl counterions and the polymer sidechains lead to increased sidechain disorder and larger polymer backbone twisting (disturbed by sidechain disorder). Moreover, the alkyl counterion distributions are broad, leading to the formation of isolated Coulomb traps. Therefore, these three factors, doping efficiency, polymer backbone planarity, and Coulomb potential landscape, collectively influence the energetic disorder, and eventually, determine the charge transport properties and thermoelectric performance.

In summary, we propose that the NCIs between counterions and polymers are critical for the charge transport in heavily doped polymer films and report a general computer-aided counterion screening approach to address this issue. Using the best counterion, we achieved a four-fold n-type electrical conductivity and an eight-fold thermoelectric power factor enhancement, demonstrating the effectiveness of our "counterion docking" strategy. Our work also suggests that carefully designed NCIs are an important approach to lowering the energetic disorder in doped polymer films. For instance, introducing specific counterion binding sites near the polymer backbone or on the sidechains could "lock" the counterions and fine-tune the docking position and distribution of counterions, which could help to further reduce the energetic disorder. This work not only presents a general approach for identifying suitable counterions but also provides more insights into the counterion effects in heavily doped polymeric semiconductors.

## Methods
### Materials
All chemical reagents were purchased and used as received unless otherwise indicated. The ion liquids were purchased from TCI. Tetrakis(dimethylamino)ethylene (TDAE) and bis(cyclopentadienyl)cobalt (CoCp$_2$) were purchased from Sigma-Aldrich. P(PzDPP-2FT)[19], oFBDPPV[57], N2200[58] were synthesized according to previous reports.

## Film characterization

UV-vis-NIR absorption spectra were recorded on PerkinElmer Lambda 750 UV-vis spectrometer. The Fourier-transform infrared spectroscopy (FTIR) was recorded on Spectrum Spotlight 200 FT-IR microscopy. The polymer films were spin-coated on the glass substrates, which were evaporated with 5 nm Au. The tests were carried out in reflection mode. UPS and XPS were conducted on a Kratos AXIS Ultra Photoelectron Spectrometer under an ultrahigh vacuum of about $3 \times 10^{-9}$ Torr with unfiltered He I gas discharge lamp source (21.22 eV) and a monochromatic Al Kα source (1486.7 eV, $\theta = 90°$) as the excitation source. Al Kα source operated at 14 kV and 15 mA. The instrumental energy resolution for UPS and XPS were 0.1 eV and 0.5 eV, respectively. Data analysis was performed by CasaXPS software. Continuous-wave EPR was conducted on a Bruker Elexsys E580 spectrometer using ER 4122 SHQE highly sensitive EPR cavity. The microwave frequency is 9.37 GHz. AFM measurements were performed with a Cypher atomic force microscope (Asylum Research, Oxford Instruments). The surface morphology and film thickness were recorded with a scan rate of 2–3 Hz at AC mode. GIWAXS experiment was performed on Xenocs Xuess 2.0 beamline, with an incident X-ray angle of 0.2° and wavelength of 1.54 Å. The scattered signals were collected by a Pilatus 1 M detector at a sample-to-detector distance of 150 mm. Diffraction data analysis was performed in Igor Pro software with Nika and the WAX Tools package.

## Doping and conductivity measurement

All devices for conductivity measurements were fabricated using glass substrates patterned with gold electrodes. The P(PzDPP-2FT) films were deposited onto the substrates by spin-coating the solution at 1500 rpm for 60 s and dried under vacuum. The film thicknesses were determined to be $23 \pm 3$ nm using AFM. The P(PzDPP-2FT) was doped by immersing the films into the dopant solutions for various doping times in the glove box. The doping solutions were prepared by dissolving 0.6 wt% TDAE in the ionic liquids. The films were then washed in $n$-butyl acetate ($n$BA) and dried with a flow of $N_2$. The conductivity was collected by the four-probe measurement method in the glovebox using a Keithley 4200 SCS semiconductor parameter analyzer.

## Thermoelectric properties measurement

The Seebeck measurement was carried out in the $N_2$ glovebox with a Keithley 4200 SCS semiconductor parameter analyzer and self-built equipment which can set the temperature difference. The Seebeck coefficient was calculated by $S = \Delta V_{therm} / \Delta T$. $\Delta V_{therm}$ is the thermal voltage between the hot and cold ends of the device at a temperature gradient of $\Delta T$. The $\Delta V_{therm}$ was monitored by the Keithley 4200 SCS, and the temperature difference was introduced by Joule heat (heater) and a cooling system. To accurately figure out the temperature difference $\Delta T$ between the two contact pads, two temperature sensing wires (5 nm Cr/20 nm Au bilayer) were introduced on the hot and cold ends of the patterned polymer layer. The temperature coefficient of resistance (TCR) of the temperature sensing wires was calculated from the slope of the measured resistance versus temperature. The resistance of the metal wires is linearly correlated with the temperature. TCR was found to be 0.307 $\Omega$ K$^{-1}$ with R$^2$ = 0.9999. By monitoring the resistance evolution of the temperature sensing electrodes, the accurate temperature of the contact pads was obtained by $T_h = T_{r.t.} + (R_h - R_{r.t.})/$TCR and $T_c = T_{r.t.} + (R_c - R_{r.t.})/$TCR. The temperature difference was the difference in temperature between the hot and the cold ends, $\Delta T = T_h - T_c$. The design principle of the device architecture for thermoelectric measurements is according to the criteria proposed by some papers to minimize measurement errors[59].

## Density functional theory calculations

All density functional theory (DFT) calculations were performed using the Gaussian 16 package. Geometry optimizations and single point energy calculations for the complex of the polymer dimer and the counterion were performed at B3LYP/6-31 + G(d,p) level with DFT-D3 dispersion correction. For energy decomposition analysis, the sobEDAw method[60] and Multiwfn[61] were employed. For absorption spectra simulations, time-dependent DFT calculations were performed under the (U)CAM-B3LYP/6-311 G(d,p) level.

## Molecular dynamic simulation

MD simulation was performed with the Materials Studio package. A force field derived from the Deriding force field was employed, where the torsion potentials between adjacent conjugated units were re-parameterized against ωB97X-D/6-311 G(d,p) calculation. Atomic charges were resigned based on RESP charges. The molar ratio of P(PzDPP-2FT) repeating unit and counterion is 4:1. The entire supercell contained 288 polymer repeating units and 72 counterions. The supercells underwent a series of consecutive molecular dynamics, and the final 200 ps of dynamics at 298 K in the NPT ensemble ($P = 1 \times 10^5$ Pa) were extracted for energy decomposition analysis and static analysis for the counterion distribution. Details of the MD simulation procedure are provided in Section 2 of Supplementary Information.

## Counterion docking

We performed counterion docking using the CDOCKER[62] protocol in the Discovery Studio 2019. The counterion docking approach includes the following 3 steps: (1) 3D stacking structure modeling. We extracted three π-stacked dimers from the MD supercell of P(PzDPP-2FT) and $o$FBDPPV. The supercell structure of P(PzDPP-2FT) was based on our previous work[19] and the supercell structure of $o$FBDPPV was provided by Dr. Ze-Fan Yao[23]. (2) To find the most stable binding sites and fast counterion docking, we generated several potential binding sites within the 3D polymer structure and selected the most stable binding sites as the refined docking pocket. We placed the counterion within the refined docking pocket and generated 100 docking poses for each counterion. (3) Validation through DFT calculations. The semiempirical docking approach neglects diatomic differential overlap approximation which is known to underestimate binding affinities significantly[63]. Thus, we supplemented DFT calculations on the most stable binding geometry of the polymer and counterions obtained from docking (at B3LYP/6-31 + G(d,p) level with counterpoise correction). The DFT-derived binding energy of different counterions and P(PzDPP-2FT) followed the same trend of CDOCKER interaction energy, providing evidence for the credibility of the docking method.

## Data availability

The data that support the findings of this study are available from the corresponding authors upon request.

## Code availability

The code that supports the findings of this study is available from the corresponding authors upon request.

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

## Acknowledgements

This work is supported by National Key R&D Program of China (2022YFE0130600), National Natural Science Foundation of China (22075001 and 92156019), and the King Abdullah University of Science and Technology (KAUST) Competitive Research Grants under Award No. ORA-CRG10-2021-4668. We thank Dr. Ze-Fan Yao for providing the crystal structure of oFBDPPV. Y.W. and X.G. thank the KAUST Office of Sponsored Research (OSR) under Award No. OSA-CRG2021- 4668 for X-ray measurements. The computational part is supported by High-performance Computing Platform of Peking University. We acknowledge Molecular Materials and Nanofabrication Laboratory (MMNL) in the College of Chemistry and Electron Microscopy Laboratory of Peking University for the use of instruments.

## Author contributions

T.L., M.X. and X.-Y.D. conceived the project. M.X. designed the experiments and performed device fabrication. X.-Y.D. performed the theoretical calculations and the data analysis. S.-Y.T. helped to perform some of the calculations. K.-K.L., G.L. and J.P.C. synthesized the polymers. X.-Y.D. and Y.-H.F. performed the EPR measurements. J.-R.W. helped to carry out some characterizations. Y.W. and X.G. performed the GIWAXS measurements. D.R.V. and D.B. provided some insightful discussions on thermoelectric measurements. M.X., X.-Y.D. and T.L. wrote the manuscript. All the authors revised and approved the manuscript.

## Competing interests

The authors declare no competing interests.
