## [Peer Review File · Nature Communications]

Counterion docking: a general approach to reducing energetic disorder in doped polymeric semiconductorsReviewers' comments:

Reviewer #1 (Remarks to the Author):

I grew increasingly enthusiastic about the manuscript by Xiong et.al. which demonstrates a simulation driven strategy to improve the mobility of polymeric semiconductors by counterion doping. Through a series of computational steps, the authors are able to rationalize the positioning of the counterions with respect to the polymer backbone, which is non-trivial in a system of such complexity. Based on the computational results they are able to characterize the interaction strength and resulting a different localization of the counterions.

They could then show experimentally that the HPy doped system significantly outperformed the MtBA based systems. They could also show, which is highly laudable, that there were no detectable changes in the morphology that are responsible for this change. They could also demonstrate that the nature of the polaron population is different for the 2 cations. Based on these findings, which I find all supported but the evidence provided, the authors conclude that "NCIs between counterions and polymers are critical" and at the moment I am not convinced that this is really the case. Reference 27 in the manuscript discusses at length the relevance of the quadrupole moment suppressing in some mysterious way the Coulomb-screening, a fact that the authors also clearly demonstrate in Fig 2e. While the relevance of the quadrupole moment is discussed in the docking procedure, its magnitude is not discussed with respect to its influence on the conductivity. Indeed, the quadrupole moment of HPy is 4times higher than that of MtBA, which would fully rationalize the observed difference in the conductivity without any need to consider the interaction strength.

Therefore, I am not convinced that on the data presented we can really conclude that the NCI are more important than the quadrupole moment. In light of this question the authors should correlate the conductivity in Fig3b with the quadrupole moment of the dopants and if they see a strong correlation there, they might have to reconsider their conclusion. Alternatively they should highlight a case of two dopants which have similar quadrupole moment, very different conductivity and very different NCI strength.

I think this manuscript should not be published until this point is resolved.

Reviewer #2 (Remarks to the Author):

The authors have systematically presented experimental and theoretical studies on dopants in n-type conjugated polymers, and have successfully revealed its counterion effects. Although the trend in doping efficiency depending on the ionic species has been studied qualitatively in precious literature, the authors has constituted a key-step forward in quantitative understandings particularly about the energy landscape in doped polymers. The theoretical studies presented in Figs. 1 and 2 have given unambiguous insight about how different dopants having different conformation and size affect on the energetic disorders in conjugated polymers.

In addition, the most of experimental studies presented in the manuscript have supported the

theoretical prediction.

The reviewer highly appreciates this study, and in principle agrees with the publication. However, the main claim “counterion docking” mechanism has to be clarified more.

1. The authors stated “Interestingly, some counterions can also “lock” the polymer backbone and reduce the backbone rotational disorder, leading to even smaller backbone energetic disorder than the undoped polymers.”

This conclusion was not been experimentally supported.

In doped polymers, the reviewer agrees that the majority of electron carriers forms condensed polarons.

Equivalently, the anionized pi-conjugated backbone turns into quinoid, where all torsion in segments should be spontaneously suppressed.

Did author calculate torsion angle distribution in anionized polymer-dopant pairs in Fig. S8?

If no, the calculation in Fig. S8 does not given any supportive evidence of “counterion docking”.

2. The authors have argued the potential crossover in electronic states from single polarons, to bipolarons, then to compressed polarons in Fig. 5.

Judging from the Figs. 5c and 5g, it seems that the majority of electron carrier forms bipolarons.

However, no characteristic absorption was found in Fig.5a.

As has been explained in the ref. 47, it is difficult to distinguish compressed polarons and bipolarons by the electron spin resonance.

The authors should clarify this more in details.

In addition, is “EPR intensity” a mere peak-to-peak intensity or integrated intensity?

Normally, the integrated intensity has to be plotted when spin susceptibility is argued.

In the reviewer’s opinion, details of electronic states, i.e., polarons, bipolarons, or condensed polarons, are beyond of this study. The author needs extra cares to identify the electronic states, for example temperature dependence measurements for EPR.

3.

Reviewer #3 (Remarks to the Author):

Xiong et. all provide a detailed study of the influence of non-covalent interactions between different counterions and polymers and their influence on thermoelectric parameters. Both the topic, the approach and to some extent the results are interesting and the authors make use of a variety of methods both in experimental and modeling. Despite all this, we do not consider the manuscript suitable for publication in Nature Communications. The reason is that it has both conceptual and methodological weaknesses that we do not believe are easily resolved, especially since there are mutually contradictory arguments throughout the manuscript. Our more detailed rationale for this statement can be found below.

1. The authors build much of their argumentation and interpretation of the measured experimental data on the results of their MD and DFT simulations. Unfortunately, the results of the simulations are not verified experimentally, e.g. no experiments are shown which prove that the dopants really dock at the positions predicted by the simulations of the authors. However, since core statements of the authors as well as the entire following interpretation of experiments are almost exclusively based on these results of these simulations, it would be

absolutely necessary from our point of view to verify them (at least partially) also experimentally. Instead, the conclusions drawn from the simulation results with regard to conductivity are only very indirect. From our point of view, the data shown cannot prove beyond doubt that there really is a connection between counterion docking and thermoelectric parameters, especially since other options apart from paracrystalline disorder are largely disregarded.

In general, we have difficulty following core statements of the authors such as "MtBA+ tends to generate deeper Coulomb traps due to its smaller Q" using the given data, for the following reasons:

2. Regarding the claims MtBA+ would generate deeper Coulomb traps (see e.g. page 7, line 166):

2.1. We cannot reconstruct from the authors' data how the characteristic depth of trap states E_b is actually defined and determined and how it correlates with the DOS shown in Fig. 2d. In particular, no significant differences in broadening are apparent to us from the plot in Fig. 2d. Also, it would be helpful to display the Fermi level.

2.2. Comparing Fig. 2e with Fig. S9 it seems like points for HPy+ are missing in the main article which are however present in the figure in the SI. Specifically the points at $(x,y)=(0.28/0.72)$ where the omission support the authors' claim. If one adds the missing points also in the main manuscript, the statement of a "flat electrostatic potential" at short distances for HPy+ is no longer obvious. Moreover, the range of Coulomb potentials in panel e) would not only be more or less the same for both counterions, but HPy+ then shows values for even lower Coulomb potentials. In connection with this, the authors should also explain how the above statement fits to the fact that the Coulomb potentials in Fig. S5 do not differ significantly for both dopants.

2.3. The interpretation of the simulation data in the framework of the concept presented in Fig. 2f and 5g thus seems to us misleading. First, as discussed above, the spread in the depth of the Coulomb potentials seems to be equal, if not reverse, to the ratios presented by the authors in Fig. 2f). On the other hand, the variation in the vertical counterion-dopant distance r is here confounded with an (ir)regular placement of counterions along the polymer.

2.4. the x-axes in panel c) and e) are not consistent, i.e. the numerical values do not match.

3. Regarding the statement that the quadrupole moment Q of MtBA+ and Hpy+ differs significantly.:

3.1. the authors claim a correlation between Coulomb trap depth and Q . However, there is no such panel in Fig S5 which one might think would support the authors argument

3.2. From the data shown in Fig S9, it is not so clear from our point of view that the Q -values differ strongly between the two counterions. For MtBA+, for example, it is not possible to say for sure from the data shown whether $Q=10$ Debye or 25 (or 35) because the lines hardly differ in the range in which the data are measured.

3.3. The authors should explain why Fig. S5 and Fig. S9 are not consistent. If one extracts from Fig. S5 for MtBA+ the Q value (~ 8 Debye), the $1/r$ value (0.3 \AA^{-1} and the value for V_c (-0.62 eV) then the Q value fits the fitted one in Fig. S9a, but the $(1/r, V_c)$ point does not match the fit.

3.4. On page 9, line 211 the authors claim: "These results suggest that aromatic cations are more favorable as the counterions for the doped polymers, which is consistent with the counterion docking screening results."

However, Fig. S5 shows that BPy+, for example, has a similar Q value to MtBA+, but the conductivity differs significantly between the two. Overall, Fig. S5 suggests that the differences in Q and V_c values between individual classes of counterions are smaller than the scatter

within classes. However, this is not true for the conductivities (Fig. 3b). For us, the above-mentioned correlation between quadrupole moment, depth of Coulomb traps and conductivity remains difficult to understand also for this reason.

In our view, there are also some ambiguities regarding the presentation and evaluation of the experiments:

4. The authors' statement on page 11, line 240: "The polymer films exchanged with HPy+ consistently exhibited higher conductivities while maintaining comparable Seebeck coefficients compared to the other two cations." is ambitious, given that in the case of both conductivity and Seebeck coefficient, Hpy+ and BMIM are predominantly the same when the stated errors are considered (see Fig. 4a and b). Consequently, it would be appropriate to indicate error bars also for the power factor (Fig. 4c).

5. The discussion and determination of charge carrier densities on page 11 and in the SI raises more questions than it answers:

5.1. Page 11, line 252/253: "Ultraviolet photoelectron spectroscopy (UPS) analysis showed that different counterions showed similarly high doping levels" this result is inconsistent with a significant (one order of magnitude) change in carrier densities as stated in the manuscript. Instead an independent assessment of the ion densities would be helpful.

5.2. If the charge carrier density is changed by one order of magnitude, we would expect that also the Fermi level is changed between different counterions. However, the UPS measurements in Fig. S12 show no such shift in Fermi level.

5.3. The densities determined by different techniques vary a lot. How large is the error of the different techniques and is it then still justified to claim differences in the charge carrier density? Apart from that, the value of Hall measurements for hopping systems is rather controversial.

5.4. It is not transparent to us which of the 6 peaks present were used in the analysis of the XPS spectra to determine the charge carrier concentration. In addition, we find it somewhat surprising that in the undoped polymer the fit does not agree with the measurement data at exactly the point where the blue peak assigned to a polaron is located in the doped polymers. With said peak, the fit in Fig. S14a would presumably also fit, but this is passed over here without comment.

5.5. Apart from all these methodological difficulties, the numbers do not fit either. In Figure 3b and 4a the maximum differences in conductivity are a factor of 3-6, but this is much less than the factor of 10 in density times the factor of 2 in conductivity that the authors claim here.

6. The authors claim on page 13 line 256 that "conductive AFM allowed a direct comparison of the conductivity differences of these polymer films (Fig. S18). While the films exchanged with MtBA+ displayed islands of conductive regions with limited connectivity, the films exchanged with HPy+ and BMIM+ exhibited clear one-dimensional transmission channels." However, the AFM images shown in Fig. S17 all look the same, depending on which cutout you look at. Looking at the images shown, it is clear that the tip destroys the sample surface and the images therefore have very little informative value. Apart from that, one-dimensional transport would not be advantageous, as it is very sensitive to defects.

7. In the discussion of the absorption spectra in Fig. 5, the "peak" at 950 nm is assigned to a "compressed polaron". However, there is no peak at 950 nm, but a discontinuity, which is

usually caused by switching to another detector or grating. Moreover, what is called a peak here is already visible in the MtBA+ spectra in the first measurement, in fact this looks very similar to the HPy+ measurement after 10 min. How can this be reconciled with the concept of compressed polarons that the authors assume here? Apart from our doubts in the interpretation of the measurements, the argument of the authors concerning the formation of compressed polarons collides with the claim that the charge transport should be more delocalized (p.14, l. 325).

8. Apart from the scientific aspects, it is difficult to follow the line of argument in the manuscript and the approach without continuously looking up the SI.

Reviewer #4 (Remarks to the Author):

Point-to-Point Response

Reviewer 1:

I grew increasingly enthusiastic about the manuscript by Xiong et.al. which demonstrates a simulation driven strategy to improve the mobility of polymeric semiconductors by counterion doping. Through a series of computational steps, the authors are able to rationalize the positioning of the counterions with respect to the polymer backbone, which is non-trivial in a system of such complexity. Based on the computational results they are able to characterize the interaction strength and resulting a different localization of the counterions.

Our response: We really appreciate your positive and constructive comments. We carefully revised the manuscript according to your comments and would like to address these comments as follows.

Q1: They could then show experimentally that the HPy doped system significantly outperformed the MtBA based systems. They could also show, which is highly laudable, that there were no detectable changes in the morphology that are responsible for this change. They could also demonstrate that the nature of the polaron population is different for the 2 cations. Based on these findings, which I find all supported but the evidence provided, the authors conclude that “NCIs between counterions and polymers are critical” and at the moment I am not convinced that this is really the case. Reference 27 in the manuscript discusses at length the relevance of the quadrupole moment suppressing in some mysterious way the Coulomb-screening, a fact that the authors also clearly demonstrate in Fig 2e. While the relevance of the quadrupole moment is discussed in the docking procedure, its magnitude is not discussed with respect to its influence on the conductivity. Indeed, the quadrupole moment of HPy is 4 times higher than that of MtBA, which would fully rationalize the observed difference in the conductivity without any need to consider the interaction strength. Therefore, I am not convinced that on the data presented we can really conclude that the NCI are more important than the quadrupole moment. In light of this question the authors should correlate the conductivity in Fig3b with the quadrupole moment of the dopants and if they see a strong correlation there, they might have to reconsider their conclusion. Alternatively they should

highlight a case of two dopants which have similar quadrupole moment, very different conductivity and very different NCI strength.

Our response: Thank you for your valuable insights. Non-covalent interactions (NCIs) can be classified into different categories, such as electrostatic (ELS) interactions, van der Waals (vdW) forces, and hydrogen bonds (*Chem. Rev.* **2000**, *100*, 143–168). Both dipole (D) and quadrupole (Q) interactions contribute to ELS interactions, which are subtypes of NCIs. Although previous studies suggested that multipole interactions play pivotal roles in energy levels, molecular assemblies, and molecular combinations (*Nat. Mater.* **2015**, *14*, 434–439; *Nat. Mater.* **2008**, *7*, 326–332; *Nat. Commun.* **2023**, *14*, 1356), we do not think that the quadrupole interaction is the only key factors that determine the conductivities. Our investigation revealed that dipole interactions surpass the quadrupole term by an order of magnitude. Additionally, even the collective influence of dipole and quadrupole terms remains considerably smaller compared to the dominance of charge-charge interactions. More importantly, no direct correlation between dipole or quadrupole moments and conductivity was observed. Therefore, we revised our manuscript by considering ELS interactions as a unified entity and removed some quadrupole discussions for better clarity. Through energy decomposition analysis (EDA), we demonstrated that the composition of NCIs determined the distribution of counterions within the polymer. Enhanced ELS interactions stabilized counterions around the polymer backbone, while stronger vdW interactions drive counterions to localize within the alkyl sidechain regions. This counterion distribution significantly impacts doping efficiency, the overlap of Coulombic potential wells, and energy disorder, ultimately influencing conductivity. Thus, we believe that NCIs, including both ELS and vdW, should be considered in this study. Detailed analysis and discussions are provided below:

(1) No direct correlation was observed between dipole/quadrupole moments and conductivity.

For the ELS interactions between the polymer and counterions, the dominant factor is the charge-charge interaction. In Fig. 2e in the manuscript, we calculated the electrostatic binding energy V_C using a recently reported model (*Nat. Commun.* **2023**, *14*, 1356). This model describes the ELS interactions of the dopant counterion by considering the monopole and

quadrupole moment (\mathbf{Q}). The formula to compute these interactions estimation reads:

$$V_C = \sum_i \left(\frac{-eq_i^p}{4\pi\epsilon_0\epsilon_r r_{ij}} + \frac{q_i^p}{8\pi\epsilon_0\epsilon_r} \frac{\mathbf{r}_i \mathbf{Q} \mathbf{r}_i}{r_i^5} \right)$$

where the dipole moments of all their studied dopant molecules are negligible. However, the counterions in our system can have substantial dipole moments, up to 14.9 Debye, and vary for different counterions, which cannot be disregarded (Fig. R1). Additionally, the initial formula simplifies the dopant counterion as a point charge. However, we observed that the electron density derived from the wavefunction exhibits a strong correlation with the molecular conformation, resulting in the quadrupole moment fluctuating with changes in ion conformations (Fig. R2). Considering the significant differences in charge distribution between alkyl counterion and aromatic counterions, treating them as point charges at the geometric center is inadequate. Therefore, we reevaluated the V_C , considering the partial charge and dipole moments of counterions. The revised formula we used is:

$$V_C = \sum_i \left(\sum_j \frac{q_j^c q_i^p}{4\pi\epsilon_0\epsilon_r r_{ij}} + \frac{q_i^p}{4\pi\epsilon_0\epsilon_r} \frac{\mathbf{D} \mathbf{r}_i}{r_i^3} + \frac{q_i^p}{8\pi\epsilon_0\epsilon_r} \frac{\mathbf{r}_i \mathbf{Q} \mathbf{r}_i}{r_i^5} \right)$$

Here, $\epsilon = \epsilon_0\epsilon_r$ is dielectric permittivity. i and j indexes the atoms within the polymer and counterions, respectively. q_i^p and q_j^c denote the partial atomic charges of the polymer and the counterions. The distance between atom i in polymer and atom j in counterion is represented by r_{ij} . \mathbf{r}_i is a vector connecting polymer partial charges and counterion center-of-geometries; \mathbf{D} and \mathbf{Q} represent the counterion dipole moment and quadrupole moment, respectively.

Fig. R1. Magnitude of **a**, the dipole moments and **b**, the quadrupole moments calculated via DFT-optimized conformation for 28 cations.

Fig. R2. Electrostatic potential (ESP) mapped molecular vdW surfaces for **a**, HPy⁺ and **b**, MtBA⁺. The color scale represents ESP values ranging from 0.18 and 0.04 a.u. The corresponding conformation for each counterion is labeled with its quadrupole moment magnitude.

In the multipole expansion of V_C , the monopole term (charge-charge interaction) is the most significant contributor (Fig. R3). When the distance between the polymer and counterion (r) decreases to below 5.0 Å ($1/r > 0.2 \text{ Å}^{-1}$), the dipole and quadrupole interactions gradually increase (Fig. R3c-d). However, the impact of these short-range interactions remains quite modest. For HPy⁺, the average dipole and quadrupole interactions are 0.038 eV and 0.012 eV, respectively. These values are considerably smaller than the average charge-charge interaction of -0.64 eV Å (Fig. R3b). MtBA⁺ has smaller dipole and quadrupole moments (Fig. R1-2), resulting in average dipole and quadrupole interaction of around 0.016 eV and 0.0039 eV. It is important to note that the dipole interactions are an order of magnitude larger than the quadrupole term for both counterions. As a result, these short-range multipole interactions result in a slight up-shift in the electrostatic potential of 0.050 eV for HPy⁺ and 0.016 eV for MtBA⁺ (Fig. R3a). Nonetheless, these shifts are tiny when compared to the charge-charge interaction, and they barely change the electrostatic potential for both cations. Indeed, the dipole and quadrupole components highlighted the importance of charge distribution on counterions (*Nat. Mater.* **2019**, *18*, 242-248).

More importantly, we found no direct correlation between dipole or quadrupole moments and conductivity. For example, despite nearly identical dipole moments of 5.14 Debye for BtMA⁺ and 5.16 Debye for BMIM⁺, their conductivities differ over fivefold (Fig. R4). Similarly, with BMIM⁺ and BPy⁺, their quadrupole moments are 5.67 and 4.60 Debye Å, respectively, yet BPy⁺ exhibit a 34.4 S/cm higher conductivity than BMIM⁺. Additionally, when comparing the same type counterions, like BPy⁺ and OPy⁺, despite remarkable differences in dipole (10.3 Debye) and quadrupole (37.4 Debye Å) moments, their conductivities remain nearly the same. These results suggest that the absence of a direct correlation between dipole or quadrupole moments and conductivity.

Fig. R3. **a**, The distance-dependent V_C for P(PzDPP-2FT) with MtBA^+ and HPy^+ . **b**, the monopole term (charge-charge interaction), **c**, the dipole term (dipole-charge interaction), and **d**, the quadrupole term (quadrupole-charge interaction) in the multipole expansion of V_C . The polymer-counterion distance (r) was measured between the geometry center of a counterion and the geometric center of the adjacent polymer backbone with two repeating units.

Fig. R4. The relationship between **a**, dipole and **b**, quadrupole moments with electrical conductivity.

(2) Both ELS and vdW contribute to counterion distribution.

Solely considering ELS energy to understand the counterion effect is inadequate in such a complex system. Employing energy decomposition analysis (EDA) based on the force field (Fig. R5), we identified two major ingredients of NCIs: ELS and vdW interactions. The ELS potential is computed from the partial atomic charges (*Science*, **1995**, 268,1144-1149) and the vdW interactions include repulsive interaction due to Pauli repulsion effect (E_{rep}) and attractive dispersion interaction (E_{disp}) (*J. Am. Chem. Soc.*, **1992**, 114, 7827-7843). The introduction of counterions notably enhances NCIs, suggesting unignorable NCIs between the polymer and counterions (Fig. R5a). Specifically, the ELS interactions in polymer-HPy⁺ and OPy⁺ notably exceed those in polymer-MtBA⁺. In contrast, polymer-MtBA⁺ exhibits more substantial vdW interactions than polymer with HPy⁺ and OPy⁺. As shown in Fig. R5b, within polymer-MtBA⁺, ELS interactions account for 64.8% of the total NCIs, with vdW interactions contributing the remaining 35.2%. However, in the polymer-HPy⁺ or OPy⁺ systems, the composition of NCIs leans more towards ELS interactions, which constitute approximately 80% with vdW interactions contributing about 20%. To better understand these differences, we evaluated the interaction energies between the counterions and separate parts of the polymer (backbone and sidechain). The ELS interactions significantly influence the attraction between the anionic polymer backbone and cationic counterions (Fig. R5c), and contribute to the repulsion between the polymer sidechain and counterions (Fig. R5d). Additionally, vdW interactions are key in facilitating attractive interactions between the polymer sidechain and counterions. Particularly for MtBA⁺, the attractive vdW interactions outweigh the repulsive ELS interactions with the polymer sidechain, leading to more pronounced attractive interactions with the polymer sidechain compared to the other two aromatic cations.

Fig. R5. a, Energy decomposition analysis (EDA) based on the force field for the interaction between P(PzDPP-2FT) and counterions. **b**, Quantitative analysis depicting the contribution of electrostatic or van der Waals interaction to the overall NCIs between P(PzDPP-2FT) and counterions. **c**, EDA for the interactions between the polymer backbone and counterions. **d**, EDA for the interactions between the polymer sidechain and counterions.

To better understand ELS and vdW interactions, we also performed EDA based on DFT-calculated wavefunctions using a recently reported sobEDAw method (*J. Phys. Chem. A* **2023**, *127*, 7023–7035). sobEDAw decomposes intermolecular interaction energy as follows:

$$\Delta E_{int} = \Delta E_{els} + \Delta E_{xrep} + \Delta E_{orb} + \Delta E_{disp}$$

where ΔE_{orb} and ΔE_{xrep} represent orbital and exchange-repulsion terms, respectively; ΔE_{els} and ΔE_{disp} denote ELS and dispersion interactions, respectively. The interaction energies between counterions and different parts of the polymer (backbone and sidechain) were summarized in Table R1. HPy⁺ and OPy⁺ consistently exhibit stronger ΔE_{els} with the polymer backbone (-56.0 kcal/mol and -59.3 kcal/mol, respectively) compared to MtBA⁺ (-52.8 kcal/mol). Conversely, MtBA⁺ displays a stronger dispersion interaction with the sidechain (-17.0 kcal/mol) compared to HPy⁺ and OPy⁺ (-8.29 kcal/mol and -11.8 kcal/mol, respectively). EDA analyses based on

both force field and wavefunctions revealed that significant electrostatic interactions effectively stabilize counterions around the polymer backbone (as observed with HPy⁺ and OPy⁺). Although vdW interactions contribute less to overall NCIs compared to electrostatic energy, their accumulation across multiple polymer sidechains remains crucial for counterion diffusion and stabilization.

Table R1. Components of the interaction energies (kcal/mol) derived by sobEDAw method.

Polymer	Counterion	ΔE_{int}	ΔE_{els}	ΔE_{orb}	ΔE_{rexc}	ΔE_{disp}
Backbone	MtBA ⁺	-57.4	-52.8	-11.6	25.7	-8.71
	HPy ⁺	-68.2	-56.0	-13.0	15.3	-14.4
	OPy ⁺	-65.3	-59.3	-14.2	22.9	-14.6
Sidechain	MtBA ⁺	-13.1	-3.15	-5.91	13.0	-17.0
	HPy ⁺	-7.00	-1.28	-2.67	5.25	-8.29
	OPy ⁺	-8.65	-2.14	-4.16	9.48	-11.8

Based on the discussions above, we have revised the section concerning the NCIs and the quadrupole moment in both the manuscript and the Supplementary Information. These revisions are outlined as follows:

1. In the manuscript, we have removed the discussion regarding the relationship between the quadrupole moments:

107 To gain deeper insights into the NCIs of the most stable polymer-counterion pair, we
108 employed density functional theory (DFT) calculations (Step 4). The DFT-calculated
109 interaction energy revealed that counterions possessing long alkyl chains or aromatic
110 structures had more atoms available for binding and thus exhibited stronger affinity with
111 the polymer chains, such as MtBA⁺ and OPy⁺ (inset in Step 4). In addition, aromatic rings
112 with a large quadrupole moment (Q) contributed more to the binding energy compared to
113 alkyl cations with a similar number of binding atoms (e.g., HPy⁺ vs. BtMA⁺)²⁵. Among the
114 NCIs of polymer-counterion pairs, the long-range electrostatic-Coulomb interactions had
115 the most profound impact on doped polymers, as they could act as Coulomb traps in the
116 density of states^{10,26}. To quantify this effect of different counterions, we calculated the
117 Coulomb interaction energy (V_C) using a recently reported model that considered the
118 distance, orientation, and electrostatics of the polymer-counterion pairs²⁷. Intriguingly, our
119 investigation revealed that aromatic counterions with a large Q exhibited shorter distances
120 to the polymer backbone while maintaining comparable V_C values to alkyl cations (Fig.
121 S5). This finding seems counter-intuitive, as the Coulomb energy typically increases as the
122 distance decreases. Nevertheless, we will show that the quadrupole moment is also
123 important, which affects the distribution of counterions and the strength of NCIs in
124 polymer-counterion pairs.[⚡]

2. In the revised manuscript, we have added the discussion about NCI composition (highlighted by yellow color) and the Fig. R5a is added as Fig. 2b:

116 dock the 28 cations into the final refined docking pocket, generating 100 docking poses for
117 each counterion (Step 3). Fig. 1c shows the various NCIs formed between the polymer
118 backbone and the counterion. Analyzing the interaction energy between the polymer and
119 counterions (Fig. 1d, Step 4), we observed that counterions possessing longer alkyl chains
120 had more available atoms for binding and thus exhibited stronger affinity with the polymer
121 chains, such as MtBA⁺ (longer chains) vs. BtMA⁺ (shorter chains). Interestingly, cations
122 with aromatic structures exhibited stronger interactions compared to alkyl ions with similar
123 sidechain lengths (e.g., BPy⁺ vs. BtMA⁺), suggesting different types of NCIs for aromatic
124 rings^{25,26}.[⚡]

135 Using energy decomposition analysis (EDA) based on the force field^{27,28}, we identified
 136 NCI compositions as the key factor behind the different counterion docking behaviors
 137 between MIBA^+ and HPy^+ . As shown in Fig. 2b, MIBA^+ exhibited almost twice higher van
 138 der Waals (vdW) interactions with the polymer than HPy^+ . In contrast, HPy^+ showed
 139 significantly stronger electrostatic (ELS) interactions, accounting for 80% of its NCIs,
 140 compared to 65% for MIBA^+ (Fig. S8b). The accumulation of the vdW interactions across
 141 multiple neutral polymer sidechains drew MIBA^+ into the sidechain regions, while the
 142 more substantial ELS interactions made HPy^+ dock closer to the charged polymer
 143 backbone (Fig. S8c-d). To validate this conclusion, we conducted NCI analysis on the MD
 144 simulation results of another aromatic cation, OPy^+ . We observed that OPy^+ exhibited the
 145 same docking behavior as HPy^+ , with a preference for docking close to the polymer
 146 backbone (Fig. S6) and an approximately 80% composition of ELS interactions (Fig. S8).
 147 These results further confirm that the counterion docking behaviors are determined by the
 148 NCI composition. Since ion-exchange doping uses ionic interactions as a driving force^{15,29},
 149 the stronger NCIs between the polymer backbone and the counterion may contribute to a
 150 higher doping level.⁴

151

152 **Fig. 2| Understanding of the counterion effects on the energetic disorder of the doped**
 153 **polymers. a, MD simulation snapshot of the polymer P(PzDPP-2FT) with MIBA^+ (top) or**

3. In the Supplementary Information, we have expanded our discussion to include NCI compositions and Coulomb interactions to the Section 3.3 and Section 3.5, respectively: Specifically, we have incorporated Fig. R5 as Fig. S8 and Table R1 as Table S2 in Section 3.3. Additionally, Fig. R1-3 have been included as Fig. S13-15.

Reviewer 2:

The authors have systematically presented experimental and theoretical studies on dopants in n-type conjugated polymers, and have successfully revealed its counterion effects. Although the trend in doping efficiency depending on the ionic species has been studied qualitatively in previous literature, the authors has constituted a key-step forward in quantitative understandings particularly about the energy landscape in doped polymers. The theoretical studies presented in Figs. 1 and 2 have given unambiguous insight about how different dopants having different conformation and size affect on the energetic disorders in conjugated polymers. In addition, the most of experimental studies presented in the manuscript have supported the theoretical prediction. The reviewer highly appreciates this study, and in principle agrees with the publication. However, the main claim “counterion docking” mechanism has to be clarified more.

Our response: We are glad to read that the Reviewer “highly appreciates” our study and agrees with the publication. We carefully revised the manuscript and performed more computational and experimental works to clarify the “counterion docking” mechanism and would like to address your concerns as follows.

Q1: The authors stated “Interestingly, some counterions can also “lock” the polymer backbone and reduce the backbone rotational disorder, leading to even smaller backbone energetic disorder than the undoped polymers.” This conclusion was not been experimentally supported. In doped polymers, the reviewer agrees that the majority of electron carriers forms condensed polarons. Equivalently, the anionized pi-conjugated backbone turns into quinoid, where all torsion in segments should be spontaneously suppressed. Did author calculate torsion angle distribution in anionized polymer-dopant pairs in Fig. S8? If no, the calculation in Fig. S8 does not given any supportive evidence of “counterion docking”.

Our response: Thank you for your insightful comments. We agree that both counterion interactions and anionized conjugated polymer contribute to the planarization of backbones, and we have included additional experimental measurements such as GIWAXS and Raman spectroscopy to provide experimental support:

(1) GIWAXS analysis: After ion-exchange, the polymer films exhibit a greater expansion in the lamellar distance compared to the π - π distance (Fig. R6). This finding indicates that counterions mainly diffused into the sidechain region rather than inserting into the π - π stacking region (*Nat. Mater.* **2016**, *15*, 896–902), in consistent with the molecular dynamic simulations. In addition, polymer films exchange-doped with MtBA⁺ exhibited larger lamellar and π - π stacking distances compared to those exchange-doped with BMIM⁺ and HPy⁺, across different doping times (Fig. R6). This result suggests that the counterions diffusion influences polymer backbone torsion and thus molecular packing. Notably, when the polymer was exchanged with HPy⁺ for 5 minutes, the π - π distance slightly decreased (Fig. R6b). However, at a higher doping level, the π - π distance of the polymer exchange-doped with HPy⁺ also increased by 0.02 Å compared to the pristine polymer. These results suggest that the backbone planarity is also influenced by the counterions, not only determined by the anionization.

Fig. R6. Variations in **a**, lamellar packing distances and **b**, π - π stacking distances of the exchange-doped P(PzDPP-2FT) at 5 and 30 minutes compared to the pristine polymer film. Data were obtained from the GIWAXS measurements.

(2) Energetic disorder: The energetic disorder reflects the conformational order of the polymer. In our study, we employed two methods to experimentally assess energetic disorder: activation energy measurement and analysis of the Seebeck-conductivity (S - σ) relationship. Our findings showed that at various doping levels, the polymer exchanged with HPy⁺ consistently displayed approximately half the activation energy (E_a) compared to that of MtBA⁺ (Fig. R7a), indicating a considerably lower barrier height for charge transport in the polymer exchanged with HPy⁺. We used the SLoT model to analyze the charge carrier transport characteristics of the doped polymers (Fig. R7b) (*Nat. Mater.* **2021**, *20*, 1414–1421). The S - σ

function of the polymer exchange-doped with MtBA⁺ exhibits a left-bottom shift compared to that with HPy⁺, indicating an increase in energetic disorder (*Nat. Mater.* **2017**, *16*, 252–257; *Nat. Mater.* **2021**, *20*, 1414–1421). Our fitting analysis also revealed that the polymer exchanged with HPy⁺ exhibited the highest transport coefficient (σ), indicating enhanced mobility compared to other cations. Additionally, we observed that the depth of the potential wells (W_H) in the polymer with HPy⁺ was approaching zero, implying significantly lower barrier height for carrier hopping compared to the polymer with other cations.

Fig. R7. a, Extracted activation energies of the polymer films exchanged with MtBA⁺ and HPy⁺ at different doping levels. **b**, Fitting the S- σ relationship using the SLoT model for the polymer films exchanged with different cations.

(3) Raman spectroscopy: The anionized pi-conjugated backbone turns into quinoid, which also play a role in suppressing the backbone torsion, as demonstrated by a previous study (*Nat. Commun.* **2022**, *13*, 5970). Here, we utilized Raman spectroscopy to probe the molecular conformation of conjugated polymers, which is a useful tool in characterizing the effects of counterions on polymers (*J. Mater. Chem. C* **2013**, *1*, 5638-5646; *J. Am. Chem. Soc.* **2020**, *142*, 15340–15348). For P(PzDPP-2FT) films, we monitored five major peaks related to carbon bond stretching (Fig. R8a). The peak at 1356 cm⁻¹, associated with the symmetric C-C stretch mode, remained unchanged between the pristine and the doped P(PzDPP-2FT) films. However, all peaks related to the C=C stretch mode (1474, 1504, 1520 and 1535 cm⁻¹) in the polymer with HPy⁺ and MtBA⁺ notably shifted to lower wavenumbers. Additionally, after ion-exchange doping, we observed a decrease in the intensity of C=C stretching mode (1474, 1504, 1520 cm⁻¹) in the pyrazine, thiophene, and DPP building blocks, while the intensity of the C=C stretching mode (1535 cm⁻¹) across the entire conjugated backbone increased (Fig. R8b),

particularly in the polymer exchanged with HPy^+ . These results imply a transition towards a more quinoid structure, which may enhance the segmental order of the ion-exchanged polymers (*Macromolecules* **2017**, *50*, 4227–4234; *J. Phys. D: Appl. Phys.* **2017**, *50*, 073001).

Fig. R8. a, Raman spectra of the pristine P(PzDPP-2FT) film, P(PzDPP-2FT) doped with TDAE, and P(PzDPP-2FT) exchange-doped with MtBA⁺ and HPy⁺. All the Raman spectra were excited at 633nm and the Raman intensity was normalized to the peak at 1356 cm^{-1} . **b,** Intensity ratio of different peaks to the peak at 1356 cm^{-1} . **c,** Diagram illustrating the dominant bond stretches associated with the main Raman active vibrational modes of P(PzDPP-2FT) centered around 1474, 1504, 1520, and 1535 cm^{-1} .

(4) DFT calculations and MD simulations support: In addition to the supplementary experimental data, we have included DFT calculations and analysis to further elucidate the “counterion lock” effect. As shown in Fig. R9, we compared the backbone planarity of the P(PzDPP-2FT) dimer in its neutral and doped states via DFT calculations. In the charged state, the length of the C-C linkage bond between repeating units shortens from 1.45 Å to 1.43 Å, indicating a more quinoid structure. Furthermore, the torsion barrier height between repeating units increases from 8.68 to 13.3 kcal/mol.

Fig. R9. a, DFT-optimized structure of the P(PzDPP-2FT) dimer in neutral and doped states. b, Relaxed potential energy scan of the torsion angle between the repeating units of P(PzDPP-2FT) dimer in neutral and doped states.

Besides, we noticed that interactions between the polymer backbone and the dopant counterion contribute to further planarization of the doped conjugated backbone. It has been well-studied that intramolecular noncovalent interaction can “lock” the backbone conformation (*J. Am. Chem. Soc.* **2013**, *135*, 10475–10483; *Chem. Rev.* **2017**, *117*, 10291–10318). Notably, NCIs also exist between the polymer and the dopant counterion (Fig. R10). We observed that, despite a slight increase in the C-C linkage bond length of (PzDPP) $_2^{1-}$ -HPy $^+$ (or MtBA $^+$) compared to (PzDPP) $_2^{1-}$, the torsion barrier height significantly increased to around 19 kcal/mol in the presence of counterions. This result indicates that intermolecular NCIs also contribute to enhancing the backbone planarity, which we refer to as “counterion locking”.

Fig. R10. Visualization of the intermolecular interaction between the polymer backbone and counterions using the independent gradient model (IGM) analysis (*Phys. Chem. Chem. Phys.* **2017**, *19*, 17928).

To assess the “counterion locking” effect, we use the same force field for pristine polymer and polymer with counterions in molecular dynamic simulations. As shown in Fig. R11, the polymer with MtBA $^+$ counterions exhibited obviously broadened torsion angle distributions,

especially for the DPP-Pz torsion angle directly linked to alkyl sidechains. However, the polymer with HPy^+ exhibits an even narrower torsion angle distribution compared to the pristine polymer. These results indicate that when the counterions locate around the polymer backbones, they can effectively “lock” the polymer backbone rotations through NCIs.

Fig. R11. **a** Chemical structure of P(PzDPP-2FT). The torsion-angle DPP-Pz and Pz-2FT were indicated by blue and red arrows, respectively. Torsion-angle distributions of **b** DPP-Pz and **c**, Pz-2FT segments in the pristine P(PzDPP-2FT) and the polymer with MtBA⁺ or HPy⁺.

The MD simulation results are also consistent with the GIWAXS analysis. As shown in Fig. R12a, the diffusion of MtBA⁺ into the sidechain regions led to increased sidechain disorder (larger lamellar expansion) and a greater twisting of the polymer backbone (larger π - π distance). Conversely, the polymer backbone surrounded by HPy⁺ maintained its zig-zag conformation and the alkyl sidechain are interdigitated more orderly (Fig. R12b). As a result, both the lamellar and π - π distances in polymer-HPy⁺ end up being smaller than those in polymer-MtBA⁺ (Fig. R12c-d).

Fig. R12. MD simulation snapshot of the polymer P(PzDPP-2FT) with **a**, MtBA⁺ or **b**, HPy⁺ as the counterions. The disorder sidechain regions in P(PzDPP-2FT)-MtBA⁺ are highlighted with red circles. Static analysis of **c**, lamellar distances and **d**, π - π stacking distances for the supercell of the polymer with HPy⁺ and MtBA⁺.

To clarify the “counterion locking” effect, we have made the following revisions in the manuscript and Supplementary Information:

1. **In the manuscript, we have included the GIWAXS analysis (Fig. R6a-b here) of variations in cell parameters when exchanged with different counterions as Fig. 4b-c. Additionally, we have added the following discussion to the manuscript (highlighted by yellow color):**

265 of the films exchanged with different cations (Fig. S23). These results indicate that the
266 counterion exchange did not disrupt the microstructure of the polymer films. Nonetheless,
267 we did observe differences in the GIWAXS spacing parameters. Polymer films exchange-
268 doped with HPy⁺ and BMIM⁺ consistently exhibited smaller lamellar and π - π stacking
269 distances compared to those exchange-doped with MtBA⁺, across different doping times
270 (Fig. 4b and 4c). This trend was also observed in MD simulations (Fig. S12). The diffusion
271 of MtBA⁺ into the sidechain regions led to increased sidechain disorder (lamellar
272 expansion) and a greater twisting of the polymer backbone (larger π - π distance).
273 Conversely, HPy⁺ docking near the polymer backbone contributed to smaller lamellar
274 expansion, and the stronger interactions between the charged polymer backbone and HPy⁺
275 contributed to the suppression of backbone torsion and smaller π - π stacking distance.
276 Better backbone planarity and smaller π - π distance could benefit intrachain and interchain
277 charge transport⁵². Conductive AFM (c-AFM) measurements revealed that the films
278 exchanged with MtBA⁺ displayed islands of conductive regions with limited connectivity.

2. **In the Supplementary Information, we have included a detailed discussion about the role of counterion interactions and anionization in reducing backbone torsion to Section 3.4 and Section 4, respectively. In addition, we have added Figures R9 to R12 as Figures S9 to S12 in Section 3.4, and Fig. R8 has been incorporated as Fig. S25 in Section 4.**

Q2: The authors have argued the potential crossover in electronic states from single polarons, to bipolarons, then to compressed polarons in Fig. 5. Judging from the Figs. 5c and 5g, it seems that the majority of electron carrier forms bipolarons. However, no characteristic absorption was found in Fig.5a. As has been explained in the ref. 47, it is difficult to distinguish compressed polarons and bipolarons by the electron spin resonance. The authors should clarify this more in details. In addition, is “EPR intensity” a mere peak-to-peak intensity or integrated intensity? Normally, the integrated intensity has to be plotted when spin susceptibility is argued. In the reviewer’s opinion, details of electronic states, i.e., polarons, bipolarons, or condensed polarons, are beyond of this study. The author needs extra cares to identify the electronic states, for example temperature dependence measurements for EPR.

Our response: Thank you for your advice. The EPR intensity used in this work is integrated intensity. We agree that the details of electronic states go beyond the scope of this study and won’t affect our conclusion. Therefore, we removed this part of the discussion from our manuscript. Nevertheless, the electronic states of carriers in heavily doped systems have received limited attention, and we find this topic intriguing. We have included some of our results below and added these discussions in the Section 4 of Supplementary Information for reference:

Fig. R13 summarizes the different possibilities of spin pairing in a polymer chain from the neutral state to different doping levels. Upon doping, an additional electron is introduced into one of the polymer chains, leading to the formation of unpaired electrons, known as polarons, which contribute to the EPR signal. Further increasing the doping level, two polarons within the same chain or neighboring chains can couple together to form a bipolaron (*Macromolecules* **2017**, *50*, 4867-4886). Bipolarons are EPR silent and result in a decrease in the spin density (N_s), despite an increase in carrier concentration (N_D) (*Acc. Chem. Res.* **1996**, *29*, 417–423). At a high doping level, the combination of a bipolaron and an unpaired polaron would give rise to the EPR signal and result in an increase in N_s again (*J. Phys. Chem.* **1996**, *100*, 15644-15653). However, it is possible that polarons in the same chain may be compressed to shorter lengths to form an array or polaron lattice (*Phys. Rev. Lett.* **1985**, *55*, 308–311; *J. Phys. Chem. C* **2014**, *118*, 114–125).

Fig. R13. Schematic diagrams illustrating the different possibilities of the spin pairing in polymer chains from neutral to different doping states.

To better distinguish between bipolarons and compressed polarons, we performed temperature-dependent EPR measurements (Fig. R14). A dramatic decrease in EPR intensity with increasing temperature was observed for polymer exchanged with both HPy⁺ and MtBA⁺, suggesting a triplet ground state (*Chem. Rev.* **2013**, *113*, 7011–7088). We used the Bleaney-Bowers equation to fit their EPR intensity, which provides the energy gap between the singlet and triplet states (ΔE_{S-T}) of 0.0157 kcal mol⁻¹ for the polymer with HPy⁺ and 0.0223 kcal mol⁻¹ for the polymer with MtBA⁺. These results suggest that at low temperatures, compressed polarons (existing in triplet state) are more stable than bipolarons (existing in singlet state). As the temperature increases, unpaired compressed polarons gradually combine to form bipolarons. Although distinguishing between compressed polarons and bipolarons becomes more challenging at room temperature, the presence of triplet compressed polarons at low temperatures is evident from these results.

Fig. R14. Temperature-dependent EPR of P(PzDPP-2FT) exchanged with **a**, HPy⁺ and **b**, MtBA⁺ and **c**, the EPR integrated intensity and corresponding Bleaney-Bowers equation fitting result. The temperature is in the range of 10 K to 100 K.

Based on the discussions above, we have made revisions to the section concerning the polarons in both the manuscript and the Supplementary Information. These revisions are outlined as follows:

- 1. In manuscript, we have moved the discussion about the electronic states of polarons to Section 4 in Supplementary Information. Correspondingly, the figures that were originally Fig. 5a-c have been relocated as Fig. S28a-c. Additionally, Fig. 5g has been deleted:**

292 significantly affect the counterion-induced energetic disorder. To further understand this
293 counterion effects, we employed ultraviolet-visible near-infrared (UV-vis-NIR) absorption
294 spectra for analysis. Upon doping, two new absorption bands at 2000-2500 nm and 1300-
295 nm were observed (Fig. 5a-b), which can be attributed to polarons or bipolaron bands,
296 respectively, defined by time-dependent DFT (TD-DFT) calculations (Fig. S19). As the
297 doping level increased, the film exchanged with HPy⁺ showed a gradual decrease in the
298 polaron band, along with the emergence of a new peak at 950 nm (Fig. 5b). This blue-
299 shifted band is attributed to compressed polarons (Fig. S19), which is similar to polarons
300 but with shorter lengths^{46,47}. In contrast, the spectra of the polymer exchanged with MtBA⁺
301 showed no significant change (Fig. 5a), suggesting that different counterions affect the
302 polaron types and their ratios. Electron paramagnetic resonance (EPR) measurements
303 further proved this conclusion (Fig. 5c and Fig. S20). At low doping levels, polarons
304 dominated. After increasing doping levels, two adjacent polarons combined to form a
305 spinless bipolaron, leading to a decrease in the EPR signal. At high doping levels, polarons
306 within the same chain were strongly localized by the counterions around the polymer
307 backbone, forming compressed polarons, resulting in an increase in the EPR signal.
308 Notably, the final stable EPR intensity indicated that more compressed polarons were
309 formed in the polymer exchanged with HPy⁺ compared to MtBA⁺, which could be
310 attributed to its closer and more uniform distribution around the polymer backbone. This
311 leads to stronger counterion polaron interactions, preventing their combination into
312 bipolarons (Fig. S21).

327
328

Fig. 5. UV-vis-NIR spectra of the polymer films exchange-doped with **a**, MtBA⁺ and **b**,

2. In the Supplementary Information, we have included a detailed discussion about the electronic states of polarons and added Fig. R13-R14 as Fig. S26-S27.

Reviewer 3:

Xiong et. all provide a detailed study of the influence of non-covalent interactions between different counterions and polymers and their influence on thermoelectric parameters. Both the topic, the approach and to some extent the results are interesting and the authors make use of a variety of methods both in experimental and modeling. Despite all this, we do not consider the manuscript suitable for publication in Nature Communications. The reason is that it has both conceptual and methodological weaknesses that we do not believe are easily resolved, especially since there are mutually contradictory arguments throughout the manuscript. Our more detailed rationale for this statement can be found below.

Our response: We are glad that you consider this work is interesting. We carefully read all your comments and really appreciate the time and effort you have taken to review our manuscript. It seems that our original manuscript did not adequately articulate the “counterion docking” concept well and provide enough experimental results to support our findings. We recognize the importance of addressing these concerns comprehensively, and we wish to present our response to the comments with enhanced clarity and conviction as follows.

In response to your comments, we have made the following key revisions:

1. **More experimental support:** We have incorporated many additional experimental results, including GIWAXS, Raman spectra, UPS, and EPR, to provide more robust evidence to support our findings.
2. **Clearer discussion:** We have reorganized the discussion sections about experimental results to build a more coherent connection between the simulation findings and experimental validations. We identified three critical factors that are directly influenced by the counterion docking: the doping efficiency, the polymer conformations, and the Coulomb potential landscape. All these factors were well supported by the simulations and experiments in the revised manuscript.
3. **In-depth analysis of NCIs:** We have delved deeper into the study of NCIs between the polymer and counterions. Our findings indicate a limited influence of the quadrupole moment to electrostatic (ELS) interactions compared to the predominant charge-charge interactions. Consequently, we revised the manuscript by considering ELS interactions as a unified entity, removing some quadrupole discussions for better clarity. Through energy

decomposition analysis, we have successfully identified that ELS as well as van der Waals interactions, subtypes of NCIs, determined the counterion docking.

4. **Clarifications:** To illustrate how counterion docking affect conductivity in such a complex polymer-counterion system, our revised manuscript has included a schematic, highlighting critical factors influenced by counterion docking, and building a bridge between counterion docking, energetic disorder, and ultimately, thermoelectric performance. Furthermore, we have addressed potential misunderstandings regarding the experimental methodologies and conclusions and provided detailed explanations in the following response to address these confusions.

We believe these substantial revisions have addressed all the conceptual and methodological concerns and enhanced the overall clarity of our manuscript. We hope that, after these revisions, you may support the publication of this manuscript in *Nature Communications*. Thank you once again for your valuable feedback and for considering our revised manuscript.

Q1: The authors build much of their argumentation and interpretation of the measured experimental data on the results of their MD and DFT simulations. Unfortunately, the results of the simulations are not verified experimentally, e.g. no experiments are shown which prove that the dopants really dock at the positions predicted by the simulations of the authors. However, since core statements of the authors as well as the entire following interpretation of experiments are almost exclusively based on these results of these simulations, it would be absolutely necessary from our point of view to verify them (at least partially) also experimentally. Instead, the conclusions drawn from the simulation results with regard to conductivity are only very indirect. From our point of view, the data shown cannot prove beyond doubt that there really is a connection between counterion docking and thermoelectric parameters, especially since other options apart from paracrystalline disorder are largely disregarded.

Our response: Thank you for your advice. To address your concern, we have included additional experimental results, such as grazing-incidence wide-angle X-ray scattering

(GIWAXS), Raman spectroscopy, ultraviolet photoelectron spectroscopy (UPS), and electron paramagnetic resonance (EPR) to support our discussions and more clearly align the simulation outcomes with the experimental data. For example, GIWAXS analysis, a widely used technique for investigating dopant counterion locations (*Nat. Mater.* **2016**, *15*, 896–902; *Nat. Mater.* **2019**, *18*, 149–155), supports our finding of the counterion docking position. After ion exchange, the polymer films exhibit a greater expansion in the lamellar distance compared to the π - π distance (Fig. R15). This finding implies that counterions predominantly diffused into the sidechain region rather than inserting into the π - π stacking region (*Nat. Commun.* **2020**, *11*, 3292), corroborating the MD simulations. Specifically, when the polymer exchanged with HPy⁺ for 5 minutes, we observed a slightly reduction in the π - π distance (Fig. R15b), indicating an enhancement in the backbone planarity. In addition, polymer films exchange-doped with HPy⁺ exhibited smaller lamellar and π - π stacking distances compared to those exchange-doped with MtBA⁺, across different doping times (Fig. R15). These results suggest that HPy⁺ docking near the polymer backbone effectively suppressed the backbone torsion, while MtBA⁺ diffused into the sidechain region, resulting in significantly increased sidechain disorder (larger lamellar expansion) and a greater twist of the polymer backbone (larger π - π stacking distances).

Fig. R15. Variations in **a**, lamellar packing distances and **b**, π - π stacking distances of the exchange-doped P(PzDPP-2FT) at 5 and 30 minutes compared to the pristine polymer film. Data were obtained from the GIWAXS measurements.

Regarding your concerns about paracrystalline disorder, we observed no significant changes in the paracrystallinity (g) of P(PzDPP-2FT) when exchange-doped with the three counterions: MtBA⁺, BMIM⁺, and HPy⁺, as detailed in our manuscript. In addition, we also performed GIWAXS measurements on three more cations, totaling six counterions: MtBA⁺ and

BMIM⁺ (alkyl cations), BMIM⁺ and BzMIM⁺ (imidazole cations), and HPy⁺ and OPy⁺ (pyridine cations). As shown in Fig. R16, there was no clear correlation between g and electrical conductivity. For instance, the polymer exchanged with BMIM⁺ and BtMA⁺ exhibited almost identical g values of 17.2, while their conductivities differed by more than 5-fold. These results indicate that although paracrystallinity disorder is an important structural characteristic, it does not predominantly determine the electrical conductivity in our system.

Fig. R16. Conductivity vs. the π - π stacking paracrystallinity (g) of the P(PzDPP-2FT) films exchange-doped with various cations.

We apologize for not initially presenting a clear relationship between counterion docking and thermoelectric performance. To improve clarity and readability, we have added a schematic to illustrate how counterion docking affects charge transport and thermoelectric performance in heavily doped polymer films (Fig. R17). Our simulations and analysis revealed that NCIs, including electrostatic interactions (ELS) and van der Waals interactions (vdW), determine the docking positions of counterions. For aromatic counterions, stronger ELS interactions stabilize the aromatic counterions around the charged polymer backbone, resulting in enhanced doping efficiency and the suppression of the backbone torsions. Additionally, this uniform counterion distribution facilitates shorter counterion-counterion distances, and consequently smoothens the Coulomb potential landscape for efficient charge carrier transport. In contrast, for alkyl counterions, the pronounced van der Waals (vdW) interactions between the alkyl counterions and the polymer sidechains lead to increased sidechain disorder and larger polymer backbone twisting (disturbed by sidechain disorder). Moreover, the alkyl counterion distributions are

broad, leading to the formation of isolated Coulomb traps. Therefore, these three factors, doping efficiency, polymer backbone planarity, and Coulomb potential landscape, collectively influence the energetic disorder, and eventually, determine the charge transport properties and thermoelectric performance.

Fig. R17. Schematic illustration of how the NCIs affect the charge transport properties and thermoelectric performance.

Despite the complexity of the polymer-counterion system, our work has successfully identified three critical factors that are directly influenced by counterion docking: doping efficiency, polymer conformations, and the Coulomb potential landscape. It is widely acknowledged that these factors significantly impact charge transport and consequently, the thermoelectric performance of polymers (*Chem. Rev.* **2023**, *123*, 7421–7497). To provide a clear understanding of how counterion docking affects these three factors, we have included a detailed discussion from five aspects, including counterion docking, doping efficiency, polymer conformation, Coulomb potential landscape, and energetic disorder:

1. Counterion docking: Through energy decomposition analysis (EDA), we observed that MtBA⁺ exhibited almost twice higher van der Waals (vdW) interactions with the polymer than HPy⁺, while HPy⁺ showed significantly stronger electrostatic (ELS) interactions (Fig. R18a). Quantitative analysis shows that within polymer-MtBA⁺, ELS interactions account for 64.8% of the total NCIs, with vdW interactions contributing the remaining 35.2% (Fig. R18b). However, in the polymer-HPy⁺ systems, the composition of NCIs leans more towards ELS interactions, which constitute approximately 80% with vdW interactions contributing about

20%. To better understand these differences, we evaluated the interaction energies between the counterions and separate parts of the polymer (backbone and sidechain). The ELS interactions significantly influence the attraction between the anionic polymer backbone and cationic counterions (Fig. R18c), and contribute to the repulsion between the polymer sidechain and counterions (Fig. R18d). Additionally, vdW interactions contribute to the attractive interactions between the polymer sidechain and counterions (Fig. R18d). Particularly for MtBA⁺, the attractive vdW interactions outweigh the repulsive ELS interactions with the polymer sidechain, leading to more pronounced attractive interactions with the polymer sidechain compared to HPy⁺.

Fig. R18. **a**, Energy decomposition analysis (EDA) based on the force field for the interaction between P(PzDPP-2FT) and counterions. **b**, Quantitative analysis depicting the contribution of electrostatic or van der Waals interaction to the overall NCIs between P(PzDPP-2FT) and counterions. **c**, EDA for the interactions between the polymer backbone and counterions. **d**, EDA for the interactions between the polymer sidechain and counterions.

We also conducted NCI analysis on the MD simulation results of another aromatic cations, OPy⁺. We observed that OPy⁺ exhibited the same docking behavior as HPy⁺, with a preference

for docking close to the polymer backbone (Fig. R19c) and an approximately 80% composition of ELS interactions (Fig. R18b). These results further confirm that the counterion docking behaviors are determined by the NCI composition.

Fig. R19. MD simulation snapshot of the polymer P(PzDPP-2FT) with **a**, MtBA⁺, **b**, HPy⁺, and **c**, OPy⁺.

2. Doping efficiency: Since ion-exchange doping uses ionic interactions as a driving force (*Nature* **2019**, 572, 634-638), the stronger NCIs between the polymer backbone and the aromatic counterion may contribute to a higher doping level. To experimentally assess the doping efficiency, we employed several techniques, including ultraviolet photoelectron spectroscopy (UPS), alternating current Hall (AC-Hall) measurement, electron paramagnetic resonance (EPR), and X-ray photoelectron spectroscopy (XPS). Ultraviolet photoelectron spectroscopy (UPS) analysis showed that the P(PzDPP-2FT) exchange-doped with HPy⁺ and BMIM⁺ experienced a larger up-shift in the polymer's Fermi level (0.26 eV for HPy⁺ and 0.19 eV for BMIM⁺, respectively) compared to MtBA⁺ (0.13 eV), suggesting higher doping levels in HPy⁺ and BMIM⁺ films (Fig. R20a). To quantitatively assess the charge carrier concentrations (n), the alternating current Hall (AC-Hall) measurement was employed (Fig. R20b). The charge carrier concentrations in the film exchange-doped with HPy⁺ ($9.37 \times 10^{20} \text{ cm}^{-3}$) and BMIM⁺ ($6.02 \times 10^{20} \text{ cm}^{-3}$) were notably higher than that of MtBA⁺ ($9.13 \times 10^{19} \text{ cm}^{-3}$). Although the AC-Hall method tends to slightly overestimate the carrier density due to partial screening by hopping carriers, the higher carrier density in the polymer with HPy⁺ and BMIM⁺ was further supported by EPR and XPS measurements (Fig. R20b). All the experimental results consistently confirmed the enhanced doping level and a higher carrier concentration in the polymer when exchanged with aromatic counterions.

Fig. R20. a, UPS binding energy of the pristine P(PzDPP-2FT) and the exchange-doped P(PzDPP-2FT). **b**, Charge carrier concentration determined by AC-Hall measurement and estimated using XPS and EPR analysis.

3. Polymer conformation. DFT calculations showed that both the polymer's anionization and the counterion interactions contribute to the planarization of the polymer backbone (Fig. R21). In the charged state, the length of the C-C linkage bond between repeating units shortens from 1.45 Å to 1.43 Å, indicating a more quinoid structure (Fig. R21a). Furthermore, the torsion barrier height between repeating units increases from 8.68 to 13.3 kcal/mol (Fig. R21b). Besides, we noticed that interactions between the polymer backbone and the dopant counterion contribute to the further planarization of the doped conjugated backbone. We observed that, despite a slight increase in the C-C linkage bond length of (PzDPP)₂¹⁻-HPy⁺ (or MtBA⁺) compared to (PzDPP)₂¹⁻, the torsion barrier height significantly increased to around 19 kcal/mol when counterions are closely docked to the polymer backbone (Fig. R21b).

Fig. R21. a, DFT-optimized structure of the P(PzDPP-2FT) dimer with different counterions in neutral and doped states. **b**, Relaxed potential energy scan of the torsion angle between the repeating units of P(PzDPP-2FT) dimer in neutral and doped states.

To probe the backbone conformation experimentally, we utilized Raman spectroscopy, which is a useful tool in characterizing the effects of counterions on polymers (*J. Mater. Chem. C* **2013**, *1*, 5638-5646; *J. Am. Chem. Soc.* **2020**, *142*, 15340–15348). For P(PzDPP-2FT) films, we monitored five major peaks related to carbon bond stretching (Fig. R22a). The peak at 1356 cm^{-1} , associated with the symmetric C-C stretch mode, remained unchanged between the pristine and the doped P(PzDPP-2FT) films. However, all peaks related to the C=C stretch mode (1474, 1504, 1520 and 1535 cm^{-1}) in the polymer with HPy⁺ and MtBA⁺ notably shifted to lower wavenumbers. Additionally, after ion-exchange doping, we observed a decrease in the intensity of C=C stretching mode (1474, 1504, 1520 cm^{-1}) in the pyrazine, thiophene, and DPP building blocks, while the intensity of the C=C stretching mode (1535 cm^{-1}) across the entire conjugated backbone increased (Fig. R22b), particularly in the polymer exchanged with HPy⁺. These results imply that the conjugated backbone exhibited a more quinoid and planar structure after ion-exchange doping with HPy⁺ (*Macromolecules* **2017**, *50*, 4227–4234; *J. Phys. D: Appl. Phys.* **2017**, *50*, 073001).

Fig. R22. a, Raman spectra of the pristine P(PzDPP-2FT) film, P(PzDPP-2FT) doped with TDAE, and P(PzDPP-2FT) exchange-doped with MtBA⁺ and HPy⁺. All the Raman spectra were excited at 633nm and the Raman intensity was normalized to the peak at 1356 cm^{-1} . **b**, Intensity ratio of different peaks to the peak at 1356 cm^{-1} . **c**, Diagram illustrating the dominant bond stretches associated with the main Raman active vibrational modes of P(PzDPP-2FT) centered around 1474, 1504, 1520, and 1535 cm^{-1} .

In MD simulations, we observed that the counterion docking position and distribution could significantly influence the polymer conformation. As shown in Fig. R23a, the diffusion of MtBA⁺ into the sidechain regions led to increased sidechain disorder (larger lamellar expansion) and a greater twisting of the polymer backbone (larger π - π distance). Conversely, the polymer backbone surrounded by HPy⁺ maintained its zig-zag conformation and the alkyl sidechain are interdigitated more orderly (Fig. R23b). As a result, both the lamellar and π - π distances in polymer-HPy⁺ end up being smaller than those in polymer-MtBA⁺ (Fig. R23c-d).

Fig. R23. MD simulation snapshot of the polymer P(PzDPP-2FT) with **a**, MtBA⁺ or **b**, HPy⁺ as the counterions. The void regions caused by sidechain disorder in P(PzDPP-2FT)-MtBA⁺ are highlighted with red circles. Cell parameter analysis of **c**, lamellar distances and **d**, π - π stacking distances for the supercell of the polymer with HPy⁺ and MtBA⁺.

The difference in cell parameters induced by counterion docking is further validated by GIWAXS measurements. As mentioned above, we observed no significant changes in the paracrystallinity (g) of P(PzDPP-2FT) when exchange-doped with various cations, and there was no clear correlation between g and electrical conductivity (Fig. R16). Nonetheless, we did observe differences in the GIWAXS spacing parameters (Fig. R24). The polymer films exchange-doped with HPy⁺ and BMIM⁺ consistently exhibited smaller lamellar and π - π stacking distances compared to those exchange-doped with MtBA⁺, across different doping

times (Fig. R24). The diffusion of MtBA⁺ into the sidechain regions led to increased sidechain disorder (lamellar expansion) and a greater twisting of the polymer backbone (larger π - π distance). Conversely, HPy⁺ docking near the polymer backbone contributed to smaller lamellar expansion, and the stronger interactions between the charged polymer backbone and HPy⁺ contributed to the suppression of backbone torsion and smaller π - π stacking distance.

Fig. R24. Variations in **a**, lamellar packing distances and **b**, π - π stacking distances of the exchange-doped P(PzDPP-2FT) at 5 and 30 minutes compared to the pristine polymer film. Data were obtained from the GIWAXS measurements.

4. Coulomb potential landscape. In addition to the polymer conformation, energetic disorder in doped polymer systems also arises from Coulomb traps, which is strongly associated with the counterion docking positions and distributions. Based on the other Reviewers' suggestions, we have recalculated the Coulomb interaction V_C , which will be discussed in detail later (in the response to **Q2** & **Q3**). As shown in Fig. R25a, V_C increases with decreasing polymer-counterion distances (r). However, when $r < 5 \text{ \AA}$ ($r^{-1} > 0.2 \text{ \AA}^{-1}$), V_C levels off to a flat electrostatic potential. This deviation from the classical Coulomb formula ($V_C \propto r^{-1}$) can be attributed to the charge distribution on the counterion (*Nat. Mater.* **2019**, *18*, 242-248; *Nat. Commun.* **2023**, *14*, 1356). Although HPy⁺ docked closer to the polymer backbone, it does not induce significantly deeper Coulomb traps compared to MtBA⁺.

Fig. R25. **a**, Coulombic interaction energies estimated for the polymer-counterion pairs, obtained from MD simulations. **b**, Schematic illustration of the Coulomb potential landscape surrounding the polymer with MtBA⁺ and HPy⁺.

In addition, MtBA⁺ exhibited a broader distribution of distance to the polymer backbone from 4 to 19 Å compared to HPy⁺ (Fig. R26a). This scattered distribution of MtBA⁺ within the polymer leads to a larger average counterion-counterion distance (R_{avg}) than that with HPy⁺ (Fig. R26b), which may generate isolated deep Coulomb traps and thus higher barrier heights for carrier transport (Fig. R25b).

Fig. R26. **a**, Statistical analysis of the polymer-counterion distance (r), defined as the distance between the geometric average line of the polymer backbone and the geometry center of a counterion. **b**, Statistical analysis of the counterion-counterion distance (R), representing the distance between the geometry centers of counterions. All statistical analyses are obtained from the last 200 ps of dynamic simulations.

These observations were supported experimentally. We estimated the average counterion-counterion distance R_{avg} from charge carrier concentrations ($R_{\text{avg}} = n^{-3}$) (*Phys. Rev. B* **2005**, 72,

235202) and found that the R_{avg} of HPy⁺ exchanged film was approximately 4-12 Å smaller than that of MtBA⁺ (Table R2). This reduced R_{avg} enhances the overlap of Coulomb traps, leading to a lower barrier height for carrier hopping and thus efficient charge transport. Based on the Seebeck-conductivity (S - σ) relationship and the SLoT model (*Nat. Mater.* **2021**, *20*, 1414-1421), we extracted the transport coefficient (σ_0) and the depth of the potential wells (W_H) in the polymer (Table R3). The polymer exchanged with HPy⁺ exhibited the highest transport coefficient (σ_0), indicating enhanced mobility compared to MtBA⁺. Additionally, the W_H in the polymer exchanged with HPy⁺ was found to be zero using the experimentally measured carrier density, suggesting a strong overlap of the Coulomb traps in this system.

Table R2. Estimated average ion-ion distance ($R_{avg} = n^{-1/3}$) based on charge carrier densities measured by AC-Hall, XPS, and EPR.

	$R_{AC-Hall}$ (Å)	R_{XPS} (Å)	R_{EPR} (Å)
MtBA ⁺	22.2	28.3	44.4
BMIM ⁺	11.7	20.1	36.6
HPy ⁺	10.2	18.3	40.6

Table R3. The fitted transport parameters for P(PzDPP-2FT) exchange-doped with MtBA⁺, BMIM⁺, and HPy⁺ using the SLoT model.

	MtBA ⁺	BMIM ⁺	HPy ⁺
σ_0 (S/cm)	7.50	44.0	60.0
W_H (meV)	55.5	5.60	0.00

5. **Energetic disorder:** To assess the energetic disorder induced by different counterions, we calculated the density of states (DOS) based on the MD results (Fig. R27). The states lying in the DOS tails strongly correlate with the energetic disorder and charge carrier mobility of polymers (*Nat. Commun.* **2019**, *10*, 2122; *Chem. Mater.* **2019**, *31*, 6889-6899). Hence, we focused on two key parameters at the DOS tail: the depth of the trap states (E_b) (*Nature* **2014**, *515*, 384-388) and the bulk orbital localization length (OLL_b) (*J. Am. Chem. Soc.* **2013**, *135*, 11247-11256). Interestingly, the E_b of the polymer with HPy⁺ was only 20 meV, significantly smaller than that of the polymer with MtBA⁺ (32 meV) and even smaller than the undoped polymer (23 meV). Moreover, the HPy⁺-exchanged polymer showed an extended OLL_b at the band edge, reaching approximately 40 Å, which is significantly larger than that of the undoped

polymer (35 Å) and the polymer with MtBA⁺ (32 Å). These results, characterized by shallower traps and more delocalized orbital lengths, strongly indicate reduced energetic disorder in the polymer when exchanged with HPy⁺.

Fig. R27. Unoccupied DOS distribution and the OLL_b of the undoped and counterion-exchanged P(PzDPP-2FT).

For experimental validation, we extracted activation energies (E_a) from the temperature-dependent conductivity measurements. The polymer films exchanged with HPy⁺ and BMIM⁺ exhibited lower E_a of 36.5 meV and 48.6 meV, respectively, lower than those exchanged with MtBA⁺ (65.6 meV) (Fig. R28a). Furthermore, at different doping levels, the polymer exchanged with HPy⁺ consistently displayed a significantly lower barrier height for charge transport compared to that of MtBA⁺ (Fig. R28b). For example, the Hall mobility of P(PzDPP-2FT) exchange-doped with HPy⁺ ($0.53 \text{ cm}^2 \text{ V}^{-1} \text{ s}^{-1}$) was twice higher than that of MtBA⁺ ($0.21 \text{ cm}^2 \text{ V}^{-1} \text{ s}^{-1}$) (Fig. R28c). This simultaneous enhancement in charge carrier density and mobility proved the improved exchange-doping efficiency and reduced energetic disorder in the polymer with HPy⁺. In addition, the Seebeck-conductivity (S - σ) relationship of the polymer exchange-doped with MtBA⁺ exhibited a shift towards lower conductivity at the same Seebeck coefficient compared to HPy⁺ (Fig. R28d). This shift is an additional indicator of the increased energetic disorder in polymers doped with MtBA⁺, as discussed in previous studies (*Nat. Mater.* **2017**, *16*, 252-257; *Nat. Mater.* **2021**, *20*, 1414-1421; *Adv. Mater.* **2023**, *35*, 2300634). These experimental results provide substantial evidence for the reduced disorder and enhanced charge transport in polymer exchange-doped with HPy⁺.

Fig. R28. **a**, Activation energies of the exchange-doped P(PzDPP-2FT) with three cations. **b**, Activation energies of the polymer films exchanged with MtBA⁺ and HPy⁺ at different doping levels. **c**, Charge carrier concentration and mobility determined by AC-Hall measurement. **d**, Fitting the S - σ relationship using the SLoT model for the polymer films exchanged with three cations.

To provide a clear understanding, we have added a table, which correlates our simulation data with experimental results (Table R4). Additionally, we have rephrased our discussion of the experimental results in the revised manuscript to improve clarity.

Table R4. Correlation between simulations and experimental findings

Counterion docking effect	Simulations	Experiments
Doping efficiency	NCI analysis	UPS, AC-Hall, EPR, XPS
Polymer conformation	DFT calculations and MD simulations	GIWAXS, Raman
Coulomb potential landscape	Static analysis of the counterion-counterion distance	Carrier concentration
	Estimation of the Coulomb interactions	Seebeck-conductivity (S - σ) relationship
Energetic disorder	Calculation of the depth of traps (E_b) and the orbital localization length (OLL_b)	Temperature-dependent conductivity measurements, AC-Hall, Seebeck-conductivity (S - σ) relationship

The above new experimental results and related discussions were all added to the revised manuscript and revised supplementary materials.

Q2.1: In general, we have difficulty following core statements of the authors such as "MtBA+ tends to generate deeper Coulomb traps due to its smaller Q" using the given data, for the following reasons:

Regarding the claims MtBA+ would generate deeper Coulomb traps (see e.g. page 7, line 166): We cannot reconstruct from the authors' data how the characteristic depth of trap states E_b is actually defined and determined and how it correlates with the DOS shown in Fig. 2d. In particular, no significant differences in broadening are apparent to us from the plot in Fig. 2d. Also, it would be helpful to display the Fermi level.

Our response: Thank you for your advice. It is important to clarify that the E_b value here is specifically related to traps resulting from polymer conformation disorder and does not include Coulomb traps generated by counterions. In Section 3.4 of the Supplementary Information, we have provided details for the calculation of the density of states (DOS) and the characteristic depth of trap states E_b . In brief, the DOS was generated by calculating the electronic structure of each polymer chain extracted from the MD simulations. The states lying in the DOS tails strongly correlate with the energetic disorder and charge carrier mobility of polymers (*Nat. Commun.* **2019**, *10*, 2122; *Chem. Mater.* **2019**, *31*, 6889-6899). A closer examination of the DOS tail distribution (Fig. R29) reveals that both the undoped polymer and the polymer with MtBA⁺ exhibited broader DOS tails compared to the polymer with HPy⁺, indicating a larger energetic disorder stemming from the polymer backbone (*Nat. Commun.* **2019**, *10*, 2827). To quantify this energetic disorder, we estimated E_b by fitting an exponential function to the unoccupied DOS tail, as described in the reference (*Nature* **2014**, *515*, 384–388):

$$DOS_{unocc}(E) = \exp\left(\frac{E + c}{E_b}\right)$$

The observation of a smaller E_b value for the polymer with HPy⁺ indicates a reduction in polymer backbone disorder. However, it is important to note that this method does not account for the energetic disorder arising from Coulomb traps.

Fig. R29. Unoccupied DOS distribution of the undoped and counterion-exchanged P(PzDPP-2FT). Compared to the Fig. 2c in the manuscript, the focus has been placed on zooming in on the DOS tail distribution for a more detailed view.

Unfortunately, we were unable to calculate the Fermi level using this method. Calculating the Fermi level requires a combination of the accurate DOS and the charge carrier concentration.

However, our MD simulations were focused only on the disordered region of the polymer, which is more conducive to dopant insertion due to its less organized structure and greater free volume compared to the crystalline region. It is reasonable that we compared the counterion docking process in disordered regions for various cations to reflect the impact of counterions on molecular packing. However, the disordered region, along with other areas such as amorphous and crystalline regions, collectively constitutes the polymer and contributes to the DOS distribution. Therefore, calculating the Fermi level of conjugated polymers is quite challenging due to their disordered nature and the presence of localized states. Despite these challenges, UPS measurements can effectively probe shifts in the Fermi level after doping. As shown in Fig. R30, P(PzDPP-2FT) exchange-doped with HPy⁺ and BMIM⁺ experienced a larger up-shift in the polymer's Fermi level (0.26 eV for HPy⁺ and 0.19 eV for BMIM⁺, respectively) compared to MtBA⁺ (0.13 eV), suggesting higher doping levels in HPy⁺ and BMIM⁺ films.

Fig. R30. UPS binding energy of the pristine and the exchange-doped P(PzDPP-2FT) films.

Q2.2: Comparing Fig. 2e with Fig. S9 it seems like points for HPy⁺ are missing in the main article which are however present in the figure in the SI. Specifically the points at (x,y)=(0.28/0.72) where the omission support the authors' claim. If one adds the missing points also in the main manuscript, the statement of a "flat electrostatic potential" at short distances for HPy⁺ is no longer obvious. Moreover, the range of Coulomb potentials in panel e) would not only be more or less the same for both counterions, but HPy⁺ then shows values for even lower Coulomb potentials. In connection with this, the authors should also explain how the above statement fits to the fact that the Coulomb potentials in Fig. S5 do not differ significantly

for both dopants.

Our response: Thank you for pointing out the discrepancy between Fig. 2e with Fig. S9. After a thorough re-examination of our data, we acknowledge the inadvertent omission of certain data points for HPy⁺, and we appreciate your attention to this detail. Based on your and the other Reviewers' suggestions, we have conducted a comprehensive reassessment of the Coulomb interactions V_C . We agree with your observation that the Coulomb interactions do not differ significantly between the two counterions. To clarify this point, we have included a detailed discussion as follows:

To assess the Coulomb traps, we initially calculated the distance-dependent Coulomb interaction energy (V_C) for both polymer-counterion systems using a recently reported model (*Nat. Commun.* **2023**, *14*, 1356), which describes the ELS interactions of the dopant counterion by considering the monopole and quadrupole moment (Q):

$$V_C = \sum_i \left(\frac{-eq_i^p}{4\pi\epsilon_0\epsilon_r r_{ij}} + \frac{q_i^p}{8\pi\epsilon_0\epsilon_r} \frac{\mathbf{r}_i \mathbf{Q} \mathbf{r}_i}{r_i^5} \right)$$

The dipole moments of all their studied dopant molecules are negligible. However, the counterions in our system can have substantial dipole moments, up to 14.9 Debye, and vary for different counterions, which cannot be disregarded (Fig. R31).

Fig. R31. Magnitude of **a**, the dipole moments and **b**, the quadrupole moments calculated via

DFT-optimized conformation for 28 cations.

Additionally, the initial formula simplifies the dopant counterion as a point charge. However, we observed that the electron density derived from the wavefunction exhibits a strong correlation with the molecular conformation, resulting in the quadrupole moment fluctuating with changes in ion conformations (Fig. R32).

Fig. R32. Electrostatic potential (ESP) mapped molecular vdW surfaces for **a**, HPy⁺ and **b**, MtBA⁺. The color scale represents ESP values ranging from 0.18 and 0.04 a.u. The corresponding conformation for each counterion is labeled with its quadrupole moment magnitude.

Considering the significant differences in charge distribution between alkyl counterion and aromatic counterions, treating them as point charges at the geometric center is inadequate. Therefore, we reevaluated the V_C , considering the partial charge and dipole moments of counterions. The revised formula we used is:

$$V_C = \sum_i \left(\sum_j \frac{q_j^c q_i^p}{4\pi\epsilon_0\epsilon_r r_{ij}} + \frac{q_i^p}{4\pi\epsilon_0\epsilon_r} \frac{\mathbf{D}r_i}{r_i^3} + \frac{q_i^p}{8\pi\epsilon_0\epsilon_r} \frac{\mathbf{r}_i \mathbf{Q} r_i}{r_i^5} \right)$$

Here, $\epsilon = \epsilon_0\epsilon_r$ is dielectric permittivity. i and j indexes the atoms within the polymer and counterions, respectively. q_i^p and q_j^c denote the partial atomic charges of the polymer and the counterions. The distance between atom i in polymer and atom j in counterion is represented by r_{ij} . \mathbf{r}_i is a vector connecting polymer partial charges and counterion center-of-geometries; \mathbf{D} and \mathbf{Q} represent the counterion dipole moment and quadrupole moment, respectively.

Fig. R33. **a**, Distance-dependent V_C for P(PzDPP-2FT) with MtBA⁺ and HPy⁺. **b**, The monopole term (charge-charge interaction), **c**, the dipole term (dipole-charge interaction), and **d**, the quadrupole term (quadrupole-charge interaction) in the multipole expansion of V_C . The polymer-counterion distance (r) was measured between the geometry center of a counterion and the geometric center of the adjacent polymer backbone with two repeating units.

When the distance between the polymer and counterion (r) decreases to below 5.0 Å ($1/r > 0.2 \text{ \AA}^{-1}$), the dipole and quadrupole interactions gradually increase (Fig. R33c-d). For HPy⁺, the average dipole and quadrupole interactions are 0.038 eV and 0.012 eV, respectively. MtBA⁺ has smaller dipole and quadrupole moments (Fig. R31-32), resulting in average dipole and quadrupole interaction of around 0.016 eV and 0.0039 eV (Fig. R33c-d). As a result, these short-range multipole interactions result in a larger up-shift in the electrostatic potential of 0.050 eV for HPy⁺ compared to 0.016 eV for MtBA⁺ (Fig. R33a). Nonetheless, these shifts are tiny when compared to the charge-charge interaction, and they barely change the electrostatic potential for both cations (Fig. R33a-b). Without dipole and quadrupole components, we also observed a flat electrostatic potential at short distances for HPy⁺ (Fig. R33b), deviating from the classical Coulomb formula ($V_C \propto r^{-1}$). Indeed, the dipole and quadrupole components

highlighted the importance of charge distribution on counterions (*Nat. Mater.* **2019**, *18*, 242-248). Although HPy⁺ docks closer to the polymer backbone, it does not induce significantly deeper Coulomb traps compared to MtBA⁺. For simplification in the manuscript discussion on Coulomb interaction (Fig. 2d), we focused on the charge-charge interactions between the polymer backbone and counterions.

Based on above discussion, we have revised the part regarding Coulomb interactions in both the manuscript and the Supplementary Information. These revisions are outlined as follows:

1. We have rewritten the computational part of the manuscript to clarify this point (highlighted by yellow color):

180 polymer systems also arises from Coulomb traps, which is strongly associated with the
181 docking position of counterions. Therefore, we calculated the distance-dependent Coulomb
182 interaction energy (V_C) for both polymer-counterion systems. As shown in Fig. 2d, V_C
183 increases with decreasing polymer-counterion distances (r). However, when $r < 5 \text{ \AA}$ ($r^{-1} >$
184 0.2 \AA^{-1}), V_C levels off to a flat electrostatic potential. This deviation from the classical
185 Coulomb formula ($V_C \propto r^{-1}$) can be attributed to the charge distribution on the
186 counterion^{38,39} (Fig. S15). Although HPy⁺ docked closer to the polymer backbone, it does
187 not induce significantly deeper Coulomb traps compared to MtBA⁺. Apart from the depth
188 of individual Coulomb traps, the energy barrier for charge carrier hopping strongly
189 correlates with the distance between these traps^{10,40}. The scattered distribution of MtBA⁺

2. In Supplementary Information, we have included above discussions about Coulomb interactions in Section 3.5. In addition, we have incorporated Fig. R31-R33 as Fig. S13-S15.

Q2.3: The interpretation of the simulation data in the framework of the concept presented in Fig. 2f and 5g thus seems to us misleading. First, as discussed above, the spread in the depth of the Coulomb potentials seems to be equal, if not reverse, to the ratios presented by the authors in Fig. 2f). On the other hand, the variation in the vertical counterion-dopant distance r is here confounded with an (ir)regular placement of counterions along the polymer.

Our response: Thank you for your advice. Firstly, as discussed in the response to Q2.2, the Coulomb interactions do not differ significantly between HPy⁺ and MtBA⁺. Secondly, we acknowledge your valuable point regarding the potential confusion caused by our initial

presentation of the two distances – the polymer-counterion distance (r) and the counterion-counterion distance (R). It is indeed crucial to distinguish between these two, as they both significantly influence the formation of Coulomb traps. As shown in Fig. R34a and b, the initial positions of HPy⁺ and MtBA⁺ counterions are almost identical. MD simulations revealed distinct docking behaviors: HPy⁺ ions tend to dock closer to the polymer backbone than MtBA⁺, leading to a narrower distribution of r (Fig. R34c). The enhanced stabilization of HPy⁺ around polymer backbone also results in a reduced average R compared to MtBA⁺ (Fig. R34d). Given that the Coulomb interactions are nearly equal for both counterions, this reduced average R may lead to a more pronounced overlap of Coulomb traps, thereby contributing to a smoother Coulomb potential landscape (*Phys. Rev. B* **2005**, 71, 045214). Therefore, both the polymer-counterion distance (r) and the counterion-counterion distance (R) play critical roles in determining the depth and distribution of Coulomb traps.

Fig. R34. Front (left) and side (right) views of the initial positions of counterions before MD simulations for **a**, MtBA⁺ and **b**, HPy⁺. **c**, Statistical analysis of the polymer-counterion distance (r), defined as the distance between the geometric center of the adjacent polymer backbone and the geometry center of a counterion. **d**, Statistical analysis of the counterion-

counterion distance (R), representing the distance between the geometry centers of counterions. All statistical analyses were performed based on the last 200 ps of MD simulations.

We have included the Fig. R34 as Fig. S7 in Section 3.2 of the Supplementary Information. In addition, we have emphasized the distinction between polymer-counterion distance and counterion-counterion distance in manuscript:

187 not induce significantly deeper Coulomb traps compared to MtBA⁺. Apart from the depth
188 of individual Coulomb traps, the energy barrier for charge carrier hopping strongly
189 correlates with the distance between these traps^{10,40}. The scattered distribution of MtBA⁺
190 within the polymer leads to a larger average ion-ion distance than those with HPy⁺ (Fig.
191 S7d), which may generate isolated deep Coulomb traps and thus higher barrier heights for
192 carrier transport (Fig. 2e). In contrast, the shorter ion-ion distance of HPy⁺ can facilitate a
193 strong overlap of the Coulomb traps at high doping concentrations, reducing activation
194 energies for carrier transport. By strategically docking counterions to suitable positions,

Q2.4: the x-axes in panel c) and e) are not consistent, i.e. the numerical values do not match.

Our response: Thank you for your advice. In our statistical analysis of counterions distribution (Fig. R34c) and the dependence of V_C on the polymer-counterion distance r (Fig. R33), the data points do not match on a one-to-one basis. The distribution of polymer-counterion distance r in Fig. R34c was obtained from an extensive statistical analysis of 14,400 polymer-counterion conformations, generated by 72 ions over a 200 ps MD simulation. Then, we selected polymer-counterion conformations at various distances to establish the relationship between the V_C and the polymer-ion distance r (Fig. R33). Calculating V_C for all 14,400 polymer-counterion conformations is challenging due to cost and limited time frame. To address your concern, we have supplemented Fig. R33 with more data points that adequately reflect the trend of V_C variation as a function of polymer-counterion distance r .

Q3.1: Regarding the statement that the quadrupole moment Q of MtBA⁺ and Hpy⁺ differs significantly:

The authors claim a correlation between Coulomb trap depth and Q . However, there is no such panel in Fig S5 which one might think would support the authors argument.

Our response: Thank you for your advice. Q is a measure of the deviation of the atom electron density from spherical symmetry, leading to a consistent observation where the magnitude Q of HPy^+ remains larger than that of MtBA^+ . The atomic quadrupole tensor is a symmetric traceless tensor defined as (*The Quantum Theory of Atoms in Molecules* 2007, p24):

$$\mathbf{Q} = \begin{bmatrix} Q_{xx} & Q_{xy} & Q_{xz} \\ Q_{yx} & Q_{yy} & Q_{yz} \\ Q_{zx} & Q_{zy} & Q_{zz} \end{bmatrix}$$

whose magnitude can be calculated as:

$$|Q| = \sqrt{\frac{2}{3}(Q_{xx}^2 + Q_{yy}^2 + Q_{zz}^2)}$$

If the atomic electron density has spherical symmetry, then $Q_{xx} = Q_{yy} = Q_{zz} = 0$. Molecular quadrupole moments Q can be obtained by summing atomic quadrupole moments. In Fig. R35, we compared the Q values of 28 cations after DFT structural optimization. Specifically, the Q value for HPy^+ was found to be 36.5 Debye Å, which is larger than 8.01 Debye Å for MtBA^+ .

Fig. R35. Magnitude of the quadrupole moments calculated via DFT-optimized conformation for 28 cations.

The correlation between Q and Coulomb traps has been reported in previous studies (*Nat. Mater.* 2019, 18, 242-248; *Nat. Commun.* 2023, 14, 1356). In our investigations, we also observed that HPy^+ with a larger Q exhibited an average quadrupole interactions of 0.012 eV, which is one magnitude larger than 0.0039 eV for MtBA^+ (Fig. R33d). However, we found these quadrupole interactions are relatively small compared to the charge-charge interactions, showing minimal impact on the electrostatic (ELS) potential landscape for both cations. Therefore, we have removed the discussions related to quadrupole interactions and revised the

manuscript to consider ELS interactions as a unified entity (highlighted in yellow):

135 Using energy decomposition analysis (EDA) based on the force field^{27,28}, we identified
 136 NCI compositions as the key factor behind the different counterion docking behaviors
 137 between MtBA⁺ and HPy⁺. As shown in Fig. 2b, MtBA⁺ exhibited almost twice higher van
 138 der Waals (vdW) interactions with the polymer than HPy⁺. In contrast, HPy⁺ showed
 139 significantly stronger electrostatic (ELS) interactions, accounting for 80% of its NCIs,
 140 compared to 65% for MtBA⁺ (Fig. S8b). The accumulation of the vdW interactions across
 141 multiple neutral polymer sidechains drew MtBA⁺ into the sidechain regions, while the
 142 more substantial ELS interactions made HPy⁺ dock closer to the charged polymer
 143 backbone (Fig. S8c-d). To validate this conclusion, we conducted NCI analysis on the MD
 144 simulation results of another aromatic cation, OPy⁺. We observed that OPy⁺ exhibited the
 145 same docking behavior as HPy⁺, with a preference for docking close to the polymer
 146 backbone (Fig. S6) and an approximately 80% composition of ELS interactions (Fig. S8).
 147 These results further confirm that the counterion docking behaviors are determined by the
 148 NCI composition. Since ion-exchange doping uses ionic interactions as a driving force^{15,29},
 149 the stronger NCIs between the polymer backbone and the counterion may contribute to a
 150 higher doping level.⁴¹

152 **Fig. 2| Understanding of the counterion effects on the energetic disorder of the doped**
 153 **polymers. a,** MD simulation snapshot of the polymer P(PzDPP-2FT) with MtBA⁺ (top) or

Q3.2: From the data shown in Fig S9, it is not so clear from our point of view that the Q-values differ strongly between the two counterions. For MtBA⁺, for example, it is not possible to say for sure from the data shown whether Q=10 Debye or 25 (or 35) because the lines hardly differ in the range in which the data are measured.

Our response: Thank you for your advice. We appreciate your attention to the detail and understand your concerns about the clarity of the differences in Q between the two counterions.

As mentioned in our response to Q3.1, DFT calculations revealed that HPy⁺ has a larger Q value compared to MtBA⁺, which is attributed to the greater deviation of HPy⁺'s atomic electron density from spherical symmetry. To provide a clearer understanding, we have presented the electrostatic potential (ESP) map for HPy⁺ and MtBA⁺. As shown in Fig. R36, HPy⁺ concentrates its positive ESP on the pyridine ring, while MtBA⁺ primarily localizes its ESP on the nitrogen atom, resulting in a smaller Q value. Additionally, we observed the fluctuations in the Q value with changes in ion conformations. These fluctuations are due to the strong correlation between electron density and the molecular conformation (Fig. R36a-b). Therefore, we extracted 200 conformations of each counterion from MD simulations to calculate their respective Q values. As shown in Fig. R36c, the average $|Q|$ for MtBA⁺ is 10.5 Debye Å, while HPy⁺ averages around 22.2 Debye Å. We acknowledge that there might be a fitting error due to the variation in Q with counterion conformation and the limited number of data points. Nevertheless, the magnitude of Q is consistently larger for HPy⁺ than that of MtBA⁺.

Fig. R36. Electrostatic potential (ESP) mapped molecular vdW surfaces for **a**, HPy⁺ and **b**, MtBA⁺. The color scale represents ESP values ranging from 0.18 and 0.04 a.u. The corresponding conformation for each counterion is labeled with its quadrupole moment magnitude. **c**, the average quadrupole moment magnitude with 200 conformations for each counterion extracted from MD simulations.

Q3.3: The authors should explain why Fig. S5 and Fig. S9 are not consistent. If one extracts

from Fig. S5 for MtBA⁺ the Q value (~8 Debye), the 1/r value (0.3 Å⁻¹) and the value for V_c (-0.62 eV) then the Q value fits the fitted one in Fig. S9a, but the (1/r, V_c) point does not match the fit.

Our response: Thank you for your advice. The discrepancy you have noticed arises from the different simulation setups employed for each figure. Fig. S5 shows the electrostatic interaction between the 3 chains of the π -stacked polymer dimers and a counterion, with data obtained from the molecular docking. In contrast, Fig. S9 shows the interaction between a single chain of the polymer dimer and a counterion as extracted from MD simulations. Due to these differences, a direct comparison between Fig. S5 and Fig. S9 is inherently limited. However, it is reasonable to use both methods to compare the V_C of different counterions under the same calculation condition.

Q3.4: On page 9, line 211 the authors claim: "These results suggest that aromatic cations are more favorable as the counterions for the doped polymers, which is consistent with the counterion docking screening results." However, Fig. S5 shows that BPy⁺, for example, has a similar Q value to MtBA⁺, but the conductivity differs significantly between the two. Overall, Fig. S5 suggests that the differences in Q and V_c values between individual classes of counterions are smaller than the scatter within classes. However, this is not true for the conductivities (Fig. 3b). For us, the above-mentioned correlation between quadrupole moment, depth of Coulomb traps and conductivity remains difficult to understand also for this reason.

Our response: Thank you for your insightful comments. We agree that Q and V_c are not the sole determinants of conductivity in doped polymers, and we have considered ELS interactions as a unified entity. As shown in Fig. R37, we found no direct correlation between the quadrupole moment and conductivity. For example, with BMIM⁺ and BPy⁺, their quadrupole moments are 5.67 and 4.60 Debye Å, respectively, yet BPy⁺ exhibit a 34.4 S/cm higher conductivity than BMIM⁺. Additionally, when comparing the same type counterions, like BPy⁺ and OPy⁺, despite remarkable differences in quadrupole (37.4 Debye Å) moments, their conductivities remain nearly the same. These results suggest that the absence of a direct

correlation between quadrupole moments and conductivity.

Fig. R37. The relationship between quadrupole moments with electrical conductivity.

As we discussed in response to **Q1**, the complexity of the polymer-counterion system does not allow for a straightforward quantification of the impact of one or two molecular parameters on conductivity. Nevertheless, our work primarily demonstrates that the counterion docking, driven by NCIs, significantly affects several key factors: the doping efficiency, the polymer conformation, and the Coulomb potential landscape. These factors collectively influence the energetic disorder, ultimately determining the charge transport and the thermoelectric performance of polymers. Thus, while differences in Q and V_C for various counterions provide valuable insights, they represent only a part of the factors influencing the conductivity of doped polymers.

Q4: In our view, there are also some ambiguities regarding the presentation and evaluation of the experiments: The authors' statement on page 11, line 240: "The polymer films exchanged with HPy+ consistently exhibited higher conductivities while maintaining comparable Seebeck coefficients compared to the other two cations." is ambitious, given that in the case of both conductivity and Seebeck coefficient, Hpy+ and BMIM are predominantly the same when the stated errors are considered (see Fig. 4a and b). Consequently, it would be appropriate to indicate error bars also for the power factor (Fig. 4c).

Our response: Thank you for your advice. We have included the error bars for all the measured thermoelectric data (Fig. R38). With these error bars taken into consideration, the maximum

power factor of the polymer exchange-doped with HPy^+ can reach up to $182.6 \mu\text{W m}^{-1} \text{K}^{-2}$. This is a significant increase compared to the power factors of $112.2 \mu\text{W m}^{-1} \text{K}^{-2}$ and $26.4 \mu\text{W m}^{-1} \text{K}^{-2}$ for BMIM^+ and MtBA^+ exchanged polymers, respectively. The inclusion of error bars not only enhances the accuracy of our data representation but also highlights the superior thermoelectric performance of polymers doped with HPy^+ compared to other cations.

Fig. R38. **a**, Electrical conductivities, **b**, Seebeck coefficients, and **c**, power factors of the P(PzDPP-2FT) films exchanged with three different cations at various doping time.

Due to the length constraints of the main text, we have included the most crucial power factor data of this article alongside values reported in other literature in Fig. 3d (as shown below). Additionally, the thermoelectric performance test graph Fig. R38 has been added as Fig. S17 in Section 4 of the Supplementary Information:

222

Fig. 3| Experimental validation of counterion docking strategy. **a**, Chemical structures of the polymers (P(PzDPP-2FT), *o*FBDPPV & N2200), n-type dopants (TDAE & CoCp₂), and the alkyl, imidazole, and pyridine cations used for this study. **b**, Comparison of the electrical conductivities of the ion-exchange doped P(PzDPP-2FT) films. **c**, Comparison of the electrical conductivities of the *N*-DMBI-doped and ion-exchange doped *o*FBDPPV and N2200 polymer films. TDAE was utilized as the n-dopant for the ion-exchange doping of *o*FBDPPV and N2200. **d**, Comparison of the power factors and conductivities among this work and other reported n-doped polymers.

Q5: The discussion and determination of charge carrier densities on page 11 and in the SI raises

more questions than it answers: Page 11, line 252/253: “Ultraviolet photoelectron spectroscopy (UPS) analysis showed that different counterions showed similarly high doping levels” this result is inconsistent with a significant (one order of magnitude) change in carrier densities as stated in the manuscript. Instead an independent assessment of the ion densities would be helpful. If the charge carrier density is changed by one order of magnitude, we would expect that also the Fermi level is changed between different counterions. However, the UPS measurements in Fig. S12 show no such shift in Fermi level.

Our response: Thank you for your points. Upon reading your comments, we recognized the need to re-examine our data to ensure accuracy and consistency. We have re-conducted the UPS measurement and found significant differences in the secondary electron cut-off (SECO) shift among the films exchange-doped with the three cations. As shown in Fig. R39, P(PzDPP-2FT) exchange-doped with HPy⁺ showed the highest up-shift in SECO by 0.26 eV compared to the films exchange doped with BMIM⁺ and MtBA⁺ (0.19 eV and 0.13 eV). These revised experimental results align more closely with the differences in the measured charge carrier concentrations that we reported.

Fig. R39. UPS binding energy of the pristine and the exchange-doped P(PzDPP-2FT) films with MtBA⁺, BMIM⁺, and HPy⁺.

In the revised manuscript, Figure S12 was replaced by the new data and added as Fig. 4d. We have changed the statement regarding UPS as follows (highlighted by yellow color):

285 In the theoretical modeling, it is intriguing that the docking position and distribution
286 of counterions significantly affect the counterion-induced energetic disorder. Ultraviolet
287 photoelectron spectroscopy (UPS) analysis showed that the P(PzDPP-2FT) exchange-
288 doped with HPy⁺ and BMIM⁺ experienced a larger up-shift in the polymer's Fermi level
289 (0.26 eV for HPy⁺ and 0.19 eV for BMIM⁺, respectively) compared to MtBA⁺ (0.13 eV),
290 suggesting higher doping levels in HPy⁺ and BMIM⁺ films (Fig. 4d). To quantitatively
291 assess the charge carrier concentrations (n), the alternating current Hall (AC-Hall)
292 measurement was employed (Fig. 4e). The charge carrier concentrations in the film

Q5.3: The densities determined by different techniques vary a lot. How large is the error of the different techniques and is it then still justified to claim differences in the charge carrier density? Apart from that, the value of Hall measurements for hopping systems is rather controversial.

Our response: Thank you for raising this important point about the variability in charge carrier density measurements in organic electronics. In organic electronics, quantitatively defined charge carrier concentration is very difficult and challenging. We acknowledge that quantitatively defining charge carrier concentration in these materials is indeed challenging due to the limitations and specific characteristics of the commonly used measurement techniques.

1. **Electron Paramagnetic Resonance (EPR):** EPR is widely used for estimating charge carrier concentration but often tends to underestimate it. This method is effective in measuring unpaired electrons, such as polarons, yet at high doping levels, species like bipolarons, which are EPR silent, remain undetectable (*Chem. Phys. Chem.* **2003**, *4*, 49).
2. **AC-Hall measurement:** while the Hall measurement is another prevalent technique, the Hall coefficient typically overestimates the true carrier density and overestimate charge carrier concentration due to partial screening by hopping carriers (*Sci. Rep.* **2016**, *6*, 23650; *Nat. Mater.* **2016**, *15*, 896).
3. **X-ray Photoelectron Spectroscopy (XPS):** Recently, XPS has been explored as a method to quantify charge carrier concentration, as reported by H. Sirringhaus et al. (*J. Am. Chem. Soc.* **2022**, *144*, 3005). XPS can provide information on the atomic species near the film

surface, with the ratio of peak integrals corresponding to the molar ratio of atoms in each species. which can help to determine the charge carrier concentration by the ratio of polymer polaron to the intrinsic polymer. However, if an element exists in more than one fragment of the polymer backbone or the dopant counterion, the manual peak fitting can be challenging.

Given these factors, while these methods may not offer precise quantitative measurements of charge carrier concentration, they can still provide valuable qualitative insights. We believe it is reasonable to use all these techniques for a qualitative comparison of charge carrier densities across different samples. **Furthermore, we have included above discussions in Section 4 of the revised Supplementary Information to clarify this point.**

Q5.4: It is not transparent to us which of the 6 peaks present were used in the analysis of the XPS spectra to determine the charge carrier concentration. In addition, we find it somewhat surprising that in the undoped polymer the fit does not agree with the measurement data at exactly the point where the blue peak assigned to a polaron is located in the doped polymers. With said peak, the fit in Fig. S14a would presumably also fit, but this is passed over here without comment.

Our response: We appreciate your attention to these details and agree that a thorough and transparent explanation is essential for a clear understanding of XPS analysis. XPS allows us to characterize the atomic species present near the film surface because the ratio of peak integrals for a given transition are proportional to the molar ratio of atoms in each species (Handbook of x-ray photoelectron spectroscopy, Physical Electronics Division, PerkinElmer Corporation, 1992; *J. Am. Chem. Soc.* **022**, 144, 3005). Fig. R40a displays the chemical structures of materials, including the polymer P(PzDPP-2FT), the dopant TDAE, the three cations (MtBA⁺, BMIM⁺, HPy⁺), and the corresponding anion TFSI⁻ in each ion liquid. Initially, we analyzed the XPS nitrogen 1s spectra of the exchange-doped P(PzDPP-2FT) system. However, due to the presence of multiple types of N (1s) in this system, assigning peaks accurately becomes challenging and complex.

Fortunately, only the polymer P(PzDPP-2FT) and the anion TFSI⁻ contain oxygen atoms, allowing for a clear distinction between the C=O in P(PzDPP-2FT) and the S=O in TFSI⁻. The

O 1s peak of S=O in TFSI⁻ is at 533.3 eV (*Batteries& Supercaps* **2022**, 5, e20220030), while the C=O peak in the polymer is at 532.6 eV (*Polymers* **2019**, 11, 2023). Therefore, we focused our analysis on the XPS O 1s spectra of the undoped P(PzDPP-2FT) and the exchange-doped P(PzDPP-2FT) with the three counterions. As shown in Fig. R40c, the undoped P(PzDPP-2FT) exhibited a main peak at 532.6 eV, corresponding to the O in the DPP moiety. After exchange-doping with the three cations, a new peak emerges at 531.5 eV (Fig. R40b), indicating a new species formed during the doping process. Since the S=O in TFSI⁻ has a higher binding energy than the C=O in DPP, this shift to a lower binding energy is attributed to the O atoms on the doped polymer backbone (labeled as C=O^{δ-}), which possess fewer positive charges compared to the pristine polymer.

Fig. R40. **a**, Chemical structure of the polymer P(PzDPP-2FT), the dopant TDAE, the three cations (MtBA⁺, BMIM⁺, HPy⁺), and the corresponding anion TFSI⁻ in each ion liquid. **b**, Normalized O(1s) XPS spectra for the undoped P(PzDPP-2FT) and the exchange-doped P(PzDPP-2FT) with the three cations. O(1s) XPS spectra and fitted curves of **c**, the undoped P(PzDPP-2FT) and P(PzDPP-2FT) exchange-doped with **d**, MtBA⁺, **e**, BMIM⁺, **f**, HPy⁺.

Consequently, the charge carrier concentration is determined by the ratio of C=O^{δ-} to the original C=O. Notably, the increase of the C=O^{δ-} peak was more pronounced in the polymer exchange-doped with HPy⁺ compared to that with MtBA⁺ (Fig. R40b), indicating a higher doping level. This observation is corroborated by the fitting results, where the ratio of C=O^{δ-} / C=O in the polymer exchange-doped with HPy⁺ was 0.571, significantly higher than the 0.108

ratio in the polymer exchange-doped with MtBA⁺. Therefore, the estimated carrier concentration in the polymer exchange-doped with HPy⁺ is 1.64×10^{20} , which is 3.7 times higher than that in the polymer exchange-doped with MtBA⁺ (as detailed in Table R5).

Table R5. Carrier concentrations n_{XPS} estimated from XPS analysis.

		⁻³⁾
P(PzDPP-2FT)	MtBA ⁺	4.40×10^{19}
	BMIM ⁺	1.23×10^{20}
	HPy ⁺	1.64×10^{20}

In Supplementary Information, we have added Fig. R40 as Fig. S31 and above discussions in the Section 4. Additionally, the carrier concentrations n_{XPS} estimated from XPS analysis in Table R5 have been added to Table S5.

Q5.5: Apart from all these methodological difficulties, the numbers do not fit either. In Figure 3b and 4a the maximum differences in conductivity are a factor of 3-6, but this is much less than the factor of 10 in density times the factor of 2 in conductivity that the authors claim here.

Our response: Thank you for your comments. To address the confusion regarding the materials and their performance, we have provided a detailed comparison in Table R6 of the electrical performance for the polymer exchange-doped with MtBA⁺ and HPy⁺. Electrical conductivity (σ) is a function of both carrier concentration (n) and mobility (μ), as described by the equation $\sigma = en\mu$. Our data indicates that the polymer exchange-doped with HPy⁺ exhibited a 1.5 to 4.2 fold increase in μ and a 1.3 to 10.2 fold increase in n compared to MtBA⁺. As discussed in the response to Q5.3, EPR measurements tend to underestimate the charge carrier concentration, while Hall measurements generally overestimate it. Taking into account the relatively accurate results from XPS analysis, where the n of HPy⁺ is 3.7 times that of MtBA⁺, and multiplying it by the 1.5 to 4.2 fold increase in mobility, we expect the conductivity to be higher by a factor of 5.5 to 15.5. This expectation aligns well with our measurement results, showing a 5.4 to 14.9 fold increase in conductivity. Both Table R6 and above discussions were added to the revised

Supplementary Materials to clarify this point.

Table R6. Comparison of the electrical performance of the polymer exchange-doped with MtBA⁺ and HPy⁺.

Measurement	MtBA ⁺	HPy ⁺	Factors (HPy ⁺ /MtBA ⁺)
σ	23.4 ± 2.4	155.2 ± 16.5	5.4~8.2
(S cm ⁻¹)	14.8 ± 3.8	149.2 ± 14.3	7.3~14.9
μ_{Hall} (cm ² V ⁻¹ s ⁻¹)	0.21 ± 0.05	0.53 ± 0.14	1.5~4.2
n_{Hall} (cm ⁻³)	9.12×10^{19}	9.37×10^{20}	10.2
n_{XPS} (cm ⁻³)	4.40×10^{19}	1.64×10^{20}	3.7
n_{EPR} (cm ⁻³)	1.14×10^{19}	1.49×10^{19}	1.3

Q6: The authors claim on page 13 line 256 that “conductive AFM allowed a direct comparison of the conductivity differences of these polymer films (Fig. S18). While the films exchanged with MtBA⁺ displayed islands of conductive regions with limited connectivity, the films exchanged with HPy⁺ and BMIM⁺ exhibited clear one-dimensional transmission channels.” However, the AFM images shown in Fig. S17 all look the same, depending on which cutout you look at. Looking at the images shown, it is clear that the tip destroys the sample surface and the images therefore have very little informative value. Apart from that, one-dimensional transport would not be advantageous, as it is very sensitive to defects.

Our response: Thank you for your advice. We believe there is some misunderstanding regarding the measurement for AFM height images and conductive AFM images.

1. **AFM:** AFM is usually used to assess the surface topography of materials with high spatial resolution, which can reflect the surface roughness of the polymer films (*Adv. Mater.* **2018**, *30*, 1802850), macroscopically evaluate the crystallinity (*Nat. Commun.* **2021**, *12*, 58), and compare the miscibility of polymer and dopants (*Adv. Mater.* **2014**, *26*, 2825-2830). The polymer P(PzDPP-2FT) is relatively amorphous and exhibits excellent miscibility with various dopants and counterions, which results in similar surface morphologies across the pristine and doped films (Fig. R41). This is consistent with our previous findings on films doped with

different dopants like *N*-DMBI, TDAE and CoCp₂ (*Nat. Commun.* **2021**, *12*, 5723).

Fig. R41. AFM height images of the pristine and the exchange-doped P(PzDPP-2FT) with MtBA⁺, BMIM⁺, and HPy⁺.

2. **Conductive AFM:** conductive AFM (c-AFM) is a technique employed to probe local electrical properties by detecting current feedback from the films under applied voltage (*Nat. Electron.* **2019**, *2*, 221; *J. Am. Chem. Soc.* **2023**, *145*, 5242). In c-AFM, the current values obtained at each point of the sample surface reflect the local conductivity of the material. Higher current values indicate areas of higher conductivity, while lower current values suggest regions of lower conductivity. In organic electronics, c-AFM has been widely used to understanding the electrical characteristics of polymers. For example, Sangmin *et al.* used conductive AFM to compare the charge transport properties of CPE-K₁₂ and CPE-K₄₈, which, despite having similar AFM images, exhibited different current in conductive AFM (*Adv. Funct. Mater.* **2023**, 2310852). Similarly, Yang *et al.* employed conductive AFM to identify conductive regions in BBL:PEI_{lin} film, where a pronounced crack line indicated non-conductive areas (*Nat. Commun.* **2021**, *12*, 2354). In our study, the conductive AFM images, particularly the dark crack lines in Figure R42, represent non-conductive regions rather than areas damaged by the AFM tip. It is also important to note the variation in the conductive AFM scales. When exchange-doped with HPy⁺, the surface of the polymer film exhibited large connected areas with current as high as 299 pA. In contrast, when exchange-doped with MtBA⁺, the film showed isolated conducting areas with a maximum current of only 28.2 pA. These images,

therefore, offer valuable insights into the electrical properties of our polymer films on a micrometer scale, complementing the surface morphology information provided by AFM.

Fig. R42. The conductive AFM current images of the pristine and the exchange-doped P(PzDPP-2FT) with MtBA⁺, BMIM⁺, and HPy⁺. The measurement was conducted in a nitrogen atmosphere, and the voltage bias was 100 mV.

3. Clarification on “one-dimensional transport channels”: We acknowledge that the term “one-dimensional transport channels” might have been misleading. Charge transport in thermoelectric devices is predominantly bulk transport, not confined to interfaces like field-effect transistors (FET). We intended to describe continuous transmission pathways in the polymer exchange-doped with HPy⁺ compared to MtBA⁺ (Fig. R42), not limited to the one-dimensional movement of carriers. **We apologize for any confusion caused and have changed the term “one-dimensional transport channels” to “higher conductivity and more continuously conductive transport channels” to reflect our findings more accurately.**

Q7: In the discussion of the absorption spectra in Fig. 5, the "peak" at 950 nm is assigned to a "compressed polaron". However, there is no peak at 950 nm, but a discontinuity, which is usually caused by switching to another detector or grating. Moreover, what is called a peak here is already visible in the MtBA⁺ spectra in the first measurement, in fact this looks very similar to the HPy⁺ measurement after 10 min. How can this be reconciled with the concept of

compressed polarons that the authors assume here? Apart from our doubts in the interpretation of the measurements, the argument of the authors concerning the formation of compressed polarons collides with the claim that the charge transport should be more delocalized (p.14, l. 325).

Our response: Thank you for your observations regarding the absorption spectra featured in Fig. 5. We understand your concerns about the interpretation of the spectral features and the implications for charge transport.

1. **Characterization of the peak at 950 nm:** The peak around 950 nm is a spectroscopic characteristic, not an artifact from detector switching. In Fig. R41, we observed a sudden signal jump at 860 nm, which consistent across pristine and doped polymer films, indicating detector switching. However, the wide absorption band extending from 890 nm to 1400 nm represents actual optical absorption, which changes with the doping levels. Additionally, the increase in absorption around 950 nm with increased doping level, contrasted with the relatively constant absorption at lower energies (~1160 nm), suggests the presence of two different species.

Fig. R41. The combined UV-vis-NIR absorption spectra of the polymer films exchange-doped with MtBA⁺ and HPy⁺. (Fig. 5a and 5b in the original manuscript)

2. **Identification of compressed polarons:** Upon doping, the polymer's neutral bandgap absorption at ≈ 728 nm decreases, with two new absorption bands emerging at 2000-2500 nm and 1160 nm (Fig. R42a-b). Typically, low-energy optical absorption from polarons in chemically doped conjugated polymers is observed near 2480 nm, while bipolarons usually

exhibit higher energy transitions (*J. Am. Chem. Soc.* **2012**, *134*, 10852–10863; *Adv. Mater.* **2021**, *33*, 2000228). Therefore, the absorption bands at 2000-2500 nm and 1160 nm are attributed to polarons and bipolaron bands, respectively, which is further supported by our time-dependent density function theory (TD-DFT) calculations (Fig. R42c). With increased doping, polymer films doped with HPy⁺ show a new peak at 950 nm alongside a gradual decrease in the polaron band, suggesting the emergence of a new polaron species. Previous studies reported that this blue-shifted absorption could be compressed polarons with enhanced ion pairing which display polaron-like behavior (*J. Am. Chem. Soc.* **2012**, *134*, 10852–10863; *J. Phys. Chem. C* **2014**, *118*, 114–125). TD-DFT simulations of the compressed polarons were also in good agreement with the experimental data, exhibiting absorptions between the neutral band and the bipolaron band (Fig. R42c).

Fig. R42. UV-vis-NIR absorption spectra of the polymer films exchange-doped with **a**, MtBA⁺ and **b**, HPy⁺, respectively. **c**, TD-DFT simulated spectra of the P(PzDPP-2FT) trimer with the four states in different ratios (pristine: polaron: bipolaron: compressed polaron = 0.3: p: 0.15: cp). **d**, TD-DFT simulated spectra of the P(PzDPP-2FT) trimer at four electronic states: pristine, polaron, bipolaron, and compressed polaron (from top to bottom). The TD-DFT calculations were performed at CAM-B3LYP/6-311G(d,p) and the 100 excited states with the lowest energy were calculated for each P(PzDPP-2FT) trimer.

A polaron has a spin of 1/2, whereas a bipolaron has no spin. Further evidence for compressed polarons can be seen from the EPR measurements, which is hence expected to give information about polarons and bipolarons generated by doping. Fig. R43 presents an overview of the polaron spin states in polymer chains from the neutral state to different doping levels.

Upon doping, an additional electron is introduced into one of the polymer chains, leading to the formation of unpaired electrons, known as polarons, which contribute to the EPR signal. Further increasing the doping level, two polarons within the same chain or neighboring chains can couple together to form a bipolaron (*Macromolecules* **2017**, *50*, 4867–4886). Bipolarons are EPR silent and result in a decrease in the spin density (N_S), despite an increase in carrier concentration (N_D) (*Acc. Chem. Res.* **1996**, *29*, 417–423). At a high doping level, the combination of a bipolaron and an unpaired polaron would give rise to the EPR signal and result in an increase in N_S again (*J. Phys. Chem.* **1996**, *100*, 15644–15653). However, it is possible that polarons in the same chain may be compressed to shorter lengths to form an array or polaron lattice, i.e., compressed polarons (*Phys. Rev. Lett.* **1985**, *55*, 308–311; *J. Phys. Chem. C* **2014**, *118*, 114–125).

Fig. R43. Schematic diagrams illustrating the different possibilities of the spin pairing in polymer chains from neutral to different doped states.

To probe these polaron transformations during doping, we performed EPR measurements for polymer films with various doping times. As shown in Fig. R44, the double-integrated EPR intensity first decreased, indicating a transition from unpaired polarons to spinless bipolarons. Subsequently, a higher doping level led to a re-increase in spin density, implying the formation of the combination of bipolaron and polarons, or the emergence of compressed polarons.

Fig. R44. EPR integrated intensity of the polymer films exchange-doped with MtBA⁺ and HPy⁺ at different doping times.

To better distinguish between bipolarons and compressed polarons, we performed temperature-dependent EPR measurements (Fig. R45). A dramatic decrease in EPR intensity with increasing temperature was observed for polymer exchanged with both HPy⁺ and MtBA⁺, suggesting a triplet ground state (*Chem. Rev.* **2013**, *113*, 7011–7088). The EPR intensity was fitted by Bleaney-Bowers equation, revealing the energy gap between the singlet and triplet states (ΔE_{S-T}) of 0.0157 kcal mol⁻¹ for the polymer with HPy⁺ and 0.0223 kcal mol⁻¹ for the polymer with MtBA⁺. These results suggest that at low temperatures, compressed polarons (existing in triplet state) are more stable than bipolarons (existing in singlet state). As the temperature increases, unpaired compressed polarons gradually combine to form bipolarons. Although distinguishing between compressed polarons and bipolarons is challenging at room temperature, the presence of triplet compressed polarons at low temperatures is evident from these results. In addition, the larger ΔE_{S-T} value of the polymer exchanged with MtBA⁺ suggests that compressed polarons are more energetically stable compared to those with HPy⁺. This may explain the more pronounced intensity ratio observed in the absorption band characteristic of compressed polarons at lower doping levels in the polymer film exchange-doped with MtBA⁺.

Fig. R45. Temperature-dependent EPR of P(PzDPP-2FT) exchanged with **a**, HPy⁺ and **b**, MtBA⁺ and **c**, the EPR integrated intensity and corresponding Bleaney-Bowers equation fitting result. The temperature is in the range of 10 K to 100 K.

3. Relationship between polarons and charge transport: Currently, the impact of the polarons on charge transport in polymers is still controversy. While polaronic polymers like polyaniline are known to exhibit high conductivity and metallic like states (*Phys. Rev. Lett.*

1987, 59, 1464–1467), the absence of polarons also have been discovered to enhance the electrical conductivity and Seebeck coefficient of polymers (*Nat. Mater.* **2014**, 13, 190–194). This underlines a broader uncertainty in the field regarding how different types of charge carriers influence the electrical properties of polymers. In our work, we cannot conclusively determine that compressed polarons or bipolarons are directly related to the efficiency of charge transport.

Combined with the Reviewer 2’s suggestions, we agreed that the details of electronic states go beyond the scope of this study and won’t affect our conclusion. **Therefore, we removed this part of the discussion from our manuscript and added to section 4 in Supplementary Information.**

Q8: Apart from the scientific aspects, it is difficult to follow the line of argument in the manuscript and the approach without continuously looking up the SI.

Our response: Thank you for your advice. As you have mentioned, we have used “a variety of methods both in experimental and modeling” in this work. Due to the limited figure numbers and lengths required by the journal, we could not put all the figures and discussions in the manuscript. During the revision, we have thoroughly revised the manuscript for better clarity. In addition, we added a new figure, Figure 5 (Fig. R15 here), to illustrate the key concept in this manuscript as follows:

Fig. R46. Schematic illustration of how the NCIs affect the charge transport properties and

thermoelectric performance.

The following discussions were also added to clarify the concept of this work:

“For aromatic counterions, the stronger electrostatic interaction (ELS) stabilizes the aromatic counterions around the charge polymer backbone, resulting in enhanced doping efficiency and the suppression of the backbone torsions. Additionally, this uniform counterion distribution facilitates shorter counterion-counterion distances, and consequently smoothens the Coulomb potential landscape for efficient charge carrier transport. In contrast, for alkyl counterions, the pronounced van der Waals (vdW) interactions between the alkyl counterions and the polymer sidechains lead to increased sidechain disorder and larger polymer backbone twisting (disturbed by sidechain disorder). Moreover, the alkyl counterion distributions are broad, leading to the formation of isolated Coulomb traps. Therefore, these three factors, doping efficiency, polymer backbone planarity, and Coulomb potential landscape, collectively influence the energetic disorder, and eventually, determine the charge transport properties and thermoelectric performance.”

Reviewer 4:

Our response: We greatly appreciate the efforts you have invested in the review process. We hope that, after these revisions, you may support the publication of this manuscript in *Nature Communications*.

REVIEWER COMMENTS

Reviewer #1 (Remarks to the Author):

I am very grateful for the extremely detailed response of the authors and overall I think the paper has moved closer to publication, but I cannot follow their conclusion that dipole and quadrupole moments do not correlate with experiment. I think Fig R4 explicitly demonstrates such a correlation. I would agree that other factors are also important, but I cannot follow the conclusion that the quadrupole moments do not correlate. For the dipole moments this is not so clear.

I would appreciate if Fig R4 was included and discussed in a revised manuscript.

Finally, there is something wrong with the Y-axis of Fig 5. The interaction energies are in the order of thousands of kcal/mol and this is either not normalized or not possible.

Reviewer #2 (Remarks to the Author):

The authors have properly addressed all my comments.

I am satisfied with the authors toning down on the nature of electronic carriers.

I believe that the ambiguous discussion has been removed from the main context, and the gist of this research has become clearer.

Although important discussions have moved to SI, the main text still presents high-quality experimental results.

This is an excellent work that deserves publication.

Reviewer #3 (Remarks to the Author):

The authors have come up with an extensive response. However, they have still not been able to convince us in some key areas. This essentially concerns two points:

- The argumentation regarding the Coulomb potential landscape, disorder and doping efficiency
- The link between simulations and experiments

In general, we could accept the manuscript if it is revised with a focus on the experiments and the position of the ions with respect to the backbone (where significant differences show up). The discussion of the direct connection to the conductivity would then need to be shortened and leave room for other effects such as screening of the ionic charge or the positioning of dopants in different regions of the polymer (amorphous or crystalline). In addition, two misleading cartoons must be removed or improved.

The authors base their argument on three key factors: the doping efficiency, the polymer conformation, and the Coulomb potential landscape. While we can follow one of these three factors from the data presented (polymer conformation), we still have doubts about the other two:

1. Reasoning regarding the Coulomb potential landscape, disorder and doping efficiency:

The authors essentially argue that doping with HPy⁺ leads to a stronger overlap of the Coulomb

potentials, which should flatten the Coulomb potentials. Unfortunately, the authors were unable to alleviate our concerns in this regard for the following reasons:

1.1 Figure 2d (or R25a but also R33 a and b) show that MtBA⁺ shows on average shallower Coulomb interaction energies than HPy⁺. In combination with the really marginal differences of R_{avg} and a similar width of the distribution in Figure R26b, this contradicts all of the authors' arguments.

1.2 The cartoon in R25b (and Fig 2e) is still misleading in the light of R25a and does not correspond to the conditions shown in R25a. The authors themselves argue in several places that HPy⁺ "does not induce significantly deeper Coulomb traps", which, however, is what R25a suggests. Even more important (and misleading) is the regular, short spacing between dopant atoms in Fig. 2e bottom, that is not at all supported by the histograms in R26b.

1.3 It is unclear whether the distance to the nearest ion was considered in R26b. If this is not the case (which is at least suggested by the following considerations in the text), the calculated average counterion-counterion distance depends only on the concentration, but not on the actual docking position of the ion.

1.4 In the following, the average counterion-counterion distance is estimated by $R_{avg} = n^{-3}$, where n is the carrier concentration from AC Hall/XPS/ESR measurements. However, AC Hall, and also XPS, measures mobile carriers but not dopant concentrations and hence it is neither a factor to determine the ion distance nor a measure of doping efficiency (as claimed in the answer to Q1).

As already noted in the last review, an independent assessment of the ion densities would be helpful instead.

1.5 The comparison of a) and b) in R23 does not seem to confirm the above assumption that the dopants are closer to each other and therefore result in a more smeared Coulomb potential. In addition, the regular pattern (1 up, 1 down in pairs) is at least suspicious, suggesting we are looking at an effect of initialization rather than a realistic random incorporation.

1.6 The authors argue that the energetic disorder induced by counterions would change. They justify this, among other things, with the DOS calculated by MD (response to Q1). However, this DOS does not show any broadening, which one might expect, and the authors themselves (correctly) write elsewhere (answer to Q2.1) that the method used by the authors to consider tail states "does not account for the energetic disorder arising from Coulomb traps." which is one of the main arguments of this manuscript. Furthermore, we assume the authors have used the Urbach tail method, which is known to always produce values in the range 20-30 meV, almost irrespective of actual disorder. See <https://www.nature.com/articles/s41467-021-24202-9>.

The third problem in relation to the discussion of this DOS: it does not look mono-exponential. In consequence it matters where E_{Fermi} is sitting, most probably at higher energy where the curves for both dopants are parallel.

2. Link between simulations and experiments

2.1 We agree that the simulations show differences between the different dopants in terms of docking positions. However, as outlined above, the considered mechanism to translate it into conductivity are not convincing. Furthermore, other possible effects that would show similar experimental results are not considered and discussed:

2.1.1 The effect of screening of the ionic charge is not considered

2.1.2 It is known that, depending on the dopant, it can be located in the crystalline or amorphous region of a polymer (<https://doi.org/10.1002/adfm.202202075>) and that this can have a massive impact, at least on conductivity. The positioning of a dopant in the amorphous

regions leads only to minor changes in lattice parameters and a significant increase in conductivity at the same dopant concentration compared to a dopant located in the crystalline region. This is exactly what is seen with HPy+. Unfortunately, this is not considered here, but cannot be ruled out from our point of view.

2.2 The authors state that the MD simulations "were focused on the disordered region of the polymer. However, if the doping is visible in XRD, which is used as an argument above, it means the dopants go in the ordered regions. This diminishes the importance of the MD simulations.

3. General issues

3.1 There are still misleading Figures such as Fig. 2e as outlined above but also Fig 4i and Fig. 5: less disorder gives higher conductivity and lower thermopower, so the arrow should point down.

3.2 The argument involving AFM images still suffer from the same issue as in our last review: the AFM images look the same, depending on which cutout you look at. This would probably also apply to the c-AFM images if the color scale had not been inverted for the MtBA+ image.

Reviewer #4 (Remarks to the Author):

Point-to-Point Response

Reviewer 1:

I am very grateful for the extremely detailed response of the authors and overall I think the paper has moved closer to publication, but I cannot follow their conclusion that dipole and quadrupole moments do not correlate with experiment. I think Fig R4 explicitly demonstrates such a correlation. I would agree that other factors are also important, but I cannot follow the conclusion that the quadrupole moments do not correlate. For the dipole moments this is not so clear.

I would appreciate if Fig R4 was included and discussed in a revised manuscript.

Finally, there is something wrong with the Y-axis of Fig 5. The interaction energies are in the order of thousands of kcal/mol and this is either not normalized or not possible.

Our response: We appreciate your positive comments that “the paper has moved closer to publication”. There are two concerns: (1) the correlation between dipole and quadrupole moments and experimental data; (2) the Y-axis scale issue in Fig 5. We would like to address these concerns as follows:

1. **We have added the discussion regarding quadrupole moment in the manuscript and included Fig R4 (Fig. R2 here) to our revised Supplementary Information.** Inspired by a recent study (*Nat. Commun.* **2023**, *14*, 1356) that underscored the importance of quadrupole components in the conductivity of doped small organic molecules, our research delves into the electrostatic interactions within polymer systems. While this previous study focused on dopant molecules with negligible dipole moments due to their symmetric structure, the counterions in our system can have substantial dipole moments and vary for different counterions. We observed that both dipole and quadrupole moments of the counterions in our systems played a crucial role in the electrostatic interactions, particularly at short distances (Fig. R1). In addition, other reviewers raised concerns about the complexity of discussing numerous factors related to electrostatic interactions. Therefore,

we have chosen to treat electrostatic interactions, including monopole, dipole, and quadrupole interactions, as a unified entity in the manuscript. This emphasizes their collective impact on counterion docking and conductivity. We have also ensured that the quadrupole moment is discussed in the manuscript and enhances understanding without overwhelming the reader with excessive detail. We appreciate your suggestion to include and discuss Fig. R4 (Fig. R2 here), and we have incorporated it into the revised Supplementary Information.

Fig. R1. **a**, The distance-dependent V_c for P(PzDPP-2FT) with MtBA⁺ and HPy⁺. **b**, the monopole term (charge-charge interaction), **c**, the dipole term (dipole-charge interaction), and **d**, the quadrupole term (quadrupole-charge interaction) in the multipole expansion of V_c . The polymer-counterion distance (r) was measured between the geometry center of a counterion and the geometric center of the adjacent polymer backbone with two repeating units.

Fig. R2. The relationship between **a**, dipole and **b**, quadrupole moments with electrical conductivity.

The specific revisions are detailed as follows:

1) In the manuscript, we have added the discussion on quadrupole moments:

177 In addition to backbone conformation, energetic disorder in doped polymer systems
 178 also arises from Coulomb traps, which is strongly associated with the counterion
 179 distributions. A recent study used distance-dependent Coulomb interaction energy (V_C) to
 180 quantify such effect and prove the importance of quadrupole components in the
 181 conductivity of doped small organic molecules³⁴. Unlike the symmetric dopant molecules
 182 used in the study, which have negligible dipole moments, our system involves counterions
 183 with significant and variable dipole moments (Fig. S17). We observed that both the dipole
 184 and quadrupole moments of counterions played a crucial role in the electrostatic
 185 interactions (Fig. S19). To better understand the electrostatic interactions, we have
 186 redefined V_C to include the contributions from monopole, dipole, and quadrupole
 187 interactions, assessing their combined effects on the system. As shown in Fig. 2e, V_C
 188 increases with decreasing polymer-counterion distances (r). However, when $r < 5 \text{ \AA}$ ($r^{-1} >$
 189 0.2 \AA^{-1}), V_C levels off to a flat electrostatic potential due to the screening effect of the ionic
 190 charges^{34,35}. Although HPy^+ docked closer to the polymer backbone, it does not induce

2) In the Supplementary Information, Fig. R2 (Fig. R4 in the previous response letter) has been included as Fig. S20, which demonstrates the correlation between dipole and quadrupole moments with experimental conductivity data, complemented by a descriptive analysis.

2. The Y-axis in Fig. R5 (also Fig. R3 here) represents the sum of interaction energies between all counterions and polymers within a supercell. In our MD simulations, we built a supercell made of 3 layers of 8 π -stacked dodecamers of polymers, along with 72 counterions. According to DFT calculations, the interaction energies between two repeating units of the polymer chain and a single counterion are approximately 70 kcal/mol. Let's simplify by assuming that each counterion interacts with a dimer of the polymer chain within the supercell. Thus, the total interaction energies between counterions and polymers in the supercell can be estimated to be around 72×70 kcal/mol, totaling, 5.04×10^3 kcal/mol. This value aligns with the static value depicted in Fig.R5 (also Fig. R3 here), confirming the rationality of the Y-axis.

Fig. R3. **a**, Energy decomposition analysis (EDA) based on the force field for the interaction between P(PzDPP-2FT) and counterions. **b**, Quantitative analysis depicting the contribution of electrostatic or van der Waals interaction to the overall NCIs between P(PzDPP-2FT) and counterions. **c**, EDA for the interactions between the polymer backbone and counterions. **d**, EDA for the interactions between the polymer sidechain and counterions.

We appreciate your valuable suggestions. To address any potential confusion, we have expanded upon the description provided in Fig. 2b of the manuscript and included a

detailed discussion in Fig. S8 of the Supplementary Information (Fig. R3 here). The specific revisions in the manuscript are highlighted as follows:

148

149 **Fig. 2 | Understanding of the counterion effects on the energetic disorder of the doped**
 150 **polymers.** **a**, MD simulation snapshot of the polymer P(PzDPP-2FT) with MtBA⁺ and
 151 HPy⁺ (bottom) as the counterions. The polymer backbone in the front is highlighted for
 152 better contrast. **b**, Energy decomposition analysis (EDA) based on the force field. The
 153 electrostatic and van der Waals interactions constitute the total NCIs between 24 polymer
 154 chains and 72 counterions within the supercell. Negative values indicate attractive
 155 interactions between polymers and counterions. **c**, FWHM of the torsion angle distributions
 156 for the dihedral angles DPP-Pz and Pz-2FT. **d**, Orbital LL of the undoped and counterion-
 157 exchanged P(PzDPP-2FT). **e**, Coulombic interaction energies estimated for the polymer-
 158 counterion pairs, obtained from MD simulations.

Reviewer 2:

The authors have properly addressed all my comments.

I am satisfied with the authors toning down on the nature of electronic carriers.

I believe that the ambiguous discussion has been removed from the main context, and the gist of this research has become clearer.

Although important discussions have moved to SI, the main text still presents high-quality experimental results.

This is an excellent work that deserves publication.

Our response: We are pleased to see that you are satisfied with our revisions and think our work deserves publication. We appreciate your time and effort in reviewing our manuscript. Your positive feedback encourages us.

Reviewer 3:

The authors have come up with an extensive response. However, they have still not been able to convince us in some key areas. This essentially concerns two points:

- The argumentation regarding the Coulomb potential landscape, disorder, and doping efficiency
- The link between simulations and experiments

In general, we could accept the manuscript if it is revised with a focus on the experiments and the position of the ions with respect to the backbone (where significant differences show up). The discussion of the direct connection to the conductivity would then need to be shortened and leave room for other effects such as screening of the ionic charge or the positioning of dopants in different regions of the polymer (amorphous or crystalline). In addition, two misleading cartoons must be removed or improved.

Our response: Thank you for your constructive feedback and advice on the manuscript revision. Your concerns have been carefully considered, and we have revised our manuscript to address the points you raised. Below are the key revisions to your comments:

1. Coulomb potential landscape, doping efficiency, and disorder:

- (1) We have enhanced our analysis with high-temperature molecular dynamics (MD) simulations, which have been included in the Supplementary Information. We observed that MtBA⁺ showed a multimodal and broader distribution, with a larger counterion-counterion distance, while HPy⁺ maintains a narrower distribution and a smaller counterion-counterion distance. We agree that the Coulomb potential landscape in Fig. 2d could not reflect the overlap of Coulomb traps in Fig. 2e. In our revised manuscript, we have removed the cartoon in Fig. 2e for better precision and improved the clarity for the discussion of the Coulomb potential landscape.
- (2) X-ray photoelectron spectroscopy (XPS) data estimating the counterion density within the polymer has been integrated into the Supplementary Information. The polymer exchange-doped with HPy⁺ exhibits an estimated counterion concentration of $7.66 \times 10^{19} \text{ cm}^{-3}$, which is about twice that of the polymer exchange-doped with MtBA⁺ ($3.70 \times 10^{19} \text{ cm}^{-3}$). We agree that the counterion-counterion distance is largely influenced by their concentration. We have demonstrated that the counterion docking positions also play a

critical role in the counterion-counterion distance and therefore the Coulomb potential landscape.

- (3) We have acknowledged the limitations of the DOS function in Fig. 2c and have removed it to avoid any misrepresentation of energetic disorder. Instead, we have focused on the torsion angle distributions of the polymer backbone and orbital localization length as indicators of polymer conformation disorder.

2. The link between simulations and experiments:

- (1) We acknowledge the importance of the ionic screening effect on Coulomb potential interactions, as reported in recent literature (*Phys. Rev. Lett.* **2023**, *131*, 248101). Although the previous study and our work employed different methods, our calculations of the Coulomb interaction corroborate this effect, showing a decrease in the Coulomb potential landscape at short polymer-counterion distances. In the revised manuscript, we have attributed the flat of Coulomb interactions at short polymer-counterion distance to the ionic screening effect.
- (2) At a high doping level, we have observed a significant increase in the lamellar spacing of the polymer with HPy⁺, indicating that HPy⁺ has docked into the crystalline region. Additionally, HPy⁺ exhibits stronger NCIs with the polymer backbone compared to MtBA⁺, suggesting that HPy⁺ is more likely to dock into the crystalline region. The polymer exchange-doped with HPy⁺ consistently displayed lower activation energies, indicating that HPy⁺ may cause less disruption to the molecular packing. In our revised manuscript, we have demonstrated that HPy⁺ can dock into the crystalline region, not just distributed in the amorphous region.
- (3) We have clarified the term "disordered region" in the manuscript to refer to the crystalline region with increased lamellar spacing and structural disorder, rather than amorphous regions. In the amorphous region, intrachain charge transport is relatively fast, and the primary limitation to charge transport is the intermolecular packing disorder found in the crystalline/disordered regions (*Nat. Mater.* **2020**, *19*, 491–502). Our GIWAXS data demonstrated that counterions can impact the molecular packing within the crystalline region. Therefore, the investigation of the counterion effects on the disordered region may hold greater significance than the amorphous region in

charge transport.

3. Other issues:

- (1) The cartoon in Fig. 2e, which is potentially misleading, has been removed. Additionally, the schematic representation of thermoelectric performance in Fig. 5 has been replaced with actual experimental data from Fig. 4i for enhanced clarity and accuracy.
- (2) We have conducted a new series of AFM scans over a larger range and randomly selected regions to ensure the validity of our AFM images. We did not observe any significant changes in the morphology of the undoped polymer films or those exchange-doped with the three counterions, indicating that our AFM images are free from cutout issues. Furthermore, we have updated c-AFM images with the same color bar range of 0 pA to 200 pA to allow for a more accurate comparison of the electrical current distribution across the different samples. When exchange-doped with HPy⁺, the surface of the polymer film exhibited large connected areas with high currents. In contrast, when exchange-doped with MtBA⁺, the film showed isolated conducting areas with low currents.

We believe that these revisions have effectively addressed your concerns and provided a clearer and more comprehensive understanding of our research. We are grateful for the opportunity to improve our manuscript and hope that it now meets the standards for publication.

The authors base their argument on three key factors: the doping efficiency, the polymer conformation, and the Coulomb potential landscape. While we can follow one of these three factors from the data presented (polymer conformation), we still have doubts about the other two:

Q1. Reasoning regarding the Coulomb potential landscape, disorder, and doping efficiency:

The authors essentially argue that doping with HPy⁺ leads to a stronger overlap of the Coulomb potentials, which should flatten the Coulomb potentials. Unfortunately, the authors were unable to alleviate our concerns in this regard for the following reasons:

Our response: Thank you for your advice. We understand the importance of providing a clearer discussion of the Coulomb potential landscape. Here is our refined response from three aspects to address your concerns:

- 1. Difference in counterion distribution.** In our previous manuscript, we observed that HPy⁺ was located near the polymer backbone, while MtBA⁺ exhibited a more chaotic distribution. Utilizing high-temperature molecular dynamic (MD) simulations to accelerate the counterion diffusion process, the distinct docking behaviors of HPy⁺ and MtBA⁺ become particularly evident. The statistical analysis of counterion-counterion distance distribution further confirmed this observation, with MtBA⁺ showing a multimodal and broadened distribution and a significantly increased average distance R_{avg} , while HPy⁺ maintains a narrower distribution and a smaller R_{avg} . The high-temperature simulations accelerated the counterion diffusion and demonstrated the difference between MtBA⁺ and HPy⁺.
- 2. Counterion density and docking position.** By analyzing X-ray photoelectron spectroscopy (XPS), we estimated the counterion density within the exchange-doped polymer. The polymer exchange-doped with HPy⁺ exhibits an estimated counterion concentration of $7.66 \times 10^{19} \text{ cm}^{-3}$, which is about twice that of the polymer exchange-doped with MtBA⁺ ($3.70 \times 10^{19} \text{ cm}^{-3}$). This higher counterion density of HPy⁺ is expected to enhance the overlap of Coulomb traps, thereby affecting the overall potential landscape. Moreover, we have demonstrated that the counterion docking position can significantly influence the counterion-counterion distance. The counterion-counterion distance is defined as the distance between the geometry centers of the nearest counterions. While the initial position of both counterions is identical, HPy⁺ can stably dock around the polymer backbone, exhibiting a shorter average counterion-counterion distance compared to MtBA⁺. This more uniform distribution, coupled with the higher counterion density, leads to a stronger overlap of Coulomb traps. In addition, both counterion density and docking position are dependent on the non-covalent interactions (NCIs) between the polymer and counterions. Stronger electrostatic interactions between the polymer and counterions not only contribute to higher exchange-doping efficiency but also result in the localization of the counterion near the polymer backbone. It is unreasonable to discuss only one of them. In the revised manuscript, we have emphasized the effect of NCIs on both doping efficiency and counterion distribution.
- 3. Enhancement of clarity and precision.** (1) We have removed the cartoon from Fig. 2e to prevent any potential confusion; (2) Recognizing the limitations of fitting the DOS tail, we

have removed the DOS function in Fig. 2c.

We hope that these revisions can address your concerns and provide a more accurate description of Coulomb potential landscape.

Q1.1. Figure 2d (or R25a but also R33 a and b) show that MtBA⁺ shows on average shallower Coulomb interaction energies than HPy⁺. In combination with the really marginal differences of R_{avg} and a similar width of the distribution in Figure R26b, this contradicts all of the authors' arguments.

Our response: Thank you for your advice. The detailed response to your two concerns is as follows:

1. **Coulomb interaction energies.** We agree that Fig. 2d (also Fig. R4b here) suggests that MtBA⁺ has slightly shallower average Coulomb interaction energies than HPy⁺ at equivalent distances from the polymers. However, both dipole and quadrupole interactions contribute an up-shift of the Coulomb interactions at short distances (Fig. R4b and R4c). The dipole and quadrupole components highlighted the importance of charge distribution on counterions (*Nat. Commun.* **2023**, *14*, 1356; *Nat. Mater.* **2019**, *18*, 242-248). When the dipole and quadrupole interactions were also included, the V_C of HPy⁺ was nearly the same as that of MtBA⁺ at short distances (Fig. R4a). In addition, we would like to emphasize that the comparison between MtBA⁺ and HPy⁺ is not solely about the average interaction energies but also about the overall potential landscape. Our analysis reveals that when the polymer-counterion distance r falls below 5 Å, V_C for HPy⁺ levels off to a flat electrostatic potential (Fig. R4a). For instance, the V_C of HPy⁺ at $r = 2.5$ Å ($r^{-1} = 0.4$ Å⁻¹) is approximately -0.65 eV, which is comparable to the V_C of MtBA⁺ at distances ranging from 5 to 6.7 Å. All these results indicate that despite HPy⁺ being closer to the polymer backbone, it does not create significantly deeper Coulomb traps compared to MtBA⁺.

Fig. R4. **a**, The distance-dependent V_C for P(PzDPP-2FT) with MtBA⁺ and HPy⁺. **b**, the monopole term (charge-charge interaction), **c**, the dipole term (dipole-charge interaction), and **d**, the quadrupole term (quadrupole-charge interaction) in the multipole expansion of V_C . The polymer-counterion distance (r) was measured between the geometry center of a counterion and the geometric center of the adjacent polymer backbone with two repeating units.

- Counterion distribution.** With MD simulations at high temperature, we demonstrated the difference in counterion distribution between MtBA⁺ and HPy⁺. Our previous dynamic simulations at 298 K for 600 ps show that MtBA⁺ has a wider distribution of distances to the polymer backbone than HPy⁺ (Fig. R5a). Consequently, the average counterion-counterion distance (R_{avg}) for MtBA⁺ is greater than that for HPy⁺ (Fig. R5b). Note that while our simulation time is confined to the picosecond scale, the actual timescale of counterion diffusion in the real-world is much longer.

Fig. R5. **a**, Statistical analysis of the polymer-counterion distance (r), defined as the distance between the geometric average line of the polymer backbone and the geometry center of a counterion. **b**, Statistical analysis of the counterion-counterion distance (R), representing the distance between the geometry centers of the nearest counterions.

To facilitate the diffusion of counterions, a series of high-temperature MD simulations were conducted. The process began with a constant temperature and pressure (NPT) simulation, where the system was maintained at 1000 K under a pressure of 1 atmosphere for 200 ps. Following this, two additional 200-ps dynamic simulations were executed at sequentially reduced temperatures of 600 K and 300 K (Fig. R6 and Fig. R7). As the temperature increased to 1000 K, the polymer backbone, which initially had a zig-zag backbone conformation, straightened to a near-linear structure (Fig. R6b and Fig. R7b). Concurrently, the distribution of counterions became broader (Fig. R8 and R9), indicating that the molecules within the systems were fully relaxed. Throughout the heating-to-cooling process, HPy^+ was observed to consistently locate near the polymer backbone, while MtBA^+ exhibited a more chaotic distribution (Fig. R6 and Fig. R7). Statistical analysis further confirmed that the polymer-counterion distribution of HPy^+ kept almost unchanged throughout the process, whereas the distribution for MtBA^+ broadened significantly (Fig. R8). This chaotic distribution of MtBA^+ within the polymer led to a more dispersed and multimodal counterion-counterion distance distribution, with the average distance R_{avg} increasing markedly from 8.96 to 13.50 Å (Fig. R9). In contrast, the counterion-counterion distribution of HPy^+ remained consistent after this process, with only a slight expansion in the average distance R_{avg} to 10.36 Å. These results suggested

that the stronger electrostatic interactions effectively stabilize HPy^+ around the polymer backbone, leading to a more ordered distribution. This stable position may also contribute to a more pronounced overlap of Coulombic traps, which could influence the overall electrostatic landscape within the polymer system. The high-temperature simulations accelerated the counterion diffusion and demonstrated the difference between MtBA^+ and HPy^+ .

Fig. R6. MD simulations of MtBA^+ in P(PzDPP-2FT) supercell for 200 ps at different temperatures: **a**, 298 K, **b**, 1000 K, **c**, 600 K, and **d**, 300 K (annealed).

Fig. R7. MD simulations of HPy⁺ in P(PzDPP-2FT) supercell for 200 ps at different temperatures: **a**, 298 K, **b**, 1000 K, **c**, 600 K, and **d**, 300 K (annealed).

Fig. R8. Statistical analysis of the polymer-counterion distance r of **a**, MtBA⁺ and **b**, HPy⁺. The distance r is defined as the distance between the geometry center of a counterion and the geometric center of the nearest polymer backbone.

Fig. R9. Statistical analysis of the counterion-counterion distance R of **a**, MtBA⁺ and **b**, HPy⁺. The distance R is defined as the distance between the geometry centers of the nearest counterions.

In summary, these findings underscore that despite HPy⁺ being closer to the polymer backbone, it does not create significantly deeper Coulomb traps compared to MtBA⁺. In addition, we have conducted MD simulations at high temperature to accelerate the counterion diffusion, which highlighted the difference between MtBA⁺ and HPy⁺. We demonstrated that HPy⁺ was consistently located near the polymer backbone, while MtBA⁺ exhibited a more chaotic distribution. As a result, the average distance R_{avg} of MtBA⁺ increases markedly to 13.50 Å, while HPy⁺ exhibits only a slight expansion in R_{avg} to 10.36 Å. The high-temperature MD simulations have been added to Section 3 of the Supplementary Information and Fig. R6-R9 have been included as Fig. S8-11.

Q1.2. The cartoon in R25b (and Fig 2e) is still misleading in the light of R25a and does not correspond to the conditions shown in R25a. The authors themselves argue in several places that HPy⁺ “does not induce significantly deeper Coulomb traps”, which, however, is what R25a suggests. Even more important (and misleading) is the regular, short spacing between dopant atoms in Fig. 2e bottom, that is not at all supported by the histograms in R26b.

Our response: Thank you for your advice. We agree that the cartoon in Fig. 2e might be misleading and not fully representative of the data presented in Fig. 2d of the manuscript. In the revised manuscript, we have removed the cartoon in Fig. 2e:

Q1.3. It is unclear whether the distance to the nearest ion was considered in R26b. If this is not the case (which is at least suggested by the following considerations in the text), the calculated average counterion-counterion distance depends only on the concentration, but not on the actual docking position of the ion.

Our response: Thank you for raising this important point. The counterion-counterion distance in our study was indeed determined by measuring the distance of the geometry centers of the nearest counterions, ensuring that the analysis accounts for the docking positions of the

counterions. We have clarified this in detail in the manuscript.

We agree that the average distance between counterions is largely influenced by their concentration. However, in this work, we would like to point out that the docking positions of the counterions are also critical. We have included a schematic illustration (Fig. R10) that demonstrates the effect of NCIs on counterion density and distribution. It highlights the interplay between counterion density and docking position, which are both influenced by the NCIs between the polymer and the counterions. The stronger non-covalent interactions (NCIs) between the polymer backbone and HPy⁺ can result in a higher concentration and, consequently, a higher doping level, as supported by our experimental findings (response to Q1.4). However, the NCIs are not only significant for the concentration but also for the docking positions of the counterions. To investigate the influence of the docking position, we have conducted MD simulations with an equal amount of HPy⁺ and MtBA⁺ in the polymer supercell. Our simulations revealed that the stronger electrostatic interactions effectively stabilize HPy⁺ around the polymer backbone, leading to a more ordered distribution and a smaller average counterion-counterion distance. In contrast, MtBA⁺ exhibited a more chaotic distribution and a larger average counterion-counterion distance. These results indicate that the docking position also influences the counterion-counterion distance.

Fig. R10. Schematic illustration of the effect of NCIs on counterion density and counterion distribution.

In the revised manuscript, we have discussed the role of NCIs in enhancing doping efficiency and have provided a comparative analysis of the counterion docking positions on counterion-counterion distance. We agree that considering both effects is more accurate to describe the counterion effects in doped polymer systems. The detailed revision is as follows:

193 with the distance between these traps^{10,36}. The stronger NCIs between the polymer
 194 backbone and HPy⁺ may contribute to a higher counterion density (we will experimentally
 195 prove this later) and consequently a stronger overlap of the Coulomb traps^{15,37}. In addition,
 196 the counterion docking position also plays a role in the Coulomb potential landscape. The
 197 scattered distribution of MtBA⁺ within the polymer leads to a larger average ion-ion
 198 distance than those with HPy⁺ (Fig. S7 and Fig. S11), which could generate isolated deep
 199 Coulomb traps and thus higher barrier heights for carrier transport. In contrast, the shorter
 200 ion-ion distance of HPy⁺ can facilitate a strong overlap of the Coulomb traps at high doping
 201 concentrations, reducing activation energies for carrier transport. By strategically docking
 202 counterions to suitable positions, we might be able to fine-tune the overall energetic
 203 disorder and optimize the charge transport performance in doped polymeric
 204 semiconductors.↵

Q1.4. In the following, the average counterion-counterion distance is estimated by $R_{avg} = n^{-3}$, where n is the carrier concentration from AC Hall/XPS/ESR measurements. However, AC Hall, and also XPS, measures mobile carriers but not dopant concentrations and hence it is neither a factor to determine the ion distance nor a measure of doping efficiency (as claimed in the answer to Q1).

As already noted in the last review, an independent assessment of the ion densities would be helpful instead.

Our response: Thank you for your advice. It is important to note that among the three methods, only AC Hall measures the mobile carriers. EPR is effective for detecting unpaired electrons, while it does not detect counterions, which are EPR silent. Consequently, neither AC Hall nor EPR can directly measure ion densities. XPS, however, may offer a more suitable approach to estimating counterion density, as it can reflect the molar ratio of atoms within counterions. We have utilized XPS to analyze and characterize the atomic species of counterions and to calculate the counterion density.

We first excluded the interference of the residual TDAE dopants. Our Fourier-transform infrared spectroscopy (FTIR) analysis indicated a significant reduction in the characteristic peaks of TDAE ($1600\text{-}1650\text{ cm}^{-1}$) after ion-exchanging (Fig. R11), suggesting a near-complete exchange of TDAE^- by cations from the ion liquids.

Fig. R11. Fourier-transform infrared spectroscopy (FTIR) of the pristine P(PzDPP-2FT), TDAE-doped P(PzDPP-2FT), and P(PzDPP-2FT) exchanged with BMIM⁺ and MtBA⁺.

After ion-exchange doping, four distinct nitrogen (N) environments were observed in the polymer films, as depicted in Fig.R12a: N from the DPP moiety, N from the pyrazine, N⁺ from cations, and N⁻ from TFSI⁻. The N (1s) XPS data for the pure ion liquid EPy⁺-TFSI⁻ revealed distinct N⁺ and N⁻ peaks at 402.2 eV and 399.1 eV, respectively (Fig. R12b). In contrast, the neutral N from the polymer backbone exhibited double peaks at 400.2 eV and 399.4 eV (Fig. R12c), allowing for a clear distinction between the neutral N from the polymer and N⁺/ N⁻ from ions. Fig. R12d-f illustrate the XPS N (1s) spectra of P(PzDPP-2FT) exchange-doped with the three counterions. The emergence of two new peaks at approximately 402 eV and 398 eV corresponds to N⁺ of the cations and N⁻ from TFSI⁻, respectively. The slight shift of 0.2 eV in the neutral N from the undoped polymer to lower binding energy after n-doping indicates the formation of new negatively charged N species during the doping process. This shift, however, is minimal compared to the peaks associated with cations and anions and does not impact the cation/polymer ratio. For simplicity, we make no distinctions between the neutral N

and charged N species of the polymer and take them as a unity. Based on the above analysis, the counterion concentration $n_{counterion}$ within the polymer can be estimated using the following formula:

$$n_{counterion} = \frac{S_{cation} - S_{TFSI^-}}{S_{poly}} n_{poly}$$

Here, n_{poly} is the number of N atoms per polymer monomer, while S_{cation} , S_{TFSI^-} , and S_{poly} correspond to the signals from cations, anion TFSI⁻, and the polymer, respectively. It is important to note that the cation concentration calculation for BMIM⁺ should account for its two N atoms per cation, thus dividing S_{cation} by 2. The estimated counterion concentrations are summarized in Table R1. The polymer exchange-doped with HPy⁺ exhibits an estimated counterion concentration of $7.66 \times 10^{19} \text{ cm}^{-3}$, which is twice that of the polymer exchange-doped with MtBA⁺ ($3.70 \times 10^{19} \text{ cm}^{-3}$). As a result, the estimated average counterion-counterion distance ($R_{counterion} = n_{counterion}^{-1/3}$) of HPy⁺ is 23.5 Å, while MtBA⁺ exhibits a larger distance of 30.0 Å.

Fig. R12. **a**, Chemical structure of the polymer P(PzDPP-2FT), the dopant TDAE, the three cations (MtBA⁺, BMIM⁺, HPy⁺), and the anion TFSI⁻. **b**, N(1s) XPS spectra and fitted curves of **c**, the undoped P(PzDPP-2FT) and P(PzDPP-2FT) exchange-doped with **d**, MtBA⁺, **e**, BMIM⁺, **f**, HPy⁺.

Table R1. Counterion concentrations $n_{counterion}$ estimated from XPS analysis. And the estimated average counterion-counterion distance ($R_{counterion} = n_{counterion}^{-1/3}$).

		⁻³⁾	
P(PzDPP-2FT)	MtBA ⁺	3.70×10 ¹⁹	30.0
	BMIM ⁺	4.18×10 ¹⁹	28.8
	HPy ⁺	7.66×10 ¹⁹	23.5

These results confirmed the higher counterion density and shorter counterion-counterion distance of HPy⁺, which could contribute to a stronger overlap of Coulomb traps. In the revised Supplementary Information, we have included the XPS analysis of Fig. R12 as Fig. S37 and added Table R1 as Table S5.

Q1.5. The comparison of a) and b) in R23 does not seem to confirm the above assumption that the dopants are closer to each other and therefore result in a more smeared Coulomb potential. In addition, the regular pattern (1 up, 1 down in pairs) is at least suspicious, suggesting we are looking at an effect of initialization rather than a realistic random incorporation.

Our response: Thank you for your advice. We acknowledge the difficulty in directly observing the distribution of the counterion-counterion distance from a single snapshot of MD simulations. Therefore, we performed the statistical analysis of the counterion distribution over time. As mentioned in our response to Q1.1, we observed that the distribution of MtBA⁺ became more disordered, leading to an average counterion-counterion distance that is significantly larger compared to that of HPy⁺. This suggests that while the initial position of both counterions is identical, the counterion diffusion over time is more indicative of their interactions with the polymer.

Regarding the initial position of counterions, we choose to place them in an alternating pattern (1 up and 1 down in pairs) around the polymer backbone, which ensured the balance of charge distribution and prevent bending in the polymer backbone due to directional forces (Fig. S5 also Fig. R13 here). We agree that in a realistic scenario, counterions would dock randomly within the polymer films. However, it is important to note that the time scale of MD simulations presents challenges when trying to capture the complex dynamics of counterion docking in such complex systems. Considering the multiple influencing factors that can influence the counterion docking process, including NCIs, polymer conformations, energetic

disorder, and others, it was necessary to standardize the counterion positions to enable a meaningful comparison of their diffusion behaviors. By doing so, we were able to observe significant differences in the diffusion patterns of MtBA^+ and HPy^+ within the 1 ns time scale of our MD simulations, despite the identical initial positions.

Fig. R13. The unit cell of P(PzDPP-2FT) with **a**, MtBA^+ , **b**, HPy^+ , which were rendered as purple, and red transparent ellipsoids, respectively.

Q1.6. The authors argue that the energetic disorder induced by counterions would change. They justify this, among other things, with the DOS calculated by MD (response to Q1). However, this DOS does not show any broadening, which one might expect, and the authors themselves (correctly) write elsewhere (answer to Q2.1) that the method used by the authors to consider tail states "does not account for the energetic disorder arising from Coulomb traps." which is one of the main arguments of this manuscript. Furthermore, we assume the authors have used the Urbach tail method, which is known to always produce values in the range 20-30 meV, almost irrespective of actual disorder. See <https://www.nature.com/articles/s41467-021-24202-9>.

The third problem in relation to the discussion of this DOS: it does not look mono-exponential. In consequence it matters where E_{Fermi} is sitting, most probably at higher energy where the curves for both dopants are parallel.

Our response: Thank you for your insightful comments. We agree that the broadening of the DOS tail is not pronounced from Fig. 2c and the fitting to the DOS tail may not accurately represent the energetic disorder. Additionally, while we have measured the shift of Fermi level in the exchange-doped polymers, accurately determining its position can be quite challenging.

Therefore, we have revised the manuscript to remove the DOS from Fig. 2d and the corresponding discussion. We have also included the torsion angle distribution of the polymer backbones, which was previously presented as Fig. S11 in the Supplementary Information, now featured as Fig. 2d in the manuscript. The calculation of the orbital localization length, originally shown in Fig. 2d, has been maintained as Fig. 2e in the revised manuscript. We believe that the torsion angle distribution and the orbital localization length provide a clearer representation of the counterion effect on the conformational disorder. Here is the detailed revision of the manuscript:

148

149 **Fig. 2| Understanding of the counterion effects on the energetic disorder of the doped**
 150 **polymers.** **a,** MD simulation snapshot of the polymer P(PzDPP-2FT) with MtBA⁺ and
 151 HPy⁺ (bottom) as the counterions. The polymer backbone in the front is highlighted for
 152 better contrast. **b,** Energy decomposition analysis (EDA) based on the force field. The
 153 electrostatic and van der Waals interactions constitute the total NCIs between 24 polymer
 154 chains and 72 counterions within the supercell. Negative values indicate attractive
 155 interactions between polymers and counterions. **c,** FWHM of the torsion angle distributions
 156 for the dihedral angles DPP-Pz and Pz-2FT. **d,** Orbital LL of the undoped and counterion-
 157 exchanged P(PzDPP-2FT). **e,** Coulombic interaction energies estimated for the polymer-
 158 counterion pairs, obtained from MD simulations.

162 backbone surrounded by HPy⁺ maintained its zig-zag conformation (Fig. 2a)¹⁸. We further
163 analyzed the torsion angle distributions of the two dihedral angles: DPP-Pz and Pz-2FT
164 (Fig. 2c). The results revealed a notable difference for the two counterions: the polymer
165 with MtBA⁺ showed a significantly broader torsion angle distributions for both dihedral
166 angles, with full width at half maximum (FWHM) recorded at 79.2° for DPP-Pz and 49.3°
167 for Pz-DPP. In contrast, the polymer with HPy⁺ displayed much narrower torsion angle
168 distributions: 43.9° for DPP-Pz and 33.6° for Pz-DPP. These values are even smaller than
169 those of the undoped polymer, suggesting an enhancement in backbone planarity in HPy⁺
170 exchange-doped film. The polymer conformation disorder directly influences the
171 electronic structure and charge transport properties. We quantify this effect by calculating
172 the orbital localization length (*LL*)^{32,33} (Fig. 2d and Table S3). We observed that the HPy⁺-
173 exchanged polymer showed an extended *LL* at the band edge, reaching approximately 40
174 Å, which is significantly larger than that of the undoped polymer (35 Å) and the polymer
175 with MtBA⁺ (32 Å). These results suggest a reduced energetic disorder in the polymer
176 conformation exchanged with HPy⁺.[←]

Q2. Link between simulations and experiments

Q2.1. We agree that the simulations show differences between the different dopants in terms of docking positions. However, as outlined above, the considered mechanism to translate it into conductivity are not convincing. Furthermore, other possible effects that would show similar experimental results are not considered and discussed:

Q2.1.1. The effect of screening of the ionic charge is not considered

Our response: Thank you for your valuable advice. We have carefully considered the ionic screening effect based on recent research (*Phys. Rev. Lett.* **2023**, *131*, 248101), which shed light on how ionic screening can modulate the electrostatic potential of dopants, thereby influencing charge transport properties. In our work, we have incorporated the impact of dopant counterions' electrostatic potential on polymer conformation, doping efficiency, and charge transport performance. We align with the findings of the aforementioned study, highlighting the importance of the electrostatic potential of dopant counterions in doped conjugated polymer systems.

However, we recognize the complexity in pinpointing the exact parameters within the modified Gaussian disorder model (GDM), specifically:

1. The distance (R_s) between the polymer and the counterions. As previously discussed in response to *Q1.4*, the distance between the polymer and counterions is not a fixed value but varies widely, ranging from 0 to 15 Å. This variability presents a challenge in applying a static model to a dynamic system.
2. The standard deviation (R_d) of Gaussian charge distribution. As mentioned in our last response, the electron density of counterions exhibits a strong correlation with the molecular conformation. Therefore, the spread of the charge R_d could fluctuate significantly with ion conformation changes.
3. When fitting the Seebeck coefficient versus conductivity curve using this modified GDM, the polymer-counterion distance and energetic disorder are treated as constants, which do not accurately reflect the system behavior at varying carrier concentrations.

While the inclusion of ionic screening in our current study presents certain complexities, we agree that it is a crucial factor, especially in heavily doped polymer systems. In our future research, we will further explore the counterion effect by considering the ionic screening, providing a deeper understanding of its impact on the overall conductivity and charge transport in doped conjugated polymers.

Q2.1.2. It is known that, depending on the dopant, it can be located in the crystalline or amorphous region of a polymer (<https://doi.org/10.1002/adfm.202202075>) and that this can have a massive impact, at least on conductivity. The positioning of a dopant in the amorphous regions leads only to minor changes in lattice parameters and a significant increase in conductivity at the same dopant concentration compared to a dopant located in the crystalline region. This is exactly what is seen with HPy⁺. Unfortunately, this is not considered here, but cannot be ruled out from our point of view.

Our response: Thank you for your advice. We agree that dopants can locate in the crystalline and amorphous region of the polymer film. In our study, we observed an increase in the lamellar distance of the polymer upon doping with HPy⁺, which suggests that HPy⁺ has docked into the crystalline region. Additionally, the counterion docking behavior is driven by NCIs. HPy⁺ exhibits stronger NCIs with the polymer backbone compared to MtBA⁺, suggesting that HPy⁺ is more likely to dock into the crystalline region. We have also quantified the activation energies

for the polymer when exchange-doped with MtBA⁺ and HPy⁺ at various doping levels. The polymer exchange-doped with HPy⁺ consistently displayed lower activation energies, indicating that HPy⁺ may cause less disruption to the molecular packing. The detailed discussion is as follows:

1. **Heterogenous doping:** Dopants are more likely to initially occupy the amorphous regions due to the difficulty in entering the ordered stacking crystalline region (*Chem. Rev.* **2023**, *123*, 7421–7497; *Adv. Funct. Mater.* **2022**, *32*, 2202075). Previous studies have shown that the π - π /lamellar stacking distance of conjugated polymer remains relatively constant at low doping levels but changes significantly at high doping levels. (*Nat. Commun.* **2020**, *11*, 3292; *Adv. Mater.* **2022**, *34*, 2102988). This is ascribed to the heterogeneous doping that involves doping of the amorphous domain first before doping of the crystalline domains (*Adv. Energy Mater.* **2023**, *13*, 2202797). Consistent with these findings, we also observed an expansion in the lamellar stacking distances after exchange-doped with the three counterions (MtBA⁺, BMIM⁺, and HPy⁺) for 5 minutes, indicating their existence in crystalline regions (Fig. R16). Increasing the doping time to 30 minutes, the lattice expansion further increased, suggesting that more counterions docked into the crystalline regions. While the π - π distance for the polymer with HPy⁺ only slightly increased by 0.02 Å at 30 minutes (Fig. R16b), the lamellar stacking distance significantly increased by 2.56 Å (Fig. R16a), confirming the docking of HPy⁺ into the crystalline region without significantly disrupting the molecular packing.

Fig. R16. Variations in **a**, lamellar packing distances and **b**, π - π stacking distances of the exchange-doped P(PzDPP-2FT) at 5 and 30 minutes compared to the pristine polymer film. Data were obtained from the GIWAXS measurements.

2. **Doping efficiency.** In our systems, we found that the aromatic cation HPy^+ exhibited stronger interactions with the polymer backbone than the alkyl cation MtBA^+ . This suggests that HPy^+ should more easily dock into the crystalline region due to enhanced NCIs. If HPy^+ were only exchanging with dopants in the amorphous region, while MtBA^+ could exchange with dopants in both regions, the doping efficiency of the polymer with MtBA^+ would be expected to surpass that of HPy^+ . However, the polymer exchange-doped with HPy^+ exhibited a higher doping level, as evidenced by our UPS, AC Hall, EPR, and XPS measurements. Therefore, both dopants can dope crystalline and amorphous regions.
3. **Energetic disorder.** We have measured the activation energies of the polymer films exchanged with MtBA^+ and HPy^+ at different doping levels (Fig. R17). From low to high doping levels, the activation energies of the polymer with HPy^+ are consistently lower than that of the polymer with MtBA^+ , indicating that the energetic disorder from introducing HPy^+ counterions is smaller than MtBA^+ . This result aligns with the higher doping efficiency and minimal disruption to the molecular packing associated with HPy^+ .

Fig. R17. Activation energies of the polymer films exchanged with MtBA^+ and HPy^+ at different doping levels.

In summary, our findings demonstrate that HPy^+ can dock into both the crystalline and amorphous regions. We have revised our manuscript as follow and included above discussions in Section 4 of the Supplementary Information:

273 of the films exchanged with different cations (Fig. S28). These results indicate that the
274 counterion exchange did not disrupt the microstructure of the polymer films. Nonetheless,
275 we did observe differences in the GIWAXS spacing parameters. The lamellar stacking
276 distance of the polymer films expanded after exchange-doped with the three counterions,
277 indicating the presence of counterions within the crystalline regions^{16,50}. Particularly,
278 polymer films exchange-doped with HPy⁺ and BMIM⁺ consistently exhibited smaller
279 lamellar and π - π stacking distances compared to those exchange-doped with MtBA⁺, across
280 different doping times (Fig. 4b and 4c). This trend was also observed in MD simulations
281 (Fig. S16). The diffusion of MtBA⁺ into the sidechain regions led to increased sidechain
282 disorder (lamellar expansion) and a greater twisting of the polymer backbone (larger π - π

Q2.2. The authors state that the MD simulations “were focused on the disordered region of the polymer. However, if the doping is visible in XRD, which is used as an argument above, it means the dopants go in the ordered regions. This diminishes the importance of the MD simulations.

Our response: Thank you for your advice. We would like to clarify that in our MD simulations, the term “disordered region” does not imply amorphous regions but rather refers to the crystalline region with expanded lamellar distance and larger structural disorder. This approach allows for a more feasible simulation of counterion diffusion. As shown in Fig. R17a, the well-packed supercell represents a crystalline region. We have simulated the diffusion of counterions within a crystalline supercell. However, the presence of dense alkyl sidechains in the polymer matrix presents a challenge for counterion diffusion on the time scale of MD simulations. Both MtBA⁺ and HPy⁺ remain localized around their initial positions due to steric hindrance (Fig. R17). Considering the dopant insertion generally causes the expanding of lamellar stacking, we increased the lamellar distance in the crystalline cell, creating a “disordered region” that facilitated counterion diffusion. In this disorder region, the counterion MtBA⁺ could diffuse into the sidechain region (Fig. R17b), while HPy⁺ still docking around the polymer backbones (Fig. R17a). This differentiation in counterion behavior is crucial for understanding how the choice of dopant can influence the polymer’s electronic properties.

Fig. R17. Snapshots from MD simulations showing the counterion diffusion in the crystalline and disordered region for **a**, P(PzDPP-2FT)-HPy⁺ and **b**, P(PzDPP-2FT)-MtBA⁺. The left panels show a top view, and the right panels show a side view.

It is worth noting that the intrachain charge transport is fast, and the primary limitation to charge transport is the interchain charge transport, which is mainly influenced by molecular packing in the crystalline/disordered regions (*Nat. Mater.* **2020**, *19*, 491–502). Our XRD data corroborate that the counterions can significantly influence the molecular packing within the crystalline region. Therefore, the investigation of the counterion effects on the crystalline/disordered region is more critical than the amorphous region in charge transport.

In Supplementary Information, we have included a clear definition of the “disordered region” to prevent any misunderstanding. We have also emphasized the importance of studying the counterion effects within the crystalline/disordered region, which is supported by both our simulation results and GIWAXS data.

Q3. General issues

Q3.1. There are still misleading Figures such as Fig. 2e as outlined above but also Fig 4i and Fig. 5: less disorder gives higher conductivity and lower thermopower, so the arrow should point down.

Our response: Thank you for your advice. We have taken the following steps to correct the figures and ensure they accurately reflect the findings of our study:

1. **Removal of Fig. 2e:** As you noted, Fig. 2e was potentially misleading and has been removed from the manuscript to prevent any confusion.

2. **Update of Fig. 5:** Fig. 4i clearly illustrates the effect of energetic disorder on thermoelectric performance. Therefore, we have replaced the schematic representation of the Seebeck coefficient (S) versus electrical conductivity (σ) curve with the actual experimental data from Fig. 4i. This change provides a more accurate and empirical basis for understanding the trends observed in our study.

Q3.2. The argument involving AFM images still suffer from the same issue as in our last review: the AFM images look the same, depending on which cutout you look at. This would probably also apply to the c-AFM images if the color scale had not been inverted for the MtBA⁺ image.

Our response: Thank you for your advice. To address your concerns and validate the data, we have taken additional measures to ensure the comprehensiveness of our AFM scans. We have conducted a new series of AFM scans over a larger range ($10 \times 10 \mu\text{m}^2$) and randomly selected three smaller regions ($2 \times 2 \mu\text{m}^2$) to analyze in detail. Upon examination of the larger area scans, we observed particles on the surface of the films, which could be attributed to dust or residual ion liquid from the doping process. Despite the presence of these particles, we did not observe any significant changes in the morphology of the undoped polymer films or those exchange-doped with the three counterions. These results confirm that our AFM images are free from cutout issues and accurately reflect the actual surface characteristics of the polymer films.

Fig. R15. AFM height images of the pristine polymer and the polymer exchange-doped with MtBA^+ , BMIM^+ , and HPy^+ . The left column features AFM height images with an area of $10 \times 10 \mu\text{m}^2$ and the right three columns showcase zoomed-in areas ($2 \times 2 \mu\text{m}^2$) randomly selected from the larger scans.

As for the c-AFM, the remeasurement and rescaling of the c-AFM images for the polymer film exchanged with MtBA^+ , as shown in Fig. R16, presents a clearer picture of the electrical characteristics of the polymer films at the micrometer scale. The updated c-AFM images with a color bar range of 0 pA to 200 pA allow for a more accurate comparison of the electrical current distribution across the different samples. The pristine polymer film appearing almost dark is consistent with its semiconducting nature, indicating the nonconductive property in the undoped state. When exchange-doped with HPy^+ , the surface of the polymer film exhibited large connected areas with high currents. In contrast, when exchange-doped with MtBA^+ , the film showed isolated conducting areas with low currents. We have replaced Fig. S29 in the

Supplementary Information with Fig. R16 here.

Fig. R16. The c-AFM current images of the pristine and the exchange-doped P(PzDPP-2FT) with MtBA⁺, BMIM⁺, and HPy⁺. The measurement was conducted in a nitrogen atmosphere, and the voltage bias was 100 mV.

Reviewer 4:
